# The Geometry of Updates: Fisher Alignment at Vocabulary Scale

**John Sweeney** [1]

## Abstract

Training-free source selection for LLM families with shared vocabularies arises in scientific string domains such as SMILES, protein, and genomic sequences, where candidate corpora share a tokenizer but differ in prediction targets. This creates an activation-dark regime: representation-similarity metrics can be uninformative without assumptions about label-conditioned error geometry, while classical update-geometry metrics are computationally prohibitive at vocabulary scale. We show that, in a shared-output head setting, representation metrics (e.g., CKA) are non-identifiable for transfer; models can share identical representations yet have orthogonal head updates. The key identity is that head Fisher alignment is exactly a cosine between kernel mean embeddings in the joint activation-error space, exposing activation, error, and coupling factors rather than requiring a materialized Fisher matrix. FisherSketch estimates this cosine directly in a single streaming pass, making $K{=}128{,}256$ head Fisher alignment practical with a 16 KB task signature ($m{=}4096$) and a 192 KB per-task streaming state–small enough to store next to a model hash, but encoding transfer-relevant update structure. Beyond source selection, the same signatures and marginals provide a diagnostic instrument for studying whether LLM task similarity is driven by activations, errors, or their coupling; shared-parameter and internal-layer validations, together with Llama-3.1-8B verbalizer-shift experiments, show that FisherSketch remains informative when activation similarity cannot distinguish tasks.

[1] Sideplane AI. Correspondence to: John Sweeney <john.sweeney@sideplane.ai>.

*Proceedings of the $43^{rd}$ International Conference on Machine Learning*, Seoul, South Korea. PMLR 306, 2026. Copyright 2026 by the author(s).

## 1. Introduction

**Activation-dark source selection.** Which source data should adapt a shared-vocabulary LLM for a target string domain such as SMILES, proteins, or genomics? This is the operational role of finite-label, label-aware transferability scores such as LogME/LEEP in small-output settings, but their standard forms are not direct plug-ins at $K \sim 10^5$: they require label maps or task-specific class statistics rather than a streaming estimate of joint activation–error geometry. The hard regime is *activation dark*: tasks share tokens and frozen activations, yet label-conditioned errors point in different update directions.

**Transfer is geometry in update space.** Transfer learning is a counterfactual question: given a pretrained model, how will its parameters move when optimized for a new objective? Despite being counterfactual, transfer prediction often uses representation-similarity scores (e.g., CKA, RSA, and SVCCA) computed from frozen activations on a probe set. These metrics implicitly assume that models (or tasks) that "see" the world similarly will also update similarly. Our results show that this assumption can fail in a head-level setting unless one makes additional assumptions about error geometry or label structure.

**FisherSketch: a telescope for update geometry at vocabulary scale.** We study transfer within shared-parameter families and quantify update similarity using Fisher alignment. The central observation is that head Fisher alignment is an ordinary cosine after the right lift: Theorem 4.2 rewrites it as the cosine between kernel mean embeddings of joint $(a, e)$ features. This turns a Fisher-matrix comparison into a distributional embedding problem while retaining activation geometry, error geometry, and their coupling. At $K{=}128{,}256$, however, even the error second moment $\Gamma_e := \mathbb{E}[ee^\top]$ costs $\approx 61$ GiB per task and naive error projections cost $O(mK)$ per sample. FisherSketch estimates the product-kernel cosine directly in one pass, producing a 16 KB signature (at $m{=}4096$), with a 192 KB split-sample streaming state (Appendix R.1). These task signatures give a portable, training-free source-selection object and a diagnostic probe for studying how activation, error, and coupling geometry organize LLM tasks (Appendix R). This makes vocabulary-scale comparisons practical in the activation-dark setting: activation-only baselines can be competitive

on natural shifts but can also be near-constant under fixed-prefix prompts, while FisherSketch remains informative (Section 6.2).

This head-level focus requires a shared output basis. Head update geometry depends on inner products between logit-space error vectors (e.g., $e_i^\top e_j$). These inner products are only well-defined when tasks share an output coordinate system. For LLM families, this is most naturally a shared vocabulary (e.g., BERT (Devlin et al., 2019), GPT (Radford et al., 2019), T5 (Raffel et al., 2020), Mistral (Jiang et al., 2023)); for disjoint label subsets we use an aligned label taxonomy/masked-softmax embedding (Appendix F). When tasks have fundamentally different output spaces, we recommend full-network or internal-layer comparisons instead of head error geometry.

**Evaluation (source selection).** We evaluate transfer as LoRA source selection across domains (Section 6). Formal metric definitions and protocol details are in Appendix B.11. Crucially, in verbalizer shift (fixed-prefix classification), activation-only collapses to random (20% top-1) while FisherSketch achieves 66.7% top-1 (Section 6.2). On Llama-3.1-8B across 100 domains, FisherSketch is competitive with activation-only baselines on natural shifts (Section 6.1). As a scientific-sequence proof of concept, Appendix R.4 evaluates nine molecular SMILES domains and finds that FisherSketch correlates with cross-domain perplexity reduction ($\rho_s$=0.53, $p$=0.006), while activation-only similarity is not significant ($p$=0.081).

**Why representation similarity can fail.** Representation-only metrics (CKA, RSA, SVCCA) compare activations on a probe set, while head update geometry depends on the joint structure of activations and label-conditioned errors. As a result, representation similarity alone does not determine head Fisher alignment. Theorem 3.2 formalizes this: there exist tasks with identical probe representations ($Z_i = Z_j$) whose head updates are orthogonal, so $A_F^{\text{head}}(i, j) = 0$.

We provide two concrete instantiations. First, in a shared-output masked-softmax embedding used to compare disjoint label sets, non-overlapping label blocks imply disjoint error support, so head gradients are orthogonal even when activations match (Appendix F). Second, in a dense-vocabulary LLM setting (verbalizer shift, Section 6.2), the prompt prefix is fixed so activations are identical under this setup, yet transfer varies because the label token changes error geometry. A compact ViT-B/16 orthogonality check under the shared-output embedding appears in Appendix A (Table 4), and Appendix R.6 shows the effect across partial label overlap. These results are specific to head-level comparisons and do not preclude other forms of transfer (e.g., if one retrains the head). We do not claim CKA is uninformative in all settings; when conditions are favorable (Appendix O), it can correlate with full-network Fisher, but such correlation is not guaranteed.

**Contributions.** We make four contributions. First, we give a non-identifiability witness showing that representation-only metrics cannot determine head Fisher alignment without assumptions on label-conditioned error geometry (Theorem 3.2). Second, we prove that shared-output head Fisher alignment is a product-kernel cosine between joint activation-error mean embeddings, yielding an exact activation/error/coupling decomposition (Theorem 4.2). Third, we introduce FisherSketch, a one-pass streaming random-feature estimator that avoids materializing $K^2$ error moments and produces portable 16 KB task signatures–comparable in size to a small model hash, but encoding transfer-relevant geometry–via SRHT at vocabulary scale. Fourth, we use these signatures as task-geometry probes, separating activation, error, and coupling effects, and validate them on ViT-B/16 and LLM transfer settings, with shared-parameter stress tests in Appendix C.

**Roadmap.** Section 2 defines Fisher alignment and the task setting; Section 3 proves the non-identifiability result; Section 4 derives the product-kernel structure; Section 5 introduces FisherSketch; and Section 6 presents experiments.

## 2. Preliminaries

We consider tasks indexed by $k$, each with a distribution $\mathcal{D}_k$ over $(x, y)$ and a per-example loss $\ell_\theta^{(k)}(x, y)$. Tasks share architecture and parameterization, and we evaluate task $k$ at a checkpoint $\theta_k$ (in the common-checkpoint setting, $\theta_k \equiv \theta_0$). For head-level comparisons, tasks must share an output coordinate system; otherwise $e_i^\top e_j$ and $\langle \Gamma_{e,i}, \Gamma_{e,j} \rangle$ are not defined (Appendix F).

For task $k$, define the per-example gradient $g_\theta^{(k)}(x, y) := \nabla_\theta \ell_\theta^{(k)}(x, y)$ and the data/empirical Fisher

$$\mathcal{F}_k(\theta_k) := \mathbb{E}_{(x,y)\sim\mathcal{D}_k}\left[ g_{\theta_k}^{(k)}(x, y)\, g_{\theta_k}^{(k)}(x, y)^\top \right].$$

Fisher alignment between tasks $i, j$ is the normalized Frobenius inner product

$$A_F(i, j) := \frac{\langle \mathcal{F}_i, \mathcal{F}_j \rangle_F}{\|\mathcal{F}_i\|_F \|\mathcal{F}_j\|_F} = S_{\text{cov}}(\mathcal{F}_i, \mathcal{F}_j), \quad (1)$$

where $S_{\text{cov}}(A, B) := \langle A, B \rangle_F / (\|A\|_F \|B\|_F)$ for nonzero symmetric matrices.

**Hilbert-space view.** The Frobenius inner product makes finite matrices a Hilbert space (Hilbert–Schmidt), so $S_{\text{cov}}(\cdot, \cdot)$ is a cosine. Section 4.2 gives an RKHS analogue: head Fisher alignment is a cosine between kernel mean embeddings in the RKHS of a product kernel over $(a, e)$. Fisher alignment compares second moments (not mean gradients); estimator details are in Appendix D.1, with full preliminaries in Appendix D.

## 3. Why Representation Metrics Fail

Representation similarity metrics (CKA, RSA, SVCCA) are widely used for transfer prediction. Without assumptions on error geometry or label mapping, they are blind to head-level error structure (Appendix K).

**Definition 3.1** (Representation-only metric)**.** A metric $M(Z_i, Z_j) \to [-1, 1]$ is **representation-only** if it depends only on representations $Z_i$ and $Z_j$ computed on a shared probe set, and does not use errors $e$ or gradients $g$. We make no normalization assumptions; for example, $M(Z_i, Z_i)$ need not equal 1.

**Theorem 3.2** (Non-identifiability witness for representation-only metrics)**.** *For any representation-only metric $M$, there exist tasks $i \neq j$ and a shared probe set such that $Z_i = Z_j$ and the head Fisher matrices are nonzero (so each self-alignment equals 1). Under a shared-output masked-softmax embedding with disjoint label blocks, the cross-task head Fisher alignment satisfies $A_{\mathrm{F}}^{\mathrm{head}}(i, j) = 0$ while $A_{\mathrm{F}}^{\mathrm{head}}(i, i) = 1$. Consequently, $M$ cannot distinguish the cross-task pair from the self-pair using representations alone, even though head Fisher alignment can:*

$$M(Z_i, Z_j) = M(Z_i, Z_i),$$
$$A_{\mathrm{F}}^{\mathrm{head}}(i, j) \neq A_{\mathrm{F}}^{\mathrm{head}}(i, i).$$

*Disjoint-label witness.* Under the shared-output embedding, tasks that share an encoder but have disjoint label subsets have identical representations, yet orthogonal head gradients. A full proof is in Appendix N. *Proof sketch.* Under the shared-output masked-softmax embedding, disjoint label blocks imply error vectors have disjoint support, so head gradients are orthogonal even when representations match. Representation-only metrics observe only activations, and therefore cannot detect this. *Scope.* We use masked-softmax to give a clean orthogonality witness, not to claim that every transfer benchmark lies in the zero-overlap regime. More broadly, without additional assumptions linking errors to activations, representation covariances do not identify head Fisher alignment. We do not claim that CKA is uninformative in all regimes. When the diagnostics are favorable (Appendix O), it can correlate with full-network Fisher. However, non-identifiability means it can still fail without assumptions on error geometry.

**Dense-vocabulary instantiation.** While the masked-softmax construction above provides a clean orthogonality witness, Section 6.2 demonstrates the same phenomenon in a fully dense-vocabulary LLM setting. Fixing only the prompt prefix makes representations identical across verbalizers, yet changing the label token alters error geometry and transfer. FisherSketch predicts this variation, while representation-only metrics collapse.

This yields a *non-identifiability* result. Linear CKA admits

the closed form (Appendix D.2)

$$\mathrm{CKA}(Z_i, Z_j) = \frac{\|\hat{\Sigma}_{ij}\|_F^2}{\|\hat{\Sigma}_i\|_F \, \|\hat{\Sigma}_j\|_F}, \tag{2}$$

Thus it depends only on $(\hat{\Sigma}_i, \hat{\Sigma}_j, \hat{\Sigma}_{ij})$ and not on error geometry $\Gamma_e$.

**Corollary 3.3** (Error geometry is not identifiable from CKA)**.** *There is no function $g : [0, 1] \to [0, 1]$ such that for all task pairs with well-defined CKA ($\|\hat{\Sigma}_i\|_F, \|\hat{\Sigma}_j\|_F > 0$),*

$$A_{\mathrm{F}}^{\mathrm{head}}(i, j) = g(\mathrm{CKA}(Z_i, Z_j)).$$

**Implication.** CKA depends only on centered representation covariances, while Fisher alignment also depends on the error second moment $\Gamma_e$. The same non-identifiability holds for RSA and SVCCA (Appendices K, K.1, and K.2). Thus representation-only metrics are fundamentally limited for head-level transfer without assumptions on error geometry or label mapping. A correlation with full-network Fisher in some regimes does not guarantee accurate transfer ranking (Appendix O). Empirically, on ViT-B/16 same-dataset pairs, CKA correlates only modestly with exact head Fisher (Pearson 0.43, Spearman 0.45; Table 5 in the Appendix).

To understand *why* error geometry matters, we next use the product-kernel structure of head Fisher alignment (Section 4). Representation metrics see only activations, whereas Fisher is a cosine in the joint (activation, error) space.

## 4. What Fisher Measures

We show that Fisher alignment has a product-kernel structure: head-layer Fisher alignment is the cosine similarity in a joint (activation, error) feature space, decomposing into representation geometry, error geometry, and a coupling term.

### 4.1. Layer-wise Kronecker Structure of Gradients

Affine layers have an exact Kronecker gradient structure, which underlies K-FAC-style approximations (Heskes, 2000; Martens & Grosse, 2015; Grosse & Martens, 2016). Let $a_{l-1}$ be the input activation and $\delta_l := \partial \ell / \partial z_l$ be the backpropagated error (with $z_l = W_l a_{l-1}$).

**Lemma 4.1** (Kronecker structure for affine layers)**.** *For an affine layer $l$ with weight matrix $W_l$, the per-example gradient satisfies $\nabla_{W_l} \ell = \delta_l a_{l-1}^\top$, or equivalently:*

$$g_l := \mathrm{vec}(\nabla_{W_l} \ell) = a_{l-1} \otimes \delta_l. \tag{3}$$

*Here* vec *stacks columns (MATLAB/Fortran convention), so* $\mathrm{vec}(uv^\top) = v \otimes u$. *Using row-major vectorization swaps the Kronecker factors (a fixed permutation) and leaves all alignment/cosine quantities unchanged. This is an* exact

*identity for affine transformations. Bias gradients can be handled by augmenting $a_{l-1}$ with a constant 1; we focus on weights for clarity.*

Since classification heads are typically affine (linear classifiers), the exact Kronecker structure (3) holds without approximation. For the output layer (head), with **uncentered activation** $a := a_{L-1}$ entering the head and error $e := \delta_L$, we have

$$g_{\text{head}} = a \otimes e. \qquad (4)$$

We analyze head-layer Fisher with respect to the output projection parameters only.[1] Crucially, this uses the *uncentered* activation $a$, not the centered representation $z = a - \mu$ (Appendix M). For shared-parameter affine maps (convolutions, token-wise linear maps), gradients are sums of Kronecker terms across positions; Appendix C gives the exact kernel/mean-embedding view and provides an unbiased FisherSketch variant. The exact activation–error product identity below applies to rank-one/no-sharing heads with a common output basis; for shared-parameter layers we sketch the corresponding shared-parameter Fisher kernel directly, and separable activation/error products are diagnostics rather than identities.

## 4.2. Head Fisher Alignment as a Product-Kernel Mean Embedding

**A product-kernel view.** Recall that for a linear head, the per-example gradient factorizes as $g = a \otimes e$ (Lemma 4.1). For two independent draws $(a, e) \sim \mathcal{D}_i$ and $(a', e') \sim \mathcal{D}_j$, the head Fisher inner product can be written as a *kernel expectation*:

$$\langle \mathcal{F}_i, \mathcal{F}_j \rangle_F = \mathbb{E}\big[(g^\top g')^2\big] = \mathbb{E}\big[(a^\top a')^2 (e^\top e')^2\big].$$

This is a degree-2 polynomial kernel in $g = a \otimes e$ (equivalently degree-4 in $(a, e)$) that factorizes. Define $k_a$, $k_e$, and $k_{ae}$ by

$$
\begin{aligned}
k_a(a, a') &:= (a^\top a')^2, \\
k_e(e, e') &:= (e^\top e')^2, \\
k_{ae}((a, e), (a', e')) &:= k_a(a, a')\, k_e(e, e').
\end{aligned}
$$

Both $k_a$ and $k_e$ are PSD kernels, hence so is their product $k_{ae}$. Kernel perspectives also appear in wide-network limits and neural tangent kernels (Jacot et al., 2018; Lee et al., 2019). A convenient explicit feature map is obtained by

---

[1]With tied input/output embeddings, this ignores the additional input-embedding gradient term, but the head gradient for the output matrix still satisfies (4) exactly.

symmetric vectorization:[2]

$$
\begin{aligned}
\phi_a(a) &:= \text{vec}_{\text{sym}}(aa^\top), \\
\phi_e(e) &:= \text{vec}_{\text{sym}}(ee^\top), \\
\phi(a, e) &:= \phi_a(a) \otimes \phi_e(e),
\end{aligned}
$$

so that $k_{ae}((a, e), (a', e')) = \langle \phi(a, e), \phi(a', e') \rangle$.

For each task $k$, define the (uncentered) *kernel mean embedding*

$$\mu_{ae,k} := \mathbb{E}_{(a,e) \sim \mathcal{D}_k}\big[\phi(a, e)\big].$$

Assume finite fourth moments and nonzero embedding norms, so the cosines below are well-defined; in practice we use $\varepsilon$-regularized norms for degenerate cases. We write $S_{ae}(i, j) := \langle \mu_{ae,i}, \mu_{ae,j} \rangle$ and define $S_a, S_e, \chi, \rho$ analogously in Appendix B.

**Theorem 4.2** (Head Fisher alignment is a product-kernel cosine). *With the definitions above,*

$$
\begin{aligned}
\langle \mathcal{F}_i, \mathcal{F}_j \rangle_F &= \langle \mu_{ae,i}, \mu_{ae,j} \rangle, \\
A_F^{\text{head}}(i, j) &= \frac{\langle \mu_{ae,i}, \mu_{ae,j} \rangle}{\|\mu_{ae,i}\|\, \|\mu_{ae,j}\|} \\
&= \cos(\mu_{ae,i}, \mu_{ae,j}).
\end{aligned}
$$

*Proof.* Since $g = a \otimes e$, we have $g^\top g' = (a^\top a')(e^\top e')$. Therefore,

$$
\begin{aligned}
\langle \mathcal{F}_i, \mathcal{F}_j \rangle_F &= \mathbb{E}[(a^\top a')^2 (e^\top e')^2] \\
&= \mathbb{E}[\langle \phi(a, e), \phi(a', e') \rangle] \\
&= \langle \mathbb{E}[\phi(a, e)], \mathbb{E}[\phi(a', e')] \rangle \\
&= \langle \mu_{ae,i}, \mu_{ae,j} \rangle.
\end{aligned}
$$

Normalization gives the cosine form. $\qquad \square$

When the joint distribution is separable ($\mu_{ae} = \mu_a \otimes \mu_e$), Fisher alignment factors into activation and error cosines. The coupling term $\rho$ measures deviation from separability.

**Three-factor form:** rank-1 proxy **plus** non-separability correction. Define the marginal embeddings

$$
\begin{aligned}
\mu_{a,k} &:= \mathbb{E}[\phi_a(a)] = \text{vec}_{\text{sym}}(M_{a,k}), \\
\mu_{e,k} &:= \mathbb{E}[\phi_e(e)] = \text{vec}_{\text{sym}}(\Gamma_{e,k}),
\end{aligned}
$$

where $M_{a,k} := \mathbb{E}[aa^\top]$ and $\Gamma_{e,k} := \mathbb{E}[ee^\top]$. We use $S_{\text{cov}}(A, B) := \langle A, B \rangle_F / (\|A\|_F \|B\|_F)$ for symmetric $A, B$; here $A$ and $B$ are uncentered moments ($M_a, \Gamma_e$). If the joint embedding were *separable* (rank-1) in the product space, then $\mu_{ae,k} = \mu_{a,k} \otimes \mu_{e,k}$. In that case,

$$
\begin{aligned}
\cos(\mu_{ae,i}, \mu_{ae,j}) &= \cos(\mu_{a,i}, \mu_{a,j}) \cos(\mu_{e,i}, \mu_{e,j}) \\
&= S_{\text{cov}}(M_{a,i}, M_{a,j}) \cdot S_{\text{cov}}(\Gamma_{e,i}, \Gamma_{e,j}).
\end{aligned}
$$

---

[2]We use upper-triangular vectorization with off-diagonals scaled by $\sqrt{2}$ (Lemma B.1, Appendix B), so $\langle \text{vec}_{\text{sym}}(aa^\top), \text{vec}_{\text{sym}}(a'a'^\top) \rangle = (a^\top a')^2$.

In general, the joint embedding need not be separable, so we define the *non-separability ratio*

$$\rho_{ij} := \frac{A_{\mathrm{F}}^{\mathrm{head}}(i,j)}{S_{\mathrm{cov}}(M_{a,i}, M_{a,j}) \cdot S_{\mathrm{cov}}(\Gamma_{e,i}, \Gamma_{e,j})}$$

(well-defined when the denominator is nonzero),

If $S_{\mathrm{cov}}(M_{a,i}, M_{a,j}) = 0$ or $S_{\mathrm{cov}}(\Gamma_{e,i}, \Gamma_{e,j}) = 0$, then $A_{\mathrm{F}}^{\mathrm{head}}(i,j) = 0$ and we set $\rho_{ij} = 1$ by convention. Empirically, we estimate this via $\hat{\rho}_{ij}$ using sample averages; see Appendix J. The following identity holds exactly:

$$\boxed{A_{\mathrm{F}}^{\mathrm{head}}(i,j) = S_{\mathrm{cov}}(M_{a,i}, M_{a,j}) \cdot S_{\mathrm{cov}}(\Gamma_{e,i}, \Gamma_{e,j}) \cdot \rho_{ij}.}$$
(5)

**Practical note.** Dropping the coupling term $\rho$ in Eq. (5) yields the separable/Kronecker proxy; this can mis-rank transfer, and on ViT-B/16 the proxy is negatively correlated with exact head Fisher (Appendix J, Table 5), so we do not rely on it in evaluation.

**Corollary 4.3** (Independence special case)**.** *If $a$ and $e$ are independent within each task (in the degree-2 feature space), then $\mu_{ae,k} = \mu_{a,k} \otimes \mu_{e,k}$ and thus $\rho_{ij} = 1$ for all pairs.*

The non-separability ratio $\rho_{ij}$ captures how the joint embedding deviates from the rank-1 proxy. It can be adversarial; see Appendix J.

*Remark* 4.4 (Coupling residual)**.** Let $R_k := \mathbb{E}[(a \otimes e)(a \otimes e)^\top] - M_{a,k} \otimes \Gamma_{e,k}$. When $R_k = 0$ (activation–error independence), the Kronecker factorization holds exactly; empirical magnitudes are reported in Appendices I.3 and J.

*Remark* 4.5 (Proxy approximation bound)**.** Let $C_k := \mu_{ae,k} - \mu_{a,k} \otimes \mu_{e,k}$ and $\varepsilon_k := \frac{\|C_k\|}{\|\mu_{a,k}\| \|\mu_{e,k}\|}$. If $\varepsilon_i, \varepsilon_j < 1$ and $\eta_k := 2\varepsilon_k/(1 - \varepsilon_k)$, then

$$\left| A_{\mathrm{F}}^{\mathrm{head}}(i,j) - A_{\mathrm{proxy}}(i,j) \right| \leq \eta_i + \eta_j + \eta_i \eta_j.$$

Appendix J.1 provides the proof and uses Appendix L.

### 4.3. Beyond the Head: Full-Network Diagnostics

To relate head/block proxies to full-network Fisher alignment, Appendix O partitions parameters into $L$ blocks (e.g., layers) and decomposes each task Fisher as $F_k = D_k + O_k$, where $D_k$ contains within-block terms and $O_k$ contains cross-block terms. For any tasks $i, j$, normalized Frobenius alignment decomposes exactly as

$$A_F^{\mathrm{full}}(i,j) = w_{\mathrm{blk}}\, A_F^{\mathrm{blk}}(i,j) + w_{\mathrm{off}}\, A_F^{\mathrm{off}}(i,j),$$
$$w_{\mathrm{blk}} := \frac{\|D_i\|_F \|D_j\|_F}{\|F_i\|_F \|F_j\|_F},$$
(6)
$$w_{\mathrm{off}} := \frac{\|O_i\|_F \|O_j\|_F}{\|F_i\|_F \|F_j\|_F}.$$

Here $A_F^{\mathrm{blk}}$ is the alignment computed from block-diagonal Fisher blocks and $A_F^{\mathrm{off}}$ is the alignment computed from off-diagonal blocks. The *profile cosine* $c_r := w_{\mathrm{blk}} + w_{\mathrm{off}} \in (0, 1]$ measures how similarly the two tasks distribute Fisher energy between block-diagonal and cross-block terms (equivalently, the cosine between $(\|D_i\|_F, \|O_i\|_F)$ and $(\|D_j\|_F, \|O_j\|_F)$). The *off-diagonal discrepancy* $\Delta_{\mathrm{off}} := \left| A_F^{\mathrm{off}} - A_F^{\mathrm{blk}} \right|$ captures whether cross-block interactions behave adversarially relative to the block-diagonal geometry. These quantities yield the deterministic certificate

$$\left| A_F^{\mathrm{full}} - A_F^{\mathrm{blk}} \right| \leq (1 - c_r) + w_{\mathrm{off}}\, \Delta_{\mathrm{off}}.$$
(7)

Table 14 (Appendix O) summarizes these diagnostics and the observed full-to-block gaps.

**How to read the diagnostics.** The certificate fails with low profile cosine or large off-diagonal discrepancy. In our ViT/LLM validations, profile cosines are high ($c_r \geq 0.90$), so proxy tightness is typically limited by the off-diagonal discrepancy term; see Table 14 (Appendix O).

**Computational consequence (Hilbert-space/RKHS view).** Theorem 4.2 can be read as a kernel mean embedding identity in an RKHS (i.e., a Hilbert space). Define the product polynomial kernel $k_{ae}((a,e),(a',e')) := (a^\top a')^2 (e^\top e')^2$, and let $\mathcal{H}_{ae}$ be its RKHS with feature map $\Phi$. Then $\mu_{ae,k} := \mathbb{E}_{(a,e) \sim \mathcal{D}_k}[\Phi(a,e)] \in \mathcal{H}_{ae}$ and $A_{\mathrm{F}}^{\mathrm{head}}(i,j) = \langle \mu_{ae,i}, \mu_{ae,j} \rangle_{\mathcal{H}_{ae}} / (\|\mu_{ae,i}\|_{\mathcal{H}_{ae}} \|\mu_{ae,j}\|_{\mathcal{H}_{ae}})$. Identifying $\Phi$ with our explicit polynomial map $\phi$ recovers the finite-dimensional product feature space statement. FisherSketch exploits this by approximating each $\mu_{ae,k}$ with an $m$-dimensional random feature map $\Psi_m$, so all $T^2$ alignments become dot products between compact task signatures. *Crucially, we do not need to compute the three factors separately; instead, we estimate the joint embedding directly in the random feature space.*

## 5. FisherSketch: Scalable Estimation

The product-kernel view suggests a streaming estimator. We fix a random feature map $\Psi_m$ and estimate $\tilde{\mu}_k := \mathbb{E}_{(a,e) \sim \mathcal{D}_k}[\Psi_m(a,e)]$. We then compute alignments as dot products in $\mathbb{R}^m$. The exact embedding $\mu_{ae,k}$ has dimension $O(d^2 K^2)$, which is too large to store, so we use *factored* Random Maclaurin features that exploit the Kronecker structure $g = a \otimes e$.

**Proposition 5.1** (Factored sketch for Kronecker products)**.** *Let $r_a, r_a' \in \{-1, +1\}^d$ and $r_e, r_e' \in \{-1, +1\}^K$ be mutually independent Rademacher vectors. Define the sketch coordinate:*

$$\psi(a,e) := (r_a^\top a)(r_e^\top e)(r_a'^\top a)(r_e'^\top e).$$

*Then (over the sketch randomness) $\mathbb{E}[\psi(a,e)\psi(a',e')] = (a^\top a')^2 (e^\top e')^2$. With $m$ i.i.d. Random Maclaurin coordi-*

*nates (dense RM), define the normalized sketch vector*

$$\Psi_m(a,e) := \frac{1}{\sqrt{m}} \left[ \psi_1(a,e), \ldots, \psi_m(a,e) \right] \in \mathbb{R}^m.$$

*Then* $\mathbb{E}[\langle \Psi_m(a,e), \Psi_m(a',e') \rangle] = (a^\top a')^2 (e^\top e')^2.$

This requires only $O(m(d+K))$ operations per sample (rather than $O(mdK)$), because projections onto $a$ and $e$ are computed separately.

**Algorithm (in words).** Sample Rademacher projections for $a$ and $e$. For each task, stream samples and compute $\psi_j(a,e)$ for $j = 1, \ldots, m$. Accumulate and normalize these values to obtain a task signature $\hat{\mu}_k$. Cosines between signatures estimate head Fisher alignment. Pseudocode and implementation details are in Algorithm 1 (Appendix).

**Estimation (informal).** For dense Random Maclaurin features, the unnormalized kernel inner product is unbiased with variance $O(1/m)$; SRHT preserves the same first-moment target but uses dependent coordinates, and the normalized cosine is a plug-in estimate. Monte Carlo sampling error decays as $O(1/\sqrt{n})$ for $n$ samples per task. We use split-half log-ratio estimation with shared projections (a low-variance within-split estimator; we also report a cross-fit variant), U-statistic diagonal correction, and small-norm clamping for stability. For robust rankings at $K \approx 10^5$, we further stabilize the coupling term by empirical-Bayes shrinkage in log space: let $\log \hat{\rho}_{ij} := \frac{1}{2}(\log \hat{\rho}_{ij}^A + \log \hat{\rho}_{ij}^B)$ and $\mathrm{SE}_{ij}^2 := \frac{1}{4}(\log \hat{\rho}_{ij}^A - \log \hat{\rho}_{ij}^B)^2$; then $w_{ij} = \frac{\tau^2}{\tau^2 + \mathrm{SE}_{ij}^2}$ and $\log \hat{\rho}_{ij}^{\mathrm{shr}} = w_{ij} \log \hat{\rho}_{ij}$ (details in Appendix B.5).

**Scaling and memory.** At $K = 128,256$, storing $\Gamma_e$ requires $\sim 61$ GiB per task, whereas FisherSketch stores a 16 KiB float32 per-task signature. During streaming, split-half estimation keeps six float64 accumulators per task (A/B splits for $S_{ae}, S_a, S_e$), which is 192 KiB. Fixed projections fit on a single GPU by (i) storing dense Rademacher sign matrices as int8 (two $m \times d$ matrices for activations; 32 MiB at $m=d=4096$) and (ii) using an SRHT for the $K$-dimensional error (about 1 MiB of sign-vectors/indices), implemented with a batched Walsh–Hadamard butterfly on GPU (Appendices B.8–B.9). This yields per-sample cost $O(md + K \log K + m)$ in the vocabulary-scale setting. Dense Rademacher and Gaussian projections are used for controlled small-dimensional estimator comparisons; vocabulary-scale LLM error sketches use the SRHT backend on the $K$-dimensional error side.

**Estimator validation.** We validate the estimator on ViT-B/16 using 20 fine-tuned checkpoints across four datasets. We compare sketched estimates with exact computation at moderate $K$. Spearman correlations are $0.97 \pm 0.01$ for $\hat{S}_{ae}$, $0.94 \pm 0.01$ for $\hat{\chi}$, and $0.95 \pm 0.01$ for $\hat{\rho}$. $\hat{A}_F$ is lower $(0.79 \pm 0.06)$ due to ratio normalization. Figure 1 shows accuracy versus sketch dimension, and Table 6 in Appendix A

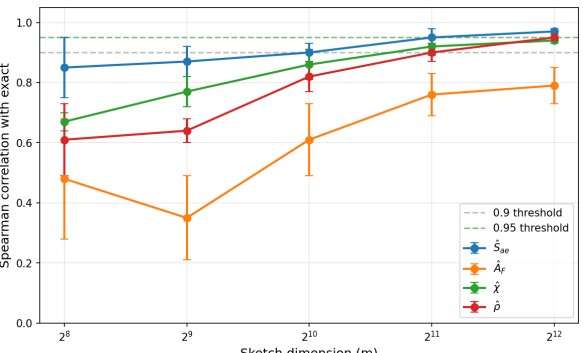

*Figure 1.* Spearman correlations on ViT-B/16 (same-dataset pairs). $\hat{S}_{ae}$ (joint similarity), $\hat{\chi} = \hat{S}_{ae}/(\hat{S}_a \hat{S}_e)$, and $\hat{\rho}$ (normalized coupling) exceed 0.94 at $m = 4096$, while $\hat{A}_F$ (Fisher-alignment cosine) is lower ($\sim 0.79$) due to ratio normalization.

shows linear-time scaling. At $n = 2000$ and $m = 4096$, the method achieves an $89\times$ speedup. For shared-parameter layers, we validate the corresponding shared-layer kernel rather than the no-sharing head factorization: at $m = 4096$, FisherSketch preserves exact-Fisher rankings with Spearman 0.997–0.998 on ResNet-18 convolutional layers and 0.994–0.997 on ViT-B/16 MLP/QKV layers (Appendix C, Table 10).

# 6. Experiments

We evaluate FisherSketch along three axes: (i) sketch accuracy relative to exact computation for moderate $K$ (Fig. 1; Table 6 in Appendix A), (ii) vocabulary-scale source selection, and (iii) ablations that isolate activation vs. error geometry. This section focuses on (ii) and (iii). See Appendices E, I, and S for experimental details, estimator diagnostics, and additional calibration checks. We also evaluate training-free task retrieval and open-set addition using task signatures (Appendix R) and a small SMILES scientific-sequence proof of concept (Appendix R.4).

**Evaluation metrics and protocol.** We evaluate source selection using log-perplexity improvements under LoRA fine-tuning. Let $\ell_t^{\mathrm{base}}$ denote the base model's log-perplexity on target domain $t$, $\ell_t^{\mathrm{self}}$ the log-perplexity after training on $t$ itself, and $\ell_{s \to t}$ the log-perplexity after training on source $s$ and evaluating on $t$. We define normalized transfer as

$$U_{s \to t} = \frac{\ell_t^{\mathrm{base}} - \ell_{s \to t}}{\max(\ell_t^{\mathrm{base}} - \ell_t^{\mathrm{self}}, 0.01)}.$$

This quantity measures the fraction of the target's self-training gain recovered by transferring from $s$; it is negative when transfer hurts. For each target $t$, the oracle source is $s^\star(t) = \arg\max_s U_{s \to t}$, and we report Top-1/Top-3 accuracy for identifying $s^\star(t)$ from the fixed candidate set. Given a predicted source $s_{\mathrm{pred}}(t)$, define per-target regret

*Table 1.* **Vocabulary-scale transfer prediction for source selection on Llama-3.1-8B.** We evaluate source selection over 100 domains with a shared vocabulary ($K{=}128{,}256$) using 24 fixed candidates per target (Appendix B.11). Entries report mean $\pm$ std over seeds 43–45. FisherSketch uses SRHT projections for the $K$-dimensional error features and dense projections for activations. $S_{\mathrm{cov}}(\Gamma_e)$ uses $\Gamma_e = \mathbb{E}[ee^\top]$; $S_{\mathrm{cov}}(M_a)$ is the uncentered activation second-moment alignment. *Caveat:* activation-only can be near-constant under fixed-prefix prompts; see Section 6.2 and Fig. 2.

| Method | Top-1 | Top-3 | Max Regret↓ | % Oracle |
|---|---|---|---|---|
| Oracle | 100% | 100% | 0 | 100% |
| FisherSketch | 45.7%±5.0 | 87.3%±1.2 | 0.119±0.013 | 98.44%±0.30 |
| $S_{\mathrm{cov}}(\Gamma_e)$ alone | 41.3%±1.7 | 86.7%±2.1 | 0.210±0.011 | 97.70%±0.20 |
| $S_{\mathrm{cov}}(\mathrm{diag}(\Gamma_e))$ proxy | 44.0%±2.2 | 87.7%±2.1 | 0.210±0.011 | 97.65%±0.24 |
| $S_{\mathrm{cov}}(M_a)$ alone | 46.3%±3.7 | 85.0%±3.3 | 0.114±0.006 | 98.42%±0.07 |
| PPL similarity | 28% | – | – | – |
| Random (1/24) | 4.2% | 12.5% | – | 14.8% |

$r_t := U_{s^\star(t)\to t} - U_{s_{\mathrm{pred}}(t)\to t}$ and report **Max Regret** (a tail-risk metric) $:= \max_t r_t$. We also report **Percent Oracle**, the mean over targets of $U_{s_{\mathrm{pred}}(t)\to t}/U_{s^\star(t)\to t}$. As a simple baseline, PPL similarity scores sources by $-|\ell_s^{\mathrm{base}} - \ell_t^{\mathrm{base}}|$.

## 6.1. Vocabulary-Scale Transfer Prediction with Tail-Risk Metrics

Table 1 reports the headline vocabulary-scale source selection results; we summarize and then unpack them below. FisherSketch compresses each task from $\sim$61 GiB to a 16 KB float32 signature at vocabulary scale ($K{=}128{,}256$); the streaming estimator maintains a 192 KB per-task state for split-sample estimation (Appendix R.1). On Llama-3.1-8B with 100 domains drawn from eight HF datasets, FisherSketch (SRHT) achieves 45.7% $\pm$ 5.0 top-1 source selection ($\approx$10.9$\times$ the random baseline) with max regret 0.119 $\pm$ 0.013 (Table 1; Appendix B.11). It reaches 98.44% $\pm$ 0.30 of oracle normalized transfer. The per-target Spearman correlation with normalized transfer is 0.47 $\pm$ 0.02 (Appendix B.11). Holding LoRA outcomes fixed, sketch stability across SRHT seeds is high (off-diagonal Spearman 0.982 $\pm$ 0.003; Appendix B.11). The strongest trivially scalable baseline (PPL similarity) reaches 28% top-1 source selection, suggesting that error geometry adds signal beyond perplexity matching. Activation-only geometry is competitive on natural shifts, but Section 6.2 shows fixed-prefix prompt regimes where it collapses and FisherSketch remains informative. As a controlled sanity check, in a small-$K$ linear-probe sweep (five overlap levels), $S_{\mathrm{cov}}(\Gamma_e)$ correlates 0.977 with transfer gain, matching LogME (0.976) and exceeding LEEP (0.964; Appendix O.7). Other label-aware baselines are similarly high because overlap varies monotonically across the sweep (Table 15; Appendix O.7).

**Ablation: activation vs. error geometry.** At 500 samples,

*Table 2.* **Verbalizer-shift source selection** (Llama-3.1-8B; 3 datasets $\times$ 3 seeds = 9 runs; 6 verbalizers per run). Activations are identical under the fixed-prefix setup ($S_{\mathrm{cov}}(M_a) \approx 1$), making activation-only scores constant. Error geometry remains informative: both FisherSketch and the error-only marginal $S_{\mathrm{cov}}(\Gamma_e)$ predict LoRA transfer.

| Method | Top-1 | % Oracle | Random |
|---|---|---|---|
| FisherSketch | 66.7% | 95.7% | – |
| $S_{\mathrm{cov}}(\Gamma_e)$ alone | 72.2% | 100.0% | – |
| $S_{\mathrm{cov}}(M_a)$ (constant) | 20% | – | 20% |

error geometry $S_{\mathrm{cov}}(\Gamma_e)$ reaches 41.3% $\pm$ 1.7 top-1 with max regret 0.210 $\pm$ 0.011. Activation geometry performs better: 46.3% $\pm$ 3.7 top-1 with lower tail risk (max regret 0.114 $\pm$ 0.006). The diagonal error proxy matches the error-only tail risk (max regret 0.210 $\pm$ 0.011). FisherSketch (SRHT) remains competitive (45.7% $\pm$ 5.0 top-1; max regret 0.119 $\pm$ 0.013) and retains high oracle performance (98.44% $\pm$ 0.30). Conditions under which an error-only metric can win are in Appendix P. We report the activation-only marginal $S_{\mathrm{cov}}(M_a)$ (not probe-centered CKA), computed in the same streaming pass as FisherSketch. Practically, if $S_{\mathrm{cov}}(M_a)$ is nearly constant across candidate sources (e.g., fixed-prefix prompt regimes), activation-only scores are uninformative for ranking, whereas FisherSketch remains informative via label-conditioned error geometry (Section 6.2). Table 1 should not be read as uniform dominance over all proxy baselines: on these natural shifts, the activation-only marginal has slightly higher top-1 accuracy than FisherSketch, while FisherSketch has competitive tail risk and oracle-normalized transfer. The intended use case is the regime where a marginal signal is misleading or incomplete, and activation, error, and coupling geometry should be considered jointly.

## 6.2. Identifiability Stress Test: Verbalizer Shift

Table 1 shows that activation geometry can be competitive on natural domain shifts. We next study verbalizer shift under fixed-prefix prompting, where activation similarity is constant. For each dataset (BoolQ, RTE, CB), we fix an identical prompt prefix ending with `The answer is` and vary only the single-token verbalizer (e.g., `Yes/No`, `True/False`, `Correct/Wrong`, `No/Yes`). By causality, the answer-position hidden state depends only on the shared prefix, so $S_{\mathrm{cov}}(M_a) \approx 1$ for all verbalizer pairs.

LoRA transfer varies sharply across verbalizers, and FisherSketch predicts this by sketching errors under the full $K{=}128{,}256$ softmax used to compute $U_{s\to t}$ (token loss on the designated verbalizer, not a renormalized two-way softmax). Across 9 runs (3 seeds $\times$ 3 datasets), FisherSketch achieves 66.7% top-1 source selection and 95.7% of oracle normalized transfer (Table 2; Fig. 2), while $S_{\mathrm{cov}}(M_a)$ re-

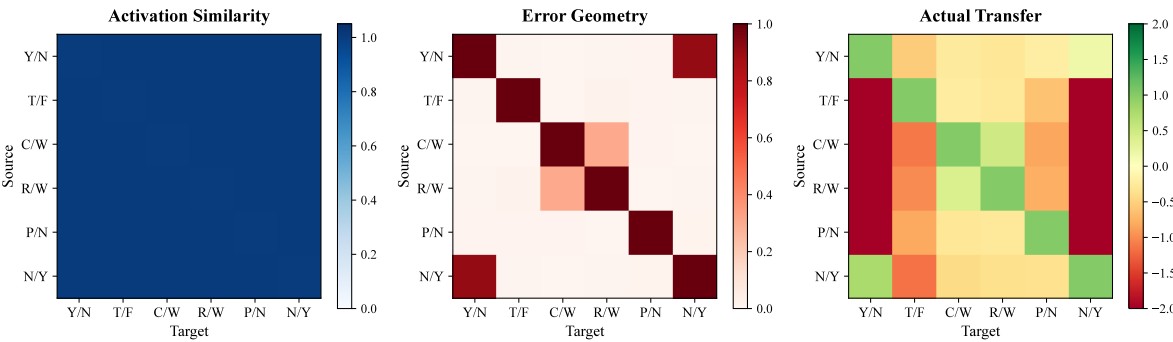

*Figure 2.* **When Activation Similarity Fails.** Activation similarity $S_{\mathrm{cov}}(M_a)$ (left) is constant across all verbalizer pairs, yet actual transfer (right) varies dramatically. Error geometry (middle) captures this variation. Under fixed-prefix prompting, representation-only metrics provide no signal for source selection. (Llama-3.1-8B, BoolQ, seed 42.)

duces to the random baseline. In this controlled regime, the error-only marginal $S_{\mathrm{cov}}(\Gamma_e)$ is also strong (72.2% top-1; 100% of oracle), as expected from the three-factor identity: when $S_{\mathrm{cov}}(M_a)$ is constant, the ranking signal must come from label-conditioned error geometry. This is a dense-vocabulary instantiation of Theorem 3.2. Holding representations fixed does not determine head update geometry; the missing information is label-conditioned error structure, which FisherSketch measures at vocabulary scale. Full experimental details and per-run breakdowns are in Appendix U. Here the error-only marginal is especially strong and exceeds FisherSketch on top-1, which is consistent with the decomposition: when activation geometry contributes no ranking signal, the error marginal can be sufficient.

**Protocol and auditability.** We release the fixed candidate sets and sketch sample indices, and enforce disjoint sketch vs. LoRA splits; correlations use the union of evaluated pairs, while top-1 and regret use the uniform 24-candidate protocol (Appendix B.11). Additional controlled-regime experiments and full-network validation details are in Appendices O.7–O.9.

## 7. Discussion

**Perspective.** FisherSketch makes the non-identifiability insight operational: it turns label-conditioned head update geometry into a comparable object at vocabulary scale by embedding each task as a random-feature approximation to the product-kernel mean embedding (Theorem 4.2). In verbalizer shift, this remains informative precisely when representation similarity is constant under fixed-prefix prompts (Section 6.2).

**Byproduct.** A separable proxy FA-CKA $:=$ CKA $\cdot$ $S_{\mathrm{cov}}(\Gamma_e)$ follows when the joint embedding is approximately rank-1. However, it requires $K^2$ storage, so we do not use it at vocabulary scale (Appendix G). Beyond source selection, the same $\ell_2$-normalized signatures serve as a task-

geometry probe: their activation marginal, error marginal, and coupling ratio ask whether task neighborhoods are representation-driven, error-driven, or activation-dark (Appendices R.3 and S.1). They also support nearest-neighbor task retrieval and open-set addition without retraining (Appendices R and R.2; 55.5% top-1 on 700-domain Natural Instructions retrieval).

### 7.1. Limitations & Broader Applicability

Our formal results concern *head (final-layer)* empirical Fisher alignment, where per-example gradients satisfy $g = a \otimes e$. For deep networks, Appendix O provides a full-network decomposition and diagnostics (profile cosine $c_r$ and off-diagonal discrepancy $\Delta_{\mathrm{off}}$), with the exact depth-coupling form in Appendix O.6; Appendix B.7 describes a structured-projection variant for sketching full-network FisherSketch. Our ViT and LLM validations satisfy these diagnostics; for new architectures, we recommend reporting $(c_r, \Delta_{\mathrm{off}})$.

Head-level error geometry further requires an aligned output space (shared vocabulary or label taxonomy); otherwise error alignment needs an explicit output mapping (Appendix F). For heterogeneous outputs, head Fisher is not directly comparable across tasks. Appendix T reports internal-layer directional Fisher descriptors as a gradient diagnostic; they are complementary to CKA rather than uniformly stronger in that validation.

We use empirical (data) Fisher at a fixed checkpoint, so results may vary across checkpoints, population Fisher, and adaptation methods. Signatures are checkpoint-conditional but require only one streaming pass and scale linearly in streamed tokens (Appendix A; Table 6; e.g., 4.7s for a 200-sample domain signature on an A100; Appendix R.5). The 16 KB figure refers to the stored fp32 descriptor at $m = 4096$; internal shared-token layers require transient $O(nm)$ sketch-feature state before pooling, e.g., about 8 MiB per

task per layer at $n = 500$, $m = 4096$ in fp32. In our heterogeneous-output ViT-B/16 validation, four internal layers across eight models used about 250 MiB of transient sketch-feature state and peaked at 4.19 GiB GPU memory during Fisher collection. The unnormalized random-feature inner-product estimates are unbiased (variance $O(1/m)$ for fixed pairs), while normalized cosines and rankings can degrade for small $n$ or noisy domains; shrinkage, proxy calibration, and sketch-dimension behavior are documented in Appendices B.5, S.1, and S.3.

Our vocabulary-scale evaluation is LoRA transfer on 100 domains under a fixed candidate-set protocol; despite disjoint sampling and released candidate sets/sketch indices for auditability, absolute numbers may shift with domain construction, candidate draws, or transfer hyperparameters (Appendix B.11). We expect strongest applicability in within-family, aligned-output settings (LLMs, subset classification); heterogeneous outputs likely require an output embedding/transport map or full-network/shared-parameter comparisons.

## 7.2. Related Work

CKA (Kornblith et al., 2019), RSA (Kriegeskorte et al., 2008), and SVCCA (Raghu et al., 2017a) compare representations. Related representation-similarity analyses and deconfounded variants have been proposed (Dwivedi & Roig, 2019; Nguyen et al., 2021; Cui et al., 2022), and we formalize their blindness to error geometry. Even after controlling for function-preserving symmetries, representation similarity can remain non-identifiable for label-conditioned update geometry.

Label-aware transfer metrics such as LEEP and LogME use finite-label information (Nguyen et al., 2020; You et al., 2021). Appendix O.7 compares them with $S_{\text{cov}}(\Gamma_e)$ in a small-$K$ overlap sweep, and Appendix Q formalizes LEEP, in a stylized hard-prediction limit, as a diagonal error-geometry proxy; we do not claim that LogME is identical to this Fisher marginal. Operationally, FisherSketch plays the source-selection role of these scores for shared-vocabulary LLMs, but compares joint update geometry instead of small-output evidence or source-model label transfer. Related checkpoint-ranking and transferability scores can scale poorly at $K \sim 10^5$ in their exact form (Li et al., 2021; Huang et al., 2022).

Task2Vec (Achille et al., 2019) embeds tasks via diagonal Fisher. Information-geometric task distances offer another Fisher-style view (Gao & Chaudhari, 2021). Neural Fisher Kernels apply Fisher-kernel ideas to neural networks for data-level representation extraction and low-rank approximations (Zhang et al., 2022); our focus is task-level Fisher alignment under a shared output basis and a streaming activation/error sketch at vocabulary scale. Mean-gradient cosine

compares the first moment $\mathbb{E}[g]$, diagonal-Fisher/task-vector similarities keep only diagonal entries of the second moment $\mathbb{E}[gg^\top]$, and influence-style scores answer pointwise perturbation questions (Appendix S.2). FisherSketch instead sketches the full head-gradient second-moment kernel mean $\mathbb{E}[\phi(a, e)]$, so it can retain off-diagonal error geometry and activation-error coupling while remaining vocabulary-scale. It is therefore best read as a vocabulary-scale source-selection analogue of finite-label transferability scoring, not as a claim that Task2Vec, LogME/LEEP, CKA, or influence baselines are uninformative in their native regimes. K-FAC (Martens & Grosse, 2015) approximates Fisher for optimization, whereas we study cross-task alignment and the coupling term $\rho$ (Theorem 4.2); Appendix H details the connection.

Broader work on transfer examines what is transferred and when transfer fails (Yosinski et al., 2014; Neyshabur et al., 2020; Raghu et al., 2019; Wang et al., 2019b). Multitask and continual learning study task grouping and gradient interactions (Caruana, 1997; Standley et al., 2020; Zenke et al., 2017; Fifty et al., 2021; Yu et al., 2020; Chen et al., 2018), Fisher-based continual-learning regularization (EWC) (Kirkpatrick et al., 2017), and sequential fine-tuning order effects (Sweeney, 2026). These order-level analyses are complementary to FisherSketch: they study how updates compose over time, while FisherSketch supplies a vocabulary-scale pairwise update-compatibility signal.

## 8. Conclusion

Representation-only metrics cannot recover head Fisher alignment without assumptions on error geometry or label mapping (Corollary 3.3). Head Fisher alignment is a product-kernel cosine with a three-factor decomposition that includes a coupling term $\rho$ (Theorem 4.2). FisherSketch makes this alignment tractable at vocabulary scale, compressing each task from $\sim$61 GiB to a 16 KB float32 signature (with a 192 KB per-task streaming state for split-sample estimation); on Llama-3.1-8B it achieves $45.7\% \pm 5.0$ top-1 ($98.44\% \pm 0.30$ of oracle normalized transfer), competitive with activation-only on natural domain shifts and robust when activation geometry collapses (Section 6.2). It thereby serves both as a LogME/LEEP-like training-free source selector in shared-vocabulary activation-dark regimes and as a compact probe for how activation geometry, error geometry, and their coupling organize LLM tasks. Diagnostics connect head alignment to full-network behavior in our ViT/LLM validations (Appendix O.8, Appendix O.9), and the shared-output requirement delineates the scope of head-level comparisons.

## Impact Statement

This work makes update-geometry comparisons and task indexing practical at LLM vocabulary scale by compressing each task into a small signature. That enables new control-plane designs for adaptation—automated fine-tuning source selection, interference-aware continual and multi-task learning, and inference-time routing/open-set expansion—grounded in predicted parameter-update compatibility rather than representation similarity. More broadly, we provide a scalable way to estimate whether parameter updates learned from one dataset/task are likely to help or harm another, supporting better transfer decisions before incurring the cost of full fine-tuning. In practical automated adaptation pipelines, this can reduce negative transfer, avoid wasted training runs, and surface interference patterns and rare but severe regressions earlier, improving reliability while saving compute and energy.

A key risk is that automated source selection or routing policies built on these signals could amplify bias or reduce coverage by systematically deprioritizing underrepresented, safety-critical, or long-tail data that appears "misaligned." Compatibility estimates are also predictive and can be wrong, potentially excluding beneficial sources or creating blind spots. We therefore recommend using update-compatibility signals as one input among others (coverage, fairness, safety evaluation, and domain requirements), and auditing any automated selection/routing mechanism for disparate impact and robustness across minority and edge-case data.

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

# Appendix Guide

The appendices serve three purposes: they prove the identities used in the main text, document the estimator and implementation choices, and give the validation/provenance details that cannot fit in the nine-page body. The guide below is exhaustive at the section level and points to subsection labels when a result is meant to support a specific main-text claim.

**Core Fisher identity and estimator.** Appendix A defines exact head empirical Fisher alignment, the separable Kronecker proxy, and the exact kernel computation used for validation tables. Appendix B gives the linear-time product-kernel mean-embedding view, factored Random Maclaurin sketch, Algorithm 1, empirical-Bayes shrinkage (Appendix B.5), structured full-network sketching (Appendix B.7), SRHT implementation details (Appendices B.8–B.9), and the vocabulary-scale storage/validation setup (Appendix B.11). Appendix C extends the estimator from no-sharing heads to shared-parameter affine maps such as convolutions and token-wise transformer projections.

**Assumptions, proofs, and representation-metric limits.** Appendix D collects Fisher/CKA preliminaries, while Appendix E records training, data, and metric details for the controlled vision experiments. Appendix F states the shared-output masked-softmax embedding used by the non-identifiability witness. Appendices G–J analyze the activation/error/coupling decomposition, its K-FAC relationship, estimator diagnostics, and empirical coupling distributions. Appendices K–N cover RSA/SVCCA non-identifiability, perturbation and mean-correction lemmas, and the full proof of Theorem 3.2.

**Full-network scope and transfer diagnostics.** Appendix O explains how head Fisher relates to block/full-network Fisher, including the exact depth-coupling decomposition (Appendix O.6) and the controlled CIFAR, ViT-B/16, and LLM validations (Appendices O.7–O.9). Appendix P explains why activation geometry can screen candidates while error geometry ranks them, and Appendix Q relates LEEP-style scoring to diagonal error geometry.

**Task signatures, retrieval, and validation diagnostics.** Appendix R documents the 16 KB signature storage claim, retrieval protocol/evaluation, Natural Instructions retrieval, molecular-SMILES proof of concept, dynamic open-set addition, and the label-overlap continuum (Appendices R.1–R.6). Appendix S collects additional validation diagnostics: rho/delta calibration, gradient and diagonal-Fisher baselines, and sketch-dimension sweep (Appendices S.1–S.3). Appendix T covers incompatible output heads and the scoped one-step LoRA/internal-layer diagnostic bridge. Appendix U gives the dense-vocabulary verbalizer-shift experiment, including SRHT backend configuration and compute details.

Captions and subsection introductions state whether each quantity is exact head Fisher, a separable proxy, FisherSketch/SRHT, a controlled dense/Gaussian sketch, a parameter-subsampled full-network diagnostic, or an internal-layer directional descriptor.

## A. Fisher Alignment: Exact vs. Proxy Computation

### A.1. Definitions

For head-layer parameters, the per-example gradient can be written in the exact Kronecker form:

$$g_i^{\text{head}} = a_i \otimes e_i,$$

where $a_i \in \mathbb{R}^d$ is the penultimate activation and $e_i \in \mathbb{R}^K$ is the logit-space error (gradient of the loss with respect to the logits). For cross-entropy, $e_i = \text{softmax}(W a_i) - y_i$. For heads with a bias term, we augment $a_i$ with a constant 1, so $d$ includes the bias. This keeps $g_i^{\text{head}} = a_i \otimes e_i$ unchanged. We use column-major vectorization, so $\text{vec}(e_i a_i^\top) = a_i \otimes e_i$. If a row-major flattening is used in code, it yields a fixed permutation (equivalently $e_i \otimes a_i$), which leaves dot products and alignment cosines invariant.

**Exact head empirical Fisher.** The exact head empirical Fisher is the uncentered second moment of per-example gradients:

$$\mathcal{F}_k^{\text{head}} = \mathbb{E}_{i \sim \mathcal{D}_k}\big[g_i^{\text{head}}(g_i^{\text{head}})^\top\big] = \mathbb{E}\big[(a_i \otimes e_i)(a_i \otimes e_i)^\top\big] = \mathbb{E}\big[(a_i a_i^\top) \otimes (e_i e_i^\top)\big].$$

The **exact head empirical Fisher alignment** (computed via the kernel identity, i.e., without a Kronecker approximation) is:

$$A_{\text{F}}^{\text{exact}}(i, j) = \frac{\langle \mathcal{F}_i^{\text{head}}, \mathcal{F}_j^{\text{head}} \rangle_{\text{F}}}{\|\mathcal{F}_i^{\text{head}}\|_{\text{F}} \, \|\mathcal{F}_j^{\text{head}}\|_{\text{F}}}.$$

This is the empirical estimator of the population head alignment $A_F^{\text{head}}$ used in the theory. We use "exact" to emphasize that no Kronecker approximation is used.

Naively computing this requires materializing a $(dK) \times (dK)$ matrix and scales as $O((dK)^2)$ storage, which motivates the proxy and sketch estimators for large $d$ and $K$.

**Kronecker proxy Fisher.** The Kronecker proxy approximates the expectation of a Kronecker product by a Kronecker product of expectations:

$$\mathcal{F}_k^{\text{head}} = \mathbb{E}\big[(a_i a_i^\top) \otimes (e_i e_i^\top)\big] \approx \mathbb{E}[a_i a_i^\top] \otimes \mathbb{E}[e_i e_i^\top] =: M_{a,k} \otimes \Gamma_{e,k}.$$

This approximation equals the exact Fisher when $a_i$ and $e_i$ are statistically independent. Independence is **generically violated** in trained networks ($\delta \approx 0.34$; Table 12). However, we can still write the exact multiplicative decomposition $A_F^{\text{head}}(i,j) = A_{\text{proxy}}(i,j) \cdot \rho_{ij}$ (Theorem 4.2) when the denominator is nonzero (we $\varepsilon$-regularize in practice). We define the population coupling coefficient as $\rho_{ij} := \chi_{ij}/\sqrt{\chi_{ii}\chi_{jj}}$ (see below for $\chi$). Under separability, the substantive approximation is $\rho_{ij} \approx 1$. Empirically, we estimate it as $\hat{\rho}_{ij} := A_F^{\text{exact}}(i,j)/A_{\text{proxy}}(i,j)$.

The **proxy Fisher alignment** is:

$$A_F^{\text{proxy}}(i,j) = S_{\text{cov}}(M_{a,i}, M_{a,j}) \cdot S_{\text{cov}}(\Gamma_{e,i}, \Gamma_{e,j}),$$

where $S_{\text{cov}}(A,B) := \langle A,B \rangle_F/(\|A\|_F\|B\|_F)$ is the Frobenius cosine for uncentered second moments. It requires only $O(d^2 + K^2)$ *storage* (for the second-moment matrices) instead of $O((dK)^2)$ for the full Fisher. However, the proxy loses the coupling factor $\rho$ and can invert rankings on some architectures (see below).

**FisherSketch.** FisherSketch (Section 5 of main text) uses factored random features to estimate Fisher alignment in a single streaming pass. Unlike the Kronecker proxy, FisherSketch *preserves* the coupling factor $\rho$ by sketching the product kernel $k(g_s, g_t) = (\langle a_s, a_t \rangle \cdot \langle e_s, e_t \rangle)^2$ directly.

**Complexity**: $O(m(d+K))$ operations per sample and $O(m)$ storage per task.

The key insight is **factored projection**. Instead of forming $g = a \otimes e$ (dimension $dK$), we compute

$$\psi_l(a,e) = (r_{a,l} \cdot a)(r_{e,l} \cdot e)(r'_{a,l} \cdot a)(r'_{e,l} \cdot e),$$

exploiting $(r_a \otimes r_e) \cdot (a \otimes e) = (r_a \cdot a)(r_e \cdot e)$. Each of the four dot products is $O(d)$ or $O(K)$; with $m$ features, the total cost is $O(m(d+K))$.

The main text vocabulary-scale experiments use FisherSketch.

## A.2. Which Method is Used Where

**Head-level validation tables use exact Fisher.** The correlations in controlled experiments are computed against the **exact head empirical Fisher alignment** $A_F^{\text{exact}}$.

**Computation method**: We exploit the kernel-alignment interpretation. For head Fisher with gradient $g_i = a_i \otimes e_i$, the Frobenius inner product can be computed as:

$$\langle \mathcal{F}_i, \mathcal{F}_j \rangle_F = \mathbb{E}_{s \sim i, t \sim j}\big[\langle g_s, g_t \rangle^2\big] = \mathbb{E}_{s \sim i, t \sim j}\big[(\langle a_s, a_t \rangle \cdot \langle e_s, e_t \rangle)^2\big].$$

This avoids materializing the $(dK) \times (dK)$ matrix and requires $O(n_i n_j(d+K))$ operations for $n_i, n_j$ samples.

**Implementation**: Exact head Fisher alignment is computed without materializing the Fisher matrix or forming $(dK)$-dimensional gradient vectors, using only inner products:

$$
\begin{aligned}
&\text{for } s \in \{1, \ldots, n_1\},\ t \in \{1, \ldots, n_2\}: \\
&\quad \text{fisher\_exact} \mathrel{+}= (\langle a_s^{(1)}, a_t^{(2)} \rangle \cdot \langle e_s^{(1)}, e_t^{(2)} \rangle)^2 \\
&\quad \text{fisher\_exact} \leftarrow \text{fisher\_exact}/(n_1 n_2 \cdot \|\mathcal{F}_1\|_F \cdot \|\mathcal{F}_2\|_F)
\end{aligned}
$$

We compute $\|\mathcal{F}_k\|_F$ using the same kernel identity on within-task pairs ($s, t \sim k$), which costs $O(n_k^2(d+K))$ per task. We use V-statistics (including $s = t$), so differences from the population identity are $O(1/n)$. For FisherSketch norms in the main text, we apply a U-statistic correction to remove diagonal bias; this does not affect the exact baseline reported here.

*Table 3.* Frozen-backbone label-overlap sweep (shared-backbone MLP; synthetic labels). The Kronecker proxy overestimates exact head Fisher alignment at intermediate overlaps.

| Label overlap | Exact $A_{\mathrm{F}}$ | Proxy $A_{\mathrm{proxy}}$ | Rel. error |
|---|---|---|---|
| 100% | 0.79 | 0.94 | 19% |
| 75% | 0.37 | 0.65 | 77% |
| 50% | 0.35 | 0.57 | 65% |
| 25% | 0.25 | 0.36 | 43% |
| 0% | 0.06 | 0.06 | 0% |

*Table 4.* ViT-B/16 regime sanity check (shared-output masked-softmax). Cross-dataset pairs have orthogonal error geometry under a disjoint-label embedding; CKA remains moderate, whereas $S_{\mathrm{cov}}(\Gamma_e)$ and CKA $\cdot$ $S_{\mathrm{cov}}(\Gamma_e)$ drop to zero.

| Regime | $n$ pairs | CKA | $S_{\mathrm{cov}}(\Gamma_e)$ | CKA$\cdot$ $S_{\mathrm{cov}}(\Gamma_e)$ |
|---|---|---|---|---|
| Same-dataset (same task, diff. seeds) | 40 | 0.75 | 0.89 | 0.67 |
| Cross-dataset (different tasks) | 10 | 0.57 | 0.00 | 0.00 |

**Why exact for validation?** Using exact Fisher eliminates concerns about circularity: the FA-CKA correction (which uses $S_{\mathrm{cov}}(\Gamma_e)$) is validated against Fisher alignment computed *without assuming Kronecker structure*.

A frozen-backbone stress test uses a shared-backbone MLP (fixed features; linear heads) and synthetic labels with controlled overlap. We report both exact and proxy values to show the discrepancy. At 75% overlap, the exact alignment is 0.37 while the proxy is 0.65 (77% relative error), and proxy error ranges from 0–77% across the sweep (Table 3).

**Practical use: Proxy accuracy is architecture-dependent.** Since $A_F^{\mathrm{head}} = A_{\mathrm{proxy}} \cdot \rho$ exactly (Theorem 4.2), a positive $\mathrm{Corr}(\hat{\rho}, A_{\mathrm{proxy}})$ typically coincides with positive proxy–Fisher correlation, while a negative value can indicate inversion (this is a diagnostic, not a guarantee). Unless stated otherwise, $\mathrm{Corr}(\cdot, \cdot)$ denotes Pearson correlation. Here $\hat{\rho}$ denotes the empirical estimate of $\rho$ from finite samples (defined in Appendix J). On ResNet-50 ($n = 75$ same-dataset pairs), $\mathrm{Corr}(\hat{\rho}, A_{\mathrm{proxy}}) = +0.88$ (Pearson), and the proxy correlates positively with exact Fisher (Pearson = 0.98; Spearman = 0.69). On ViT-B/16 ($n = 40$ same-dataset pairs), $\mathrm{Corr}(\hat{\rho}, A_{\mathrm{proxy}}) = -0.85$ (Pearson), and the proxy is negatively correlated with exact Fisher (Pearson = $-0.34$; Spearman = $-0.30$; Table 5). All head-level validation tables use exact Fisher; the proxy is conceptual motivation only. Full-network LLM validation uses parameter-subsampled gradients across all layers for computational tractability.

**Estimator usage across experiments.** Head-level validation tables use exact Fisher via the kernel identity described above (Appendix A). Vocabulary-scale experiments use FisherSketch with factored random features (Section 5). Full-network LLM validation uses parameter-subsampled gradients across layers; Appendix B.7 describes a structured-projection full-network sketch.

**Why proxy shows negative rank correlation on ViT (and why FA-CKA still works).** On ViT-B/16 ($n = 40$ same-dataset pairs), $\mathrm{Corr}(\hat{\rho}, A_{\mathrm{proxy}}) = -0.85$ (Pearson). High-proxy pairs tend to have low $\hat{\rho}$, so the proxy is negatively correlated with exact Fisher (Pearson = $-0.34$; Spearman = $-0.30$; Table 5), indicating rank inversion. FA-CKA remains strongly correlated with exact Fisher (Pearson = 0.79; Spearman = 0.90) because it is validated directly against exact Fisher—it does not use the proxy. CKA alone is only moderately correlated with exact Fisher (Pearson corr. = 0.43, Spearman corr. = 0.45), reflecting stable representations but varying error geometry.

The 40 pairs comprise all same-dataset seed pairs across 4 datasets (CIFAR-100, Flowers102, DTD, Oxford-Pets) with 5 training seeds each. Cross-dataset pairs (10 total, subsampled) have exact Fisher = 0 by construction because their label spaces are disjoint under the shared-output masked-softmax embedding (Assumption F.1). These pairs serve as a sanity check that CKA $\cdot$ $S_{\mathrm{cov}}(\Gamma_e)$ outputs zero in this regime (Table 4).

*Table 5.* ViT-B/16: correlation with exact head Fisher on same-dataset pairs (4 datasets × 5 seeds). Proxy is negatively correlated with exact Fisher; FA-CKA is validated against exact Fisher.

| Method | Pearson corr. | Spearman corr. |
|---|---|---|
| Kronecker proxy | −0.34 | −0.30 |
| CKA | 0.43 | 0.45 |
| FA-CKA | **0.79** | **0.90** |

# B. Linear-Time Head Fisher Alignment via Kernel Mean Embeddings

This appendix develops an $O(n)$-in-samples estimator for the head Fisher alignment quantities in Theorem 4.2. The method applies when the output dimension $K$ is moderate (classification, multiple-choice, retrieval). Vocabulary-scale validation appears in Section B.11.

## B.1. Quadratic Feature Map Identity

**Lemma B.1** (Symmetric vectorization preserves Frobenius inner products). *Define* $\mathrm{vec}_{\mathrm{sym}}(M)$ *as the vector of upper-triangular entries (including the diagonal) of a symmetric matrix* $M \in \mathbb{R}^{d \times d}$, *with off-diagonals scaled by* $\sqrt{2}$. *Then for any symmetric* $A, B$:

$$\langle \mathrm{vec}_{\mathrm{sym}}(A), \mathrm{vec}_{\mathrm{sym}}(B) \rangle = \langle A, B \rangle_F = \mathrm{tr}(A^\top B)$$

*Proof.* For symmetric matrices, the Frobenius inner product is

$$\langle A, B \rangle_F = \sum_i A_{ii} B_{ii} + 2 \sum_{i<j} A_{ij} B_{ij}.$$

The vectorization leaves diagonal entries unchanged and scales off-diagonals by $\sqrt{2}$:

$$\langle \mathrm{vec}_{\mathrm{sym}}(A), \mathrm{vec}_{\mathrm{sym}}(B) \rangle = \sum_i A_{ii} B_{ii} + \sum_{i<j} (\sqrt{2} A_{ij})(\sqrt{2} B_{ij})$$
$$= \langle A, B \rangle_F.$$

$\square$

## B.2. Mean Embedding Representation

Define feature maps:

$$\phi_a(a) := \mathrm{vec}_{\mathrm{sym}}(aa^\top) \in \mathbb{R}^{d(d+1)/2},$$
$$\phi_e(e) := \mathrm{vec}_{\mathrm{sym}}(ee^\top) \in \mathbb{R}^{K(K+1)/2},$$
$$\phi(a, e) := \phi_a(a) \otimes \phi_e(e) \in \mathbb{R}^{d(d+1)/2 \cdot K(K+1)/2}$$

We assume finite fourth moments so the expectations below exist. When we state explicit bounds, we assume $\|a\|_2 \leq R_a$ and $\|e\|_2 \leq R_e$ almost surely.

**Lemma B.2** (Kronecker second-moment structure). *If* $(a, e)$ *are independent, then*

$$\Sigma_{\psi, k} := \mathbb{E}[(a \otimes e)(a \otimes e)^\top] = M_{a,k} \otimes \Gamma_{e,k},$$

*where* $M_{a,k} := \mathbb{E}[aa^\top]$ *and* $\Gamma_{e,k} := \mathbb{E}[ee^\top]$.

*Proof.* By independence, $\mathbb{E}[(a \otimes e)(a \otimes e)^\top] = \mathbb{E}[aa^\top] \otimes \mathbb{E}[ee^\top]$. $\square$

For tasks $i, j$, define the kernel expectations (with independent draws from each task):

$$S_{ae}(i, j) := \mathbb{E}_{(a,e) \sim \mathcal{D}_i, \, (a',e') \sim \mathcal{D}_j} \big[ (a^\top a')^2 (e^\top e')^2 \big],$$
$$S_a(i, j) := \mathbb{E}_{a \sim \mathcal{D}_i, \, a' \sim \mathcal{D}_j} \big[ (a^\top a')^2 \big],$$
$$S_e(i, j) := \mathbb{E}_{e \sim \mathcal{D}_i, \, e' \sim \mathcal{D}_j} \big[ (e^\top e')^2 \big].$$

When the denominators are nonzero, we also define the coupling ratios

$$\chi_{ij} := \frac{S_{ae}(i,j)}{S_a(i,j)\,S_e(i,j)}, \qquad \rho_{ij} := \frac{\chi_{ij}}{\sqrt{\chi_{ii}\chi_{jj}}}.$$

In practice we $\varepsilon$-regularize near-zero denominators (i.e., add $\varepsilon$ before division).

**Corollary B.3** (Fourth moments are inner products of mean embeddings). *For **independent** samples* $(a,e) \sim \mathcal{D}_i$ *and* $(a',e') \sim \mathcal{D}_j$:

$$S_{ae}(i,j) = \langle \mu_{ae,i}, \mu_{ae,j} \rangle \tag{8}$$
$$S_a(i,j) = \langle \mu_{a,i}, \mu_{a,j} \rangle \tag{9}$$
$$S_e(i,j) = \langle \mu_{e,i}, \mu_{e,j} \rangle \tag{10}$$

*where*

$$\mu_{ae,i} := \mathbb{E}[\phi(a,e)],$$
$$\mu_{a,i} := \mathbb{E}[\phi_a(a)],$$
$$\mu_{e,i} := \mathbb{E}[\phi_e(e)].$$

This is the standard kernel mean embedding identity; see, e.g., Smola et al. (2007).

**Key insight**: The $n^2$ barrier is not fundamental—it arises from computing the expectation naively. The true barrier is that $\mu_{ae,i}$ has dimension $O(d^2 K^2)$, which is too large to store exactly.

## B.3. Factored Random Maclaurin Sketching

We use the Random Maclaurin construction (Kar & Karnick, 2012), which provides an unbiased estimator for squared inner products.

**Proposition B.4** (Factored Random Maclaurin for Kronecker products). *Let* $r_a, r'_a \in \{-1,+1\}^d$ *and* $r_e, r'_e \in \{-1,+1\}^K$ *be independent Rademacher vectors. Define the sketch coordinate:*

$$\psi(a,e) := (r_a^\top a)(r_e^\top e)(r'_a{}^\top a)(r'_e{}^\top e)$$

*Then* $\mathbb{E}[\psi(a,e)\psi(a',e')] = (a^\top a')^2(e^\top e')^2.$

*For analysis, define the normalized sketch vector* $\Psi_m(a,e) := \frac{1}{\sqrt{m}}[\psi_1(a,e),\dots,\psi_m(a,e)] \in \mathbb{R}^m$. *Then the kernel estimator*

$$\hat{k}_m := \langle \Psi_m(a,e), \Psi_m(a',e') \rangle = \frac{1}{m}\sum_{j=1}^m \psi_j(a,e)\psi_j(a',e')$$

*is unbiased for* $(a^\top a')^2(e^\top e')^2$ *with variance*

$$\mathrm{Var}[\hat{k}_m] \leq \frac{81}{m}\|a\|^4\|a'\|^4\|e\|^4\|e'\|^4.$$

*Proof of variance bound.* Write $\hat{k}_m = \frac{1}{m}\sum_{j=1}^m X_j$ where $X_j := \psi_j(a,e)\psi_j(a',e')$ are i.i.d. Then $\mathrm{Var}(\hat{k}_m) = \mathrm{Var}(X_1)/m \leq \mathbb{E}[X_1^2]/m$. Expanding $X_1^2$ and using independence across $r_a, r'_a, r_e, r'_e$ yields

$$\mathbb{E}[X_1^2] = \left(\mathbb{E}[(r_a^\top a)^2(r_a^\top a')^2]\right)^2$$
$$\cdot \left(\mathbb{E}[(r_e^\top e)^2(r_e^\top e')^2]\right)^2.$$

For Rademacher $r$, Khintchine gives $\mathbb{E}(r^\top u)^4 \leq 3\|u\|_2^4$, so by Cauchy–Schwarz

$$\mathbb{E}[(r^\top u)^2(r^\top v)^2] \leq 3\|u\|_2^2\|v\|_2^2$$

(tight in the dense aligned limit). Applying this to activation and error terms:

$$\mathbb{E}[X_1^2] \leq (3\|a\|^2\|a'\|^2)^2(3\|e\|^2\|e'\|^2)^2 = 81\|a\|^4\|a'\|^4\|e\|^4\|e'\|^4.$$

$\square$

---

**Algorithm 1** FisherSketch: Streaming Head Fisher Alignment

---

**Require:** Tasks $\{k\}_{k=1}^T$, sketch dimension $m$, samples per task $n_k$
1: Sample i.i.d. Rademacher $r_a^{(j)}, r_a'^{(j)} \in \{-1, +1\}^d, r_e^{(j)}, r_e'^{(j)} \in \{-1, +1\}^K$ for $j = 1, \ldots, m$
2: **for** each task $k$ **do**
3:     $\hat{\mu}_k \leftarrow \mathbf{0} \in \mathbb{R}^m$
4:     **for** each sample $(a, e)$ from task $k$ **do**
5:        $\psi_j \leftarrow (r_a^{(j)\top} a)(r_e^{(j)\top} e)(r_a'^{(j)\top} a)(r_e'^{(j)\top} e)$ for $j = 1, \ldots, m$
6:        $\Psi_m(a, e) \leftarrow \frac{1}{\sqrt{m}}[\psi_1, \ldots, \psi_m]$
7:        $\hat{\mu}_k \leftarrow \hat{\mu}_k + \Psi_m(a, e)$
8:     **end for**
9:     $\hat{\mu}_k \leftarrow \hat{\mu}_k / n_k$
10: **end for**
11: **return** $\hat{A}_F(i, j) = \langle \hat{\mu}_i, \hat{\mu}_j \rangle / (\|\hat{\mu}_i\| \|\hat{\mu}_j\|)$ for all pairs

---

*Proof of streaming mean embedding error.* Fix the random feature map $\Psi_m$ and define $\tilde{\mu}_k := \mathbb{E}_{(a,e) \sim \mathcal{D}_k}[\Psi_m(a, e)]$. The streaming mean embedding is $\hat{\mu}_k = \frac{1}{n} \sum_{t=1}^n \Psi_m(a_t, e_t)$ for i.i.d. samples $(a_t, e_t)$, so by independence

$$\mathbb{E}_{\text{data}} \|\hat{\mu}_k - \tilde{\mu}_k\|_2^2 = \frac{1}{n} \text{Var}_{\text{data}}(\Psi_m(a, e) \mid \text{sketch}) \leq \frac{1}{n} \mathbb{E}_{\text{data}} \|\Psi_m(a, e)\|_2^2.$$

For a single coordinate, $\psi_j(a, e) = (r_a^\top a)(r_e^\top e)(r_a'^\top a)(r_e'^\top e)$, and taking expectation over the sketch yields

$$\mathbb{E}_{\text{sketch}}[\psi_j(a, e)^2] = \|a\|^4 \|e\|^4.$$

Therefore $\mathbb{E}_{\text{data,sketch}} \|\Psi_m(a, e)\|_2^2 = \mathbb{E}_{\text{data}} \|a\|^4 \|e\|^4 \leq R_a^4 R_e^4$, and

$$\mathbb{E}_{\text{data,sketch}} \|\hat{\mu}_k - \tilde{\mu}_k\|_2^2 \leq \frac{R_a^4 R_e^4}{n}.$$

All bounds above are in expectation over the sketch; for a fixed sketch,

$$\mathbb{E}_{\text{data}} \|\hat{\mu}_k - \tilde{\mu}_k\|_2^2 \leq \frac{1}{n} \mathbb{E}_{\text{data}} \|\Psi_m(a, e)\|_2^2,$$

with the right-hand side depending on the realized Rademacher draws. Moreover, for any fixed sketch, $|r_a^\top a| \leq \sqrt{d} \|a\|_2$ and $|r_e^\top e| \leq \sqrt{K} \|e\|_2$, so $|\psi_j(a, e)| \leq dK \|a\|_2^2 \|e\|_2^2$ and

$$\|\Psi_m(a, e)\|_2^2 \leq d^2 K^2 \|a\|_2^4 \|e\|_2^4,$$

which yields $\mathbb{E}_{\text{data}} \|\hat{\mu}_k - \tilde{\mu}_k\|_2^2 \leq d^2 K^2 R_a^4 R_e^4 / n$ for a fixed sketch. $\qquad\square$

**Crucially**, this requires only $O(m(d + K))$ operations per sample—not $O(mdK)$—because we compute $(r_a^\top a)$ and $(r_e^\top e)$ separately.

### B.4. Implementation Details

**Split-sample estimation.** To control ratio bias when computing $\chi = S_{ae}/(S_a \cdot S_e)$, we primarily use split-half log-ratio estimation with shared projections (within-split) for lower variance; we also report a cross-fit variant that computes numerator and denominator on disjoint data splits, which reduces finite-sample bias and yields a consistent estimator as $n \to \infty$; we also report jackknife standard errors as a robustness check.

**U-statistic correction.** For diagonal entries $(i = j)$, the mean embedding inner product includes self-terms $(s = t)$ that overestimate the population quantity. We apply the U-statistic correction to all three matrices $S_{ae}(i, i)$, $S_a(i, i)$, $S_e(i, i)$ (since $\rho_{ij}$ depends on $\chi_{ii}$ and $\chi_{jj}$):

$$\widehat{S}^U = \frac{1}{n(n-1)} \left( \left\| \sum_{s=1}^n \psi(x_s) \right\|^2 - \sum_{s=1}^n \|\psi(x_s)\|^2 \right)$$

where $\psi$ is the feature map for the quantity of interest ($\psi = \Psi_m$, $x_s = (a_s, e_s)$ for $S_{ae}$; $\psi = \phi_a$ with $x_s = a_s$ for $S_a$; and $\psi = \phi_e$ with $x_s = e_s$ for $S_e$), or their corresponding sketches.

**Positivity clamping.** True values $S_a, S_e, S_{ae}$ are nonnegative, but sketch estimates can be slightly negative at modest $m$ due to variance. We clamp: $\hat{S} \leftarrow \max(\hat{S}, \epsilon)$ with fixed $\epsilon = 10^{-10}$ before computing ratios.

### B.5. Noise Robustness via Empirical-Bayes Shrinkage

**Split-half identifiability and Empirical Bayes shrinkage.** For each task $k$, we randomly split its samples into two halves $s \in \{A, B\}$ (50/50 in expectation) and compute $S_{ae}^s, S_a^s, S_e^s$ on each half using the same sketch projections, U-statistic diagonal correction, and the clamping above. Because the sketch is fixed across halves, the diagnostic below conditions on the projection draws and primarily reflects sampling variability. (This split-half diagnostic uses within-split estimates; the cross-fit ratio in the preceding paragraph is a separate bias-control variant.) Define

$$\log \hat{\chi}_{ij}^s := \log S_{ae}^s(i,j) - \log S_a^s(i,j) - \log S_e^s(i,j), \qquad \log \hat{\rho}_{ij}^s := \log \hat{\chi}_{ij}^s - \tfrac{1}{2} \log \hat{\chi}_{ii}^s - \tfrac{1}{2} \log \hat{\chi}_{jj}^s.$$

The split-half reliability diagnostic is the off-diagonal correlation

$$r := \mathrm{Corr}_{i \neq j}\left(\log \hat{\rho}_{ij}^A, \log \hat{\rho}_{ij}^B\right),$$

computed over off-diagonal pairs (equivalently $i < j$ up to duplication). Let $\bar{\ell}_{ij} = \tfrac{1}{2}(\log \hat{\rho}_{ij}^A + \log \hat{\rho}_{ij}^B)$ and estimate per-pair noise via

$$\widehat{\mathrm{SE}}_{ij}^2 = \frac{(\log \hat{\rho}_{ij}^A - \log \hat{\rho}_{ij}^B)^2}{4},$$

the usual variance estimate for the mean under independent halves. Define

$$V_{\text{total}} = \mathrm{Var}_{i \neq j}(\bar{\ell}_{ij}), \quad V_{\text{noise}} = \mathbb{E}_{i \neq j}[\widehat{\mathrm{SE}}_{ij}^2], \quad \tau^2 = \max(0, V_{\text{total}} - c\, V_{\text{noise}}),$$

with conservative $c = 1.5$ (we use $c \in [1.5, 2.0]$). This is a conservative method-of-moments estimate of between-pair variance, inflating noise to encourage shrinkage toward $\rho = 1$ when coupling is weak. The empirical Bayes shrinkage map is

$$w_{ij} = \frac{\tau^2}{\tau^2 + \widehat{\mathrm{SE}}_{ij}^2 + 10^{-10}}, \qquad \log \hat{\rho}_{ij}^{\text{EB}} = w_{ij} \bar{\ell}_{ij}, \qquad \hat{\rho}_{ij}^{\text{EB}} = \exp(\log \hat{\rho}_{ij}^{\text{EB}}),$$

which shrinks toward $\log \rho = 0$ (i.e., $\rho = 1$, the separable proxy). Because shrinkage is applied in log space, $\hat{\rho}_{ij}^{\text{EB}}$ corresponds to the posterior median of $\rho_{ij}$ under the log-normal model. We report the scalar diagnostic $\alpha_{\text{EB}} := \mathbb{E}_{i \neq j}[w_{ij}]$ but apply the pairwise weights in the final estimator. Using split-averaged $S_a = \tfrac{1}{2}(S_a^A + S_a^B)$ and $S_e = \tfrac{1}{2}(S_e^A + S_e^B)$, the shrunk alignment is

$$\hat{A}_F^{\text{EB}}(i,j) = \hat{A}_{\text{proxy}}(i,j)\, \hat{\rho}_{ij}^{\text{EB}}, \qquad \hat{A}_{\text{proxy}}(i,j) = \frac{S_a(i,j) S_e(i,j)}{\sqrt{S_a(i,i) S_a(j,j) S_e(i,i) S_e(j,j)}}.$$

All diagnostics are computed after clamping; when clamping is active for some pairs, it can inflate $\widehat{\mathrm{SE}}_{ij}^2$ and thus increase shrinkage, which is conservative.

**Complexity.** Per-task: $O(n_i \cdot m \cdot (d + K))$ for sketch computation. Pairwise: $O(m)$ per dot product. Total: $O(\sum_i n_i \cdot m(d + K) + T^2 m)$.

**Memory and reproducibility.** Storing Rademacher matrices requires $O(m(d + K))$ memory, plus $O(Tm)$ for task embeddings. For large $d$ and $m = 4096$, this can be nontrivial. An alternative is stateless on-the-fly generation using seeded hashing (4-wise independent), which trades compute for memory. Seed sensitivity is reported for multi-run validations; Table 11 summarizes seeds and averaging across experiment blocks.

*Table 6.* Wall-clock timing: exact vs. sketched Fisher alignment computation at $m = 4096$ on synthetic tasks (5 tasks; $d{=}256$, $K{=}100$; mean over 3 seeds).

| Samples/task | Exact (s) | Sketched (s) | Speedup |
|---:|---:|---:|---:|
| 50 | 0.16 | 0.11 | $1.5\times$ |
| 100 | 0.62 | 0.17 | $3.7\times$ |
| 200 | 2.50 | 0.30 | $8\times$ |
| 500 | 15.6 | 0.68 | $23\times$ |
| 1000 | 62.3 | 1.31 | $48\times$ |
| 2000 | 247.4 | 2.78 | $89\times$ |

### B.6. Validation

**Synthetic data.** On synthetic data with varying activation-error coupling, the sketched estimator achieves Spearman corr. $> 0.99$ with exact $S_{ae}$ computation at $m = 4096$. The diagonal $\chi_{ii}$ values (within-task coupling) achieve Spearman corr. $> 0.94$ with exact values. Note that off-diagonal $\chi_{ij}$ values are typically close to 1.0 for settings where $a$ and $e$ are approximately independent within each task (so $S_{ae}(i, j) \approx S_a(i, j)S_e(i, j)$ for any pair), which is expected and correctly captured by the sketched estimator.

**Real models.** On ViT-B/16 with 20 fine-tuned checkpoints across 4 datasets (CIFAR-100, Flowers102, DTD, Oxford-Pets), we validate all key quantities by comparing sketched estimates to exact computation. In this moderate-$K$ setting, we compute $S_a$ and $S_e$ exactly from their quadratic feature maps; for large $K$ we sketch any quantity that cannot be materialized. For same-dataset pairs (where Fisher alignment is non-zero), at $m = 4096$:

- $\hat{S}_{ae}$: Spearman corr. $0.97 \pm 0.01$

- $\hat{\chi}$: Spearman corr. $0.94 \pm 0.01$

- $\hat{\rho}$: Spearman corr. $0.95 \pm 0.01$

- $\hat{A}_F$: Spearman corr. $0.79 \pm 0.06$ (lower due to ratio normalization)

Rankings for $\hat{S}_{ae}$, $\hat{\chi}$, and $\hat{\rho}$ remain $> 0.90$ at $m = 2048$, while $\hat{A}_F$ is 0.76. Cross-dataset pairs (disjoint label spaces under the shared-output embedding) have Fisher $\approx 0$ by construction and are excluded from correlation analysis. Figure 1 shows how correlation improves with sketch dimension.

**Wall-clock timing.** To demonstrate the practical speedup from linear-time estimation, we benchmark exact $O(n^2)$ computation against sketched $O(n)$ computation. Setup: 5 synthetic tasks, $d = 256$ (activation dimension), $K = 100$ (error dimension), $m = 4096$ (sketch dimension), averaged over 3 seeds. Table 6 shows runtime scaling: The sketched estimator achieves $89\times$ speedup at $n = 2000$ samples/task with $m = 4096$, with runtime scaling linearly in $n$ (vs. quadratically for exact computation). Smaller sketch dimensions yield larger speedups (e.g., $437\times$ at $m = 1024$) with slightly reduced accuracy. This makes head Fisher alignment practical at scale. Full-network sketching via structured projections is also linear in hidden sizes and sequence length (independent of parameter count; Appendix B.7); our LLM experiments instead use parameter-subsampled gradients across all layers for computational tractability, while head-level validation uses the kernel identity.

### B.7. Full-Network FisherSketch via Structured Rademacher Projections

The head-layer sketching above extends naturally to full-network Fisher alignment by exploiting the per-layer Kronecker structure of gradients.

**The computational challenge.** For a full network with $L$ layers and total parameters $p = \sum_{\ell=1}^{L} d_{\ell-1}d_\ell$, naively computing $r^\top g$ for a random Rademacher vector $r \in \{\pm1\}^p$ requires $O(p)$ operations per sample—intractable for modern LLMs with billions of parameters.

**Structured projection.** Because the full gradient $g = [a_0 \otimes \delta_1; \dots; a_{L-1} \otimes \delta_L]$ is a concatenation of Kronecker products, we can factor the projection. For each layer $\ell$, generate independent Rademacher vectors:

$$r_\ell^{(a)} \in \{\pm 1\}^{d_{\ell-1}}, \quad r_\ell^{(\delta)} \in \{\pm 1\}^{d_\ell}.$$

The block of $r$ corresponding to layer $\ell$ is implicitly $r_\ell = r_\ell^{(a)} \otimes r_\ell^{(\delta)}$, and the projection becomes:

$$r^\top g(x) = \sum_{\ell=1}^L (r_\ell^{(a)\top} a_{\ell-1}(x)) \cdot (r_\ell^{(\delta)\top} \delta_\ell(x)). \tag{11}$$

This requires only $O(\sum_\ell (d_{\ell-1} + d_\ell))$ operations—linear in hidden sizes, not in parameter count (with a linear factor in positions for shared-parameter layers). For a shared-parameter affine layer used at positions $t \in \{1, \dots, S_\ell\}$, the per-layer term becomes

$$(r_\ell^{(a)} \otimes r_\ell^{(\delta)})^\top g_\ell(x) = \sum_{t=1}^{S_\ell} (r_\ell^{(a)\top} a_{\ell-1,t}(x)) (r_\ell^{(\delta)\top} \delta_{\ell,t}(x)),$$

so the structured projection remains computable in time linear in hidden sizes and the number of positions.

**Algorithm: Full FisherSketch.** For task $k$ with samples $\{x_t\}_{t=1}^{n_k}$:

1. Sample $m$ pairs of structured Rademacher vectors $(r^{(j)}, s^{(j)})$ with per-layer factorization.

2. For each sample $x_t$: run forward pass to get $\{a_{\ell-1}(x_t)\}_{\ell=1}^L$; run backward pass to get $\{\delta_\ell(x_t)\}_{\ell=1}^L$.

3. Compute sketch features $\psi^{(j)}(x_t) = (r^{(j)\top} g(x_t))(s^{(j)\top} g(x_t))$ using (11).

4. Average to get mean embedding $\mu_k = \frac{1}{n_k} \sum_t \psi(x_t) \in \mathbb{R}^m$.

Full-network Fisher alignment is then $A_F^{\text{full}}(i,j) \approx \cos(\mu_i, \mu_j)$.

**Complexity.** Per sample: $O(L \cdot d_{\max} \cdot m)$ for $m$ sketch dimensions. Per task: $O(n_k \cdot L \cdot d_{\max} \cdot m)$. Cross-task alignment: $O(T^2 \cdot m)$. This is independent of parameter count $p$ (linear in hidden sizes and positions), enabling full-network Fisher at 70B scale.

**Implementation note.** We present this structured-projection estimator for completeness; its per-sample cost is linear in layer widths. Our experiments use parameter-subsampled gradients across all layers for computational tractability; the structured-projection estimator provides a full-network sketching option.

**Relation to exact theorem.** Full FisherSketch approximates the exact full-network Fisher alignment (Theorem O.18), just as head FisherSketch approximates head Fisher. The structured Rademacher projection preserves the polynomial kernel structure: $\mathbb{E}[\psi(x)\psi(x')] = (g(x)^\top g(x'))^2 = (u^\top v)^2$, where $u, v$ are the depth-profile vectors from Definition O.15.

### B.8. Fast FisherSketch via SRHT (Implementation Note)

For vocab-scale applications ($K > 10^5$), even the factored Random Maclaurin projection can become compute-limited: computing $m$ dot products of the form $r^\top e$ with $r \in \{\pm 1\}^K$ requires $O(mK)$ operations per sample. The Subsampled Randomized Hadamard Transform (SRHT) reduces this $K$-side cost to $O(K \log K + m)$; if activations use dense projections, the total per-sample cost is $O(md + K \log K + m)$.

**Key construction.** Walsh–Hadamard transforms are defined for lengths $N = 2^p$. Let $N := 2^{\lceil \log_2 K \rceil}$ and $H_N \in \{\pm 1\}^{N \times N}$ be the (unnormalized) Walsh–Hadamard matrix. For $e \in \mathbb{R}^K$, pad with zeros to $\tilde{e} \in \mathbb{R}^N$. Let $D = \text{diag}(s)$ with $s \in \{\pm 1\}^N$ i.i.d. Rademacher. For a fixed row index $t \in [N]$, define $r := H_{t,:} \cdot D$. Then $r$ has i.i.d. Rademacher entries (each coordinate is a fixed $\pm 1$ from $H_N$ times an independent Rademacher sign from $D$); in particular, the first $K$ coordinates are i.i.d. Rademacher and $r^\top \tilde{e} = \sum_{k=1}^K r_k e_k$.

**Fast projection.** Given indices $t_1, \ldots, t_m \in [N]$, all $m$ projections can be computed as:

1. Pad and sign flip: $\tilde{e}' \leftarrow D \odot \tilde{e}$            $O(N)$

2. Compute Hadamard transform: $u \leftarrow H_N \tilde{e}'$            $O(N \log N)$

3. Gather entries: $\{u_{t_j}\}_{j=1}^m$            $O(m)$

Total: $O(N \log N + m) = O(K \log K + m)$ (since $N < 2K$), instead of $O(mK)$ on the error side.

**Memory reduction.** Dense Rademacher matrices require $O(mK)$ entries (about $\sim 2\,$GB if stored as `float32`, or $\sim 0.5\,$GB if stored as `int8` signs). SRHT requires only $O(N)$ float32 signs for diagonal matrices $(D, D')$ plus $O(m)$ integers for row indices; since $N < 2K$ this is still $O(K)$. At $K = 128{,}256$ (so $N = 131{,}072$) and $m = 4096$, this reduces memory to $\sim 1\,$MB.

**Unbiasedness.** The SRHT construction preserves the polynomial kernel property. For independent $(D, t)$ and $(D', t')$, and zero-padded $\tilde{x}, \tilde{y} \in \mathbb{R}^N$:

$$\mathbb{E}_{D,D',t,t'}\left[(r^\top \tilde{x})(r^\top \tilde{y})(r'^\top \tilde{x})(r'^\top \tilde{y})\right] = (x^\top y)^2,$$

exactly as with dense Rademacher. Thus the SRHT backend targets the same unnormalized product-kernel inner product in expectation; the reported Fisher alignment cosine is the corresponding normalized plug-in estimate.

**Dependence across coordinates.** SRHT uses a shared $(H_N, D)$ and row subsampling, so the $m$ coordinates are not independent. Unbiasedness still holds, but variance constants can differ from the i.i.d. Rademacher case; we use the $1/m$ scaling as a practical guide.

**When to use SRHT.** SRHT is beneficial when $mK$ is large. For typical LLM settings ($K \approx 128{,}000$, $m \approx 4096$), the theoretical complexity reduction is $O(mK)/O(K \log K) \approx 255\times$, though empirical speedups depend on hardware (memory bandwidth, cache effects). For smaller vocabularies or moderate $m$, dense Rademacher may be faster due to simpler cache patterns and better batching. Our implementation automatically selects based on dimension.

### B.9. Fast SRHT Error Projections in PyTorch

For vocabulary-scale errors $e \in \mathbb{R}^K$, FisherSketch uses an SRHT to avoid storing dense $m \times K$ Rademacher matrices. Let $\tilde{K}$ be the next power of two $\geq K$ and let $H_{\tilde{K}}$ denote the $\tilde{K} \times \tilde{K}$ Hadamard matrix. We sample sign vectors $D, D' \in \{\pm 1\}^{\tilde{K}}$ and index sets $I, I' \subseteq [\tilde{K}]$ with $|I| = |I'| = m$. Given a batch $E \in \mathbb{R}^{B \times K}$, we pad to $\tilde{K}$ and compute

$$U = H_{\tilde{K}}\left((E_{\text{pad}}) \odot D\right), \qquad U' = H_{\tilde{K}}\left((E_{\text{pad}}) \odot D'\right),$$

then return $U_{[:,I]}$ and $U'_{[:,I']}$. This gives $O(\tilde{K} \log \tilde{K} + m)$ cost per batch for the error-side projections.

**Batched FWHT (vectorized butterfly).** PyTorch does not provide a native FWHT; we implement the Hadamard transform via an iterative butterfly that uses only reshape/slice/concatenate operations. This avoids Python recursion and runs efficiently on GPU:

### B.10. Low-Precision Sign Storage for Fixed Projection Memory

Dense Rademacher projections for activations (and errors, when dense) are stored as `int8` signs on GPU and cast on-the-fly to the compute dtype during matmul. This reduces persistent projection storage by $4\times$ compared to float32 while leaving the estimator unchanged.

### B.11. Vocabulary-Scale Validation

We validate FisherSketch at vocabulary scale ($K = 128{,}256$) using Llama-3.1-8B on 100 domains constructed from eight HF datasets (Table 7). At this scale, storing the error second moment $\Gamma_e$ per task requires $K^2 \times 4$ bytes $\approx 61\,$GiB, so 100 tasks would require $\sim 6.0\,$TiB. FisherSketch stores activation projections as dense `int8` sign matrices: $2dm$ entries (32 MiB

---

**Algorithm 2** Batched FWHT used by SRHT (`view+cat` butterfly)

---

**Require:** $Y \in \mathbb{R}^{B \times \tilde{K}}$ with $\tilde{K}$ a power of two
1: $h \leftarrow 1$
2: **while** $h < \tilde{K}$ **do**
3:    $Y \leftarrow \mathrm{view}(Y, B, -1, 2, h)$
4:    $A \leftarrow Y[:,:,0,:], \quad B \leftarrow Y[:,:,1,:]$
5:    $Y \leftarrow \mathrm{cat}(A{+}B, A{-}B, \mathrm{dim}{=}2)$
6:    $Y \leftarrow \mathrm{view}(Y, B, \tilde{K})$
7:    $h \leftarrow 2h$
8: **end while**
9: **return** $Y$

---

*Table 7.* **Vocabulary-scale datasets** (HF *load_dataset* arguments). Each family contributes 12–13 domains via deterministic subsampling.

| Family | Dataset ID | Config/version | Split | Text field |
|---|---|---|---|---|
| Wikipedia | wikitext | wikitext-103-raw-v1 | train | text |
| QA | squad | – | train | context |
| Reviews | imdb | – | train | text |
| NLI | multi_nli | – | train | premise |
| Paraphrase | glue | mrpc | train | sentence1 |
| News | cnn_dailymail | 3.0.0 | train | article |
| Translation | wmt14 | de-en | train | translation.en |
| Commonsense | hellaswag | – | train | ctx |

for $d{=}m{=}4096$). In our PyTorch implementation these signs are cast to the compute dtype (`bf16/fp16/fp32`) on-the-fly for GEMM; this is an exact re-encoding of the same Rademacher $\{\pm 1\}$ projections and yields indistinguishable transfer metrics versus storing the same signs in `float32`. For error projections we use SRHT, which stores only two sign vectors $(D, D')$ and row indices (approximately 1 MB at $K{=}128{,}256$, $m{=}4096$). During streaming, we maintain six float64 accumulators per task for split-sample estimation (192 KB per task), but the stored signature is just the float32 $\hat{\mu}_{ae}$ vector (16 KB per task; 48 KB if $\hat{\mu}_a, \hat{\mu}_e$ are also stored). Fixed projection memory is about 33 MiB; for retrieval, the FAISS index stores another copy of $\hat{\mu}_{ae}$. SRHT on activations can reduce fixed memory further. This is a $\sim 4.0 \times 10^6$ reduction in per-task storage (61 GiB $\rightarrow$ 16 KB); the working-memory reduction is $\sim 3.3 \times 10^5$ if counting the streaming state (192 KB per task). Accounting for the fixed projections, the total reduction for 100 tasks is $\sim 1.8 \times 10^5$; with dense error projections, storing the two $m \times K$ sign matrices would require $\approx 1.0$ GiB fixed if stored as `int8` (or $\approx 3.9$ GiB if stored as `float32`), and the total reduction would be $\sim 6 \times 10^3$ (or $\sim 1.6 \times 10^3$ for `float32`).

**Datasets and domain construction.** We draw from eight HF datasets, using the train split and the specific text fields shown in Table 7. Each domain is a deterministic subsample; domain names follow the pattern *family* and *family_i* (12–13 slices per family), with a fixed seed per domain. We record dataset provenance and sample indices in *domain_metadata.json*; no fallback or synthetic data were used in the reported runs. We do not redistribute raw text, and dataset licenses follow the respective HF dataset cards.

**Sampling and splits.** FisherSketch uses 500 samples per domain ($m{=}4096$), keeping texts with $\geq 100$ characters and truncating to *max_seq_length*$\times 4$. LoRA validation draws 300 texts per domain (200 train, 100 eval) using a separate data seed (default: *seed+1*); eval samples are disjoint from train by construction. Early stopping uses the source-domain eval split.

**Disjointness and auditability.** Validation explicitly excludes the FisherSketch sample indices for each domain (recorded in *domain_metadata.json*), preventing overlap between sketching and LoRA train/eval data.

**Candidate sampling and pair counts.** For each target, we sample 24 candidate sources uniformly from the other 99 domains (uniform protocol), giving $100 \times 24 = 2400$ off-diagonal pairs. We also sample a stratified set (3 sources per family across 8 families, filled to 24 if needed); we evaluate the union of uniform and stratified pairs. For the reported run this yields 4,188 unique off-diagonal pairs, plus 100 diagonal pairs for self-PPL. Top-1 and regret are computed within the

*Table 8.* Vocabulary-scale validation summary. Baseline and FisherSketch columns report *per-task* stored signature size (float32, $\hat{\mu}_{ae}$ only); FisherSketch also requires $\sim$33 MiB fixed projections with SRHT for $d=m=4096$ and maintains a 192 KB per-task streaming state during extraction. Top-1 is mean $\pm$ std over seeds 43–45; improvement uses mean top-1 / random.

| Model | $K$ | Domains | Baseline | FisherSketch | Reduction | Top-1 | Improv. |
|---|---|---|---|---|---|---|---|
| Llama-3.1-8B | 128,256 | 100 | 61 GiB | 16 KB | $4.0\times10^6$ | $45.7\% \pm 5.0$ | $10.9\times$ |

uniform 24-candidate sets; correlation analyses use the union of evaluated pairs. Candidate sets and metadata are saved in *candidate_sources_\*.json* and *pair_sampling_metadata.json*.

**Transfer validation.** To verify that FisherSketch at $K>10^5$ produces meaningful alignment, we validate against actual LoRA transfer performance: for each source-target pair, we fine-tune a LoRA adapter on the source domain and evaluate perplexity on the target. The normalized transfer score measures what fraction of possible improvement (relative to self-training) the source provides in loss space (log PPL). On 4,188 off-diagonal pairs (100 domains), FisherSketch with SRHT error projections achieves $45.7\% \pm 5.0$ top-1 source selection ($\approx$10.9$\times$ random) with max regret $0.119 \pm 0.013$ (mean $\pm$ std over seeds 43–45), confirming that FisherSketch predicts cross-domain transfer at vocabulary scale where baseline methods cannot run.

**Loss-space regret.** For completeness, Table 9 reports per-target regret in log-PPL units for the uniform 24-candidate protocol. These values are unnormalized and not comparable across targets; normalized-transfer regret in Table 1 remains the primary metric.

*Table 9.* Loss-space (log PPL) regret summary (predicted minus oracle) under the uniform 24-candidate protocol. Values are mean $\pm$ std over seeds 43–45 for the mean/median/95th percentile of per-target regret.

| Method | Mean | Median | 95th |
|---|---|---|---|
| FisherSketch (SRHT) | $0.0025 \pm 0.0004$ | $0.0005 \pm 0.0005$ | $0.0087 \pm 0.0007$ |
| $S_{\mathrm{cov}}(M_a)$ | $0.0023 \pm 0.0002$ | $0.0003 \pm 0.0002$ | $0.0086 \pm 0.0003$ |
| $S_{\mathrm{cov}}(\Gamma_e)$ | $0.0030 \pm 0.0003$ | $0.0009 \pm 0.0001$ | $0.0118 \pm 0.0007$ |

## C. Shared-Parameter Affine Layers: Convolutions and Token-Wise Transformer Projections

Many modern architectures reuse the same affine parameters across multiple *positions* within an example: convolution kernels are shared across spatial locations, and transformer projection matrices (e.g., $W_Q, W_K, W_V, W_O$) and MLP matrices are shared across tokens. In this regime, the per-example gradient is a *sum of outer products* over positions rather than a single outer product. This section provides (i) an exact kernel identity for the shared-parameter gradient and (ii) an efficient factored sketch that generalizes FisherSketch. Table 10 gives additional exact-Fisher validation for shared-parameter layers. These results validate the shared-layer FisherSketch estimator itself; they do not assume the separable activation/error proxy that is specific to no-sharing heads. As an implementation check, the shared-layer kernel identity matched materialized per-example gradients to machine precision in the Q1 validation (mean absolute discrepancies at most $2 \times 10^{-16}$ and maxima at most $6 \times 10^{-16}$).

*Table 10.* Shared-parameter FisherSketch validation. Spearman values compare sketched shared-layer Fisher alignment against exact materialized-gradient Fisher alignment over task pairs; mean relative error is for $m = 4096$.

| Layer | Tokens | $m = 256$ | $m = 1024$ | $m = 4096$ / rel. err. |
|---|---|---|---|---|
| ResNet-18 conv | 196 | 0.893 | 0.976 | 0.998 / 6.1% |
| ResNet-18 conv | 784 | 0.970 | 0.985 | 0.997 / 3.9% |
| ViT-B/16 MLP | 197 | 0.971 | 0.991 | 0.997 / 4.8% |
| ViT-B/16 QKV | 197 | 0.964 | 0.992 | 0.994 / 0.5% |

**Setup.** Fix a task distribution $\mathcal{D}_k$ over examples $(x, y)$ and fix model parameters $\theta$. Consider a shared affine parameter matrix $W \in \mathbb{R}^{d_{\mathrm{out}} \times d_{\mathrm{in}}}$ used at $S$ positions (possibly varying by example). For an example $(x, y)$, let $a_t(x) \in \mathbb{R}^{d_{\mathrm{in}}}$ denote

the input vector at position $t \in \{1, \ldots, S\}$. Define

$$z_t(x) = W\, a_t(x) \in \mathbb{R}^{d_{\text{out}}}.$$

Let $\delta_t(x) := \nabla_{z_t} \ell_\theta(x, y) \in \mathbb{R}^{d_{\text{out}}}$ be the backpropagated gradient with respect to the pre-activation $z_t$. Stack these as column matrices

$$A(x) := [a_1(x), \ldots, a_S(x)] \in \mathbb{R}^{d_{\text{in}} \times S}, \qquad \Delta(x) := [\delta_1(x), \ldots, \delta_S(x)] \in \mathbb{R}^{d_{\text{out}} \times S}.$$

We focus on the per-example gradient with respect to $W$ and its induced Fisher block. If the shared affine map includes a bias term $b$, we can augment each $a_t$ with a constant 1 and treat $b$ as an extra column of $W$. All identities below then apply unchanged.

**Lemma C.1** (Per-example gradient for a shared affine parameter). *The per-example gradient of $\ell_\theta(x, y)$ with respect to $W$ is*

$$\nabla_W \ell_\theta(x, y) = \sum_{t=1}^{S} \delta_t(x)\, a_t(x)^\top = \Delta(x)\, A(x)^\top. \tag{12}$$

*In vectorized form (where* vec *stacks columns),*

$$g_W(x, y) := \text{vec}(\nabla_W \ell_\theta(x, y)) = \sum_{t=1}^{S} a_t(x) \otimes \delta_t(x). \tag{13}$$

*Proof.* For each position $t$, $z_t = W a_t$ depends linearly on $W$. For any entry $(i, j)$ of $W$, $\partial z_{t,i}/\partial W_{ij} = a_{t,j}$, whereas $\partial z_{t,i'}/\partial W_{ij} = 0$ for $i' \neq i$. Thus, by the chain rule, only the $i' = i$ term contributes, and

$$\frac{\partial \ell}{\partial W_{ij}} = \sum_{t=1}^{S} \sum_{i'=1}^{d_{\text{out}}} \frac{\partial \ell}{\partial z_{t,i'}} \frac{\partial z_{t,i'}}{\partial W_{ij}} = \sum_{t=1}^{S} \frac{\partial \ell}{\partial z_{t,i}} a_{t,j} = \sum_{t=1}^{S} \delta_{t,i}\, a_{t,j}.$$

This is exactly the $(i, j)$ entry of $\sum_t \delta_t a_t^\top = \Delta A^\top$. Vectorization uses the identity $\text{vec}(uv^\top) = v \otimes u$, so $\text{vec}(\delta_t a_t^\top) = a_t \otimes \delta_t$. Linearity then yields the sum. $\square$

**Lemma C.2** (Token/position Gram identity for gradient inner products). *Let $g_W$ and $g'_W$ be the vectorized shared-parameter gradients for two examples, corresponding to $(A, \Delta)$ and $(A', \Delta')$ (possibly with different numbers of positions $S$ and $S'$). Then*

$$g_W^\top g'_W = \sum_{t=1}^{S} \sum_{s=1}^{S'} (a_t^\top a'_s)\, (\delta_t^\top \delta'_s) \tag{14}$$

$$= \langle A^\top A',\ \Delta^\top \Delta' \rangle_F, \tag{15}$$

*where $\langle X, Y \rangle_F := \text{tr}(X^\top Y)$ is the Frobenius inner product.*

*Proof.* By Lemma C.1, $g_W = \sum_t a_t \otimes \delta_t$ and $g'_W = \sum_s a'_s \otimes \delta'_s$. Using bilinearity and $(u \otimes v)^\top (u' \otimes v') = (u^\top u')(v^\top v')$, we obtain

$$g_W^\top g'_W = \sum_{t,s} (a_t^\top a'_s)(\delta_t^\top \delta'_s),$$

which is (14). Now note $(A^\top A')_{t,s} = a_t^\top a'_s$ and $(\Delta^\top \Delta')_{t,s} = \delta_t^\top \delta'_s$. Therefore,

$$\langle A^\top A', \Delta^\top \Delta' \rangle_F = \sum_{t,s} (A^\top A')_{t,s} (\Delta^\top \Delta')_{t,s} = \sum_{t,s} (a_t^\top a'_s)(\delta_t^\top \delta'_s),$$

which is (15). $\square$

**A shared-layer Fisher kernel.** Define the degree-2 polynomial kernel induced by the shared-parameter gradient:

$$k_{\text{share}}\big((A, \Delta), (A', \Delta')\big) := \big(g_W^\top g_W'\big)^2 = \langle A^\top A', \Delta^\top \Delta' \rangle_F^2. \tag{16}$$

This kernel is positive semidefinite because it is the homogeneous polynomial kernel of degree 2 applied to the feature vector $g_W$.

**Theorem C.3** (RKHS mean embedding view (shared parameters)). *Let $F_{k,W} := \mathbb{E}_{(x,y)\sim\mathcal{D}_k}[g_W(x, y)g_W(x, y)^\top]$ be the empirical Fisher block associated with the shared parameter $W$ under task $k$. Then for any tasks $i, j$,*

$$\langle F_{i,W}, F_{j,W} \rangle_F = \mathbb{E}_{(A,\Delta)\sim\mathcal{D}_i,\ (A',\Delta')\sim\mathcal{D}_j}\Big[ k_{\text{share}}\big((A, \Delta), (A', \Delta')\big) \Big]. \tag{17}$$

*Equivalently, letting $\mathcal{H}_{\text{share}}$ denote the RKHS of $k_{\text{share}}$ and $\mu_{k,W} \in \mathcal{H}_{\text{share}}$ its mean embedding,*

$$\langle F_{i,W}, F_{j,W} \rangle_F = \langle \mu_{i,W}, \mu_{j,W} \rangle_{\mathcal{H}_{\text{share}}}, \qquad A_{F,W}(i, j) = \frac{\langle \mu_{i,W}, \mu_{j,W} \rangle}{\|\mu_{i,W}\| \, \|\mu_{j,W}\|}. \tag{18}$$

*Proof.* By definition, $F_{k,W} = \mathbb{E}[g_W g_W^\top]$. Using bilinearity and trace-cyclicity, and taking independent draws from tasks $i$ and $j$,

$$\langle F_{i,W}, F_{j,W} \rangle_F = \text{tr}\Big( \mathbb{E}[g_W g_W^\top] \, \mathbb{E}[g_W' g_W'^\top] \Big) = \mathbb{E}\Big[ \text{tr}(g_W g_W^\top g_W' g_W'^\top) \Big] = \mathbb{E}\big[ (g_W^\top g_W')^2 \big],$$

where $g_W$ and $g_W'$ are independent draws from tasks $i$ and $j$ respectively. By Lemma C.2, $(g_W^\top g_W')^2 = k_{\text{share}}\big((A, \Delta)$ $, (A', \Delta')\big)$, proving the first identity. The RKHS statement follows from standard kernel mean embedding theory. $\square$

**SharedFisherSketch: an efficient factored sketch.** Although $g_W$ (as a vector in $\mathbb{R}^{d_{\text{in}} d_{\text{out}}}$) can be very large, it admits a structured projection via Lemma C.1.

**Proposition C.4** (Factored Random Maclaurin sketch for shared-parameter gradients). *Let $r_a, r_a' \in \{\pm 1\}^{d_{\text{in}}}$ and $r_\delta, r_\delta' \in \{\pm 1\}^{d_{\text{out}}}$ be independent Rademacher vectors. Define the scalar projection*

$$s(A, \Delta; r_a, r_\delta) := \sum_{t=1}^S (r_a^\top a_t)(r_\delta^\top \delta_t) = (r_a \otimes r_\delta)^\top g_W. \tag{19}$$

*Define one sketch coordinate*

$$\psi(A, \Delta) := s(A, \Delta; r_a, r_\delta) \ \cdot \ s(A, \Delta; r_a', r_\delta'). \tag{20}$$

*Then for independent examples $(A, \Delta)$ and $(A', \Delta')$,*

$$\mathbb{E}\big[ \psi(A, \Delta)\psi(A', \Delta') \big] = k_{\text{share}}\big((A, \Delta), (A', \Delta')\big) = (g_W^\top g_W')^2. \tag{21}$$

*With $m$ i.i.d. coordinates $\psi_1, \ldots, \psi_m$, define*

$$\Psi_m := \frac{1}{\sqrt{m}}[\psi_1, \ldots, \psi_m] \in \mathbb{R}^m.$$

*Then the kernel estimator $\hat{k}_m = \langle \Psi_m(A, \Delta), \Psi_m(A', \Delta') \rangle$ is unbiased for $k_{\text{share}}$.*

*Proof.* Fix $(A, \Delta)$ and $(A', \Delta')$ and abbreviate $g = g_W$, $g' = g_W'$. Let $r := r_a \otimes r_\delta$ and $r' := r_a' \otimes r_\delta'$. By (19), $\psi = (r^\top g)(r'^\top g)$ and similarly $\psi' = (r^\top g')(r'^\top g')$. Since $(r, r')$ are independent and each has $\mathbb{E}[rr^\top] = I$,

$$\mathbb{E}[\psi\psi'] = \mathbb{E}_r[(r^\top g)(r^\top g')] \, \mathbb{E}_{r'}[(r'^\top g)(r'^\top g')] = (g^\top g') \, (g^\top g') = (g^\top g')^2.$$

For $r = r_a \otimes r_\delta$ with independent Rademachers, $\mathbb{E}[r_{ij}r_{i'j'}] = \mathbb{E}[r_{a,i}r_{a,i'}]\mathbb{E}[r_{\delta,j}r_{\delta,j'}] = \mathbf{1}\{i = i'\}\mathbf{1}\{j = j'\}$. Hence $\mathbb{E}[rr^\top] = I$. Finally, $\hat{k}_m$ is the average of $m$ i.i.d. unbiased coordinates, hence unbiased. $\square$

**Sketch variance and estimator caveats.** Unbiasedness uses only second moments, while the variance depends on higher moments and can be large for polynomial-kernel sketches. See standard analyses of Random Maclaurin features and TensorSketch for variance/accuracy tradeoffs (Kar & Karnick, 2012; Pham & Pagh, 2013); the shared-parameter sketch inherits the same behavior.

**Computation and relation to the head.** Computing $s(A, \Delta; r_a, r_\delta)$ can be done without forming $g_W$: we first compute $\alpha := r_a^\top A \in \mathbb{R}^S$ and $\beta := r_\delta^\top \Delta \in \mathbb{R}^S$, then $s = \alpha^\top \beta$. This costs $O(S(d_{\text{in}} + d_{\text{out}}))$ per coordinate and is independent of the parameter count $d_{\text{in}} d_{\text{out}}$. When $S = 1$, so $A = a$ and $\Delta = \delta$, Proposition C.4 reduces exactly to the head-layer factored sketch (Proposition 5.1).

## D. Preliminaries (Detailed)

This appendix provides full definitions and derivations for Fisher information and linear CKA, expanding the condensed treatment in Section 2.

### D.1. Fisher Information and Fisher Alignment

The Fisher information matrix is central to optimization and natural-gradient methods and is widely used in continual learning (Amari, 1998; Martens, 2020; Pascanu & Bengio, 2014; Kirkpatrick et al., 2017). For a given task $k$, define the *per-example gradient* at $\theta$ as

$$g_\theta^{(k)}(x, y) := \nabla_\theta \ell_\theta^{(k)}(x, y) \in \mathbb{R}^p.$$

We work with the **data Fisher**—also called the empirical Fisher in population form—defined as the *uncentered second moment* of per-example gradients under $\mathcal{D}_k$:

$$\mathcal{F}_k(\theta) := \mathbb{E}_{(x,y) \sim \mathcal{D}_k}\big[g_\theta^{(k)}(x, y)\, g_\theta^{(k)}(x, y)^\top\big] \in \mathbb{R}^{p \times p}. \tag{22}$$

This is the gradient outer product Fisher, standard in continual learning (Kirkpatrick et al., 2017); see Remark D.1 for Fisher variants. We assume $\mathcal{F}_k(\theta)$ exists and is positive semidefinite, and we additionally assume positive definiteness where convenient.

In practice, we approximate $\mathcal{F}_k(\theta)$ from $n_k$ i.i.d. samples $(x_i^{(k)}, y_i^{(k)}) \sim \mathcal{D}_k$ using the sample average:

$$\hat{\mathcal{F}}_k := \frac{1}{n_k} \sum_{i=1}^{n_k} g_\theta^{(k)}(x_i^{(k)}, y_i^{(k)})\, g_\theta^{(k)}(x_i^{(k)}, y_i^{(k)})^\top. \tag{23}$$

*Remark* D.1 (Fisher variants). Several Fisher definitions appear in the literature:

(i) **True Fisher**: $\mathcal{F}_k^{\text{true}}(\theta) = \mathbb{E}_{x \sim \mathcal{D}_{k,x}} \mathbb{E}_{y \sim p_\theta(\cdot|x)}[\nabla_\theta \log p_\theta(y|x) \nabla_\theta \log p_\theta(y|x)^\top]$, where $\mathcal{D}_{k,x}$ is the input marginal of $\mathcal{D}_k$;

(ii) **Empirical Fisher (sample)**: $\hat{\mathcal{F}} = \frac{1}{n} \sum_i \nabla \ell_i \nabla \ell_i^\top$, averaging over data labels (Eq. (23));

(iii) **Head Fisher**: restricting to final-layer parameters only.

The empirical Fisher can be a poor approximation to the true Fisher in some settings (Kunstner et al., 2019). Our analysis uses the empirical Fisher on head parameters, which is standard in practice (Kirkpatrick et al., 2017). For an affine head (bias absorbed by augmenting $a$ with a constant), the gradient has Kronecker form $g^{\text{head}} = a \otimes e$. With column-major vectorization, $\text{vec}(ea^\top) = a \otimes e$, where $a$ is the penultimate activation and $e = \hat{p} - y$ with $\hat{p} := \text{softmax}(Wa)$ and $y$ a one-hot label.

To compare the Fisher geometry of two tasks, we use the cosine similarity defined in the main text (Eq. (1)). High Fisher alignment indicates that tasks $i$ and $j$ emphasize similar directions in parameter space. Low Fisher alignment indicates that the tasks are decoupled in the Fisher metric.

In practice, we estimate alignment $\hat{A}_{\text{F}}(i, j)$ by replacing $\mathcal{F}_i, \mathcal{F}_j$ in (1) with $\hat{\mathcal{F}}_i, \hat{\mathcal{F}}_j$.

**Lemma D.2** (Expected squared gradient agreement). *Let* $(x_i, y_i) \sim \mathcal{D}_i$ *and* $(x_j, y_j) \sim \mathcal{D}_j$ *be independent draws. Define* $g_i := g_\theta^{(i)}(x_i, y_i)$ *and* $g_j := g_\theta^{(j)}(x_j, y_j)$. *Assume* $\|\mathcal{F}_i\|_F, \|\mathcal{F}_j\|_F > 0$; *otherwise set* $A_{\text{F}}(i, j) := 0$. *Define the transfer proxy*

$$U(i, j) := \mathbb{E}\big[(g_i^\top g_j)^2\big].$$

*Then*

$$U(i, j) = \langle \mathcal{F}_i, \mathcal{F}_j \rangle_F.$$

*Normalizing yields*

$$\frac{U(i,j)}{\|\mathcal{F}_i\|_F \|\mathcal{F}_j\|_F} = A_{\mathrm{F}}(i,j).$$

*Proof.* Using independence and $\mathrm{tr}(AB) = \langle A, B \rangle_F$,

$$\mathbb{E}\big[(g_i^\top g_j)^2\big] = \mathbb{E}\big[\mathrm{tr}(g_i g_i^\top g_j g_j^\top)\big] = \mathrm{tr}\big(\mathbb{E}[g_i g_i^\top]\,\mathbb{E}[g_j g_j^\top]\big) = \langle \mathcal{F}_i, \mathcal{F}_j \rangle_F.$$

Dividing by $\|\mathcal{F}_i\|_F \|\mathcal{F}_j\|_F$ gives the normalized alignment identity. $\qquad\square$

**Gradient-space kernel alignment.** It is useful to view Fisher matrices as Gram matrices of gradient features. Assuming $n_k > 0$, define the gradient feature matrix for task $k$ as

$$\Phi_k := \begin{bmatrix} g_\theta^{(k)}(x_1^{(k)}, y_1^{(k)})^\top \\ \vdots \\ g_\theta^{(k)}(x_{n_k}^{(k)}, y_{n_k}^{(k)})^\top \end{bmatrix} \in \mathbb{R}^{n_k \times p}.$$

Then $\hat{\mathcal{F}}_k = \frac{1}{n_k} \Phi_k^\top \Phi_k$. Since the cosine normalization cancels constant factors, Fisher alignment equals a *kernel alignment* between gradient Gram matrices:

$$\hat{A}_{\mathrm{F}}(i,j) = \frac{\langle \Phi_i^\top \Phi_i, \Phi_j^\top \Phi_j \rangle_F}{\|\Phi_i^\top \Phi_i\|_F \|\Phi_j^\top \Phi_j\|_F}.$$

This is formally analogous to CKA, which we now detail for representation features.

### D.2. Linear CKA for Representations

Linear CKA (Kornblith et al., 2019) extends kernel alignment (Cristianini et al., 2001; Cortes et al., 2012) to compare neural network representations. Let $Z_i \in \mathbb{R}^{n \times d_i}$ and $Z_j \in \mathbb{R}^{n \times d_j}$ be representation matrices with $n$ samples (rows) and $d_i, d_j$ features (columns). We assume each column is sample-centered (zero empirical mean). Write $K_i := Z_i Z_i^\top$ and $K_j := Z_j Z_j^\top$ for the corresponding Gram matrices.

The *linear CKA* between $Z_i$ and $Z_j$ is defined as:

$$\mathrm{CKA}(Z_i, Z_j) := \frac{\|Z_i^\top Z_j\|_F^2}{\|Z_i^\top Z_i\|_F \|Z_j^\top Z_j\|_F} \tag{24}$$

We assume $\|Z_i^\top Z_i\|_F, \|Z_j^\top Z_j\|_F > 0$ so CKA is well-defined.

Linear CKA is invariant to orthogonal transformations and isotropic rescaling of the feature space. It is typically computed on *paired* data: the rows of $Z_i$ and $Z_j$ correspond to the same inputs $x_1, \ldots, x_n$ fed through two different networks, layers, or heads. SVCCA (Raghu et al., 2017a) (with PCA truncation) is invariant to orthogonal rotations and isotropic rescaling, but is not invariant to arbitrary invertible linear transforms in general because the PCA truncation can change the retained subspace. Plain CCA is invariant to invertible linear transforms on the full space. However, practical CCA/SVCCA pipelines (Morcos et al., 2018) can be sensitive to preprocessing choices. CKA is invariant to orthogonal rotations and isotropic rescaling without relying on a truncation step, making it popular for comparing representations across random initializations or architectures.

The key distinction we explore in the main text is between *auto-covariance alignment* (which governs Fisher geometry) and *cross-covariance alignment* (which CKA measures). This distinction is made precise in Section 3.

## E. Experimental Protocol Details

This appendix provides complete experimental details for the supplementary CIFAR-100 regime experiments and the synthetic overlap/similarity sweeps. The main paper FisherSketch experiments use ViT-B/16 for sketch accuracy and Llama-3.1-8B for vocab-scale transfer; full-network validation additionally includes Llama-3.1-70B (Appendix O.9).

*Table 11.* Random seed usage and averaging by experiment block.

| Experiment block | Seeds / runs | Averaging |
|---|---|---|
| Supplementary CIFAR-100 regime + synthetic overlap/similarity sweeps (App. E) | 1 (seed 42) | none |
| Sketch timing benchmark (Table 6) | 3 | mean over seeds |
| Vocab-scale transfer (Table 1; App. B.11) | 3 (seeds 43–45) | mean $\pm$ std; pairwise stability |
| ViT-B/16 head Fisher validation (Table 5) | 5 | correlations across seed pairs |
| ResNet-50 coupling correction (Table 13) | 3 | correlations across seed pairs |
| ViT-B/16 full-network validation (Table 16) | 5 | statistics over 50 pairs |
| LLM transfer (App. O.9) | deterministic splits | no averaging reported |

### E.1. Architectures and Training (Supplementary)

Supplementary CIFAR-100 regime experiments use torchvision ResNet-18 (He et al., 2016) (ImageNet-style stem: $7 \times 7$ conv, stride 2, maxpool; no CIFAR-specific stem) on CIFAR-100; synthetic overlap/similarity sweeps use small shared-backbone MLPs with randomly generated labels. Training uses SGD (momentum 0.9, weight decay $5 \times 10^{-4}$ applied to all parameters, including biases and BatchNorm parameters) with PyTorch cosine annealing (CosineAnnealingLR with $T_{\max} = $ epochs, $\eta_{\min} = 0$, stepped once per epoch; no warmup or restarts); source models are trained for 100 epochs with learning rate 0.2 and batch size 1024, and transfer runs use 50 epochs with the same optimizer settings. Data augmentation uses random crop (32, padding 4) and random horizontal flip; test-time uses only normalization (mean $(0.5071, 0.4867, 0.4408)$, std $(0.2675, 0.2565, 0.2761)$). For class-subset tasks, we filter CIFAR-100 to the specified 50 classes and remap labels to $\{0, \ldots, 49\}$ (25,000 train / 5,000 test; $\approx$25 steps/epoch at batch size 1024). Linear-probe heads on frozen backbones are trained with SGD (learning rate 0.1, weight decay $1 \times 10^{-4}$, cosine annealing) for 100 epochs. For the supplementary CIFAR-100 regime experiments and synthetic overlap/similarity sweeps in this appendix, we use a single fixed seed (42) and do not average over seeds unless explicitly noted; multi-seed validations are called out in Table 11.

**Random seeds and averaging.** Table 11 summarizes which experiments use multiple seeds and whether results are averaged.

### E.2. Task Pairs, Overlap, and Similarity Sweeps

**CIFAR-100 overlap tasks.** We define a fixed source task using classes 0–49 (50 classes). Target tasks are contiguous class ranges with controlled overlap: 0–49 (100% overlap, 50/50 shared), 13–62 (37/50 shared; 0.74 overlap), 25–74 (25/50 shared; 0.50 overlap), 38–87 (12/50 shared; 0.24 overlap), and 50–99 (0% overlap, disjoint). Overlap fraction is $|\mathcal{C}_{\mathrm{src}} \cap \mathcal{C}_{\mathrm{tgt}}|/50$. Class identities are preserved (no label permutation).

**Synthetic similarity sweeps.** For class-count/overlap sweeps in the synthetic experiments, label overlap is defined by class-set overlap: overlap $= 1$ uses identical labels, overlap $= 0$ uses disjoint class halves, and intermediate overlap samples labels from the overlapping class ranges. Network similarity is controlled by linearly interpolating two random networks across all layers: $W_l^{(2)} \leftarrow \alpha W_l^{(1)} + (1 - \alpha) W_l^{(2,\mathrm{rand})}$ with $\alpha \in \{0, 1/(N-1), \ldots, 1\}$ (default $N=100$). We do not renormalize weights after interpolation. Because we do not renormalize, interpolation scales weight norms by $\sqrt{\alpha^2 + (1-\alpha)^2}$, so $\alpha$ jointly controls similarity and scale.

### E.3. Metric Computation

- **CKA**: Computed on *centered* penultimate-layer activations using the full CIFAR-100 test set (10,000 images) shared across models; optional deterministic subsampling is used only for fast runs.

- $S_{\mathrm{cov}}(\Gamma_e)$: Error second-moment matrices $\Gamma_e = \frac{1}{n} \sum e_i e_i^\top$ computed on each task's test split, where $e = \mathrm{softmax}(Wa) - y$ (predictions minus one-hot labels, with masked softmax: logits set to $-\infty$ outside the task's label block; Assumption F.1). Errors are embedded into the shared 100-class space via that shared-output embedding, so coordinates outside the block are zero; linear-probe runs use the trained probe head.

- **Fisher alignment**: Exact head Fisher via the kernel identity (Appendix A) is used only for head-level validation. The CIFAR-100 regime experiments report FA-CKA $= \mathrm{CKA} \cdot S_{\mathrm{cov}}(\Gamma_e)$ and the separable proxy $S_{\mathrm{cov}}(\Sigma_z) \cdot S_{\mathrm{cov}}(\Gamma_e)$ instead, where $\Sigma_z$ is the centered covariance of penultimate activations (computed on the same test set as CKA).

### E.4. Reported Correlations

For CIFAR-100 regime experiments, correlations (Pearson corr.) are computed across the five target tasks within each regime (five overlap levels). For synthetic similarity sweeps, correlations are computed across the sweep points ($N$ trials). Because $N$ is small in the CIFAR-100 regime, correlations are reported for completeness and interpreted qualitatively.

# F. Shared Output Space: Details

**Assumption F.1** (Shared output space embedding). Tasks $i$ and $j$ predict over a common $K$-dimensional output space (e.g., CIFAR-100's 100 classes), where each task uses only a subset of coordinates (classes). Let $\mathcal{C}_k \subseteq \{1, \ldots, K\}$ be task $k$'s label subset. Let $P_k \in \{0, 1\}^{K \times |\mathcal{C}_k|}$ be the coordinate injection (it embeds subset coordinates into the shared $K$-space). Let $\tilde{y}_k \in \mathbb{R}^{|\mathcal{C}_k|}$ be the one-hot vector in the subset space (equivalently, $\tilde{y}_k = P_k^\top y$ for a $K$-way one-hot label $y$). Given subset logits $\tilde{z}_k \in \mathbb{R}^{|\mathcal{C}_k|}$, let $\tilde{p}_k = \mathrm{softmax}(\tilde{z}_k)$ and $\tilde{e}_k = \tilde{p}_k - \tilde{y}_k$. We embed the error as $e_k := P_k \tilde{e}_k \in \mathbb{R}^K$. Equivalently, this corresponds to a shared $K$-way head with logits set to $-\infty$ outside $\mathcal{C}_k$.

### F.1. Setup

To compare error covariances $\Gamma_{e,i}$ and $\Gamma_{e,j}$ across tasks with different label vocabularies, we use a common coordinate system.

We assume each task $k$ predicts over its label subset $\mathcal{C}_k \subseteq \{1, \ldots, K\}$, either via a subset head or a shared $K$-way head with masked softmax (logits set to $-\infty$ outside $\mathcal{C}_k$). If tasks use subset heads, we identify their head parameters with a shared $K$-way matrix by zero-padding rows outside $\mathcal{C}_k$. Predictions still use the masked softmax, so gradients and errors live in the common coordinate system. We assume label identities are aligned across tasks in this shared coordinate system. To compare error covariances across tasks, we embed these predictions into a common $K$-dimensional space via the injection $P_k$ in Assumption F.1. The embedded error vector $e_k \in \mathbb{R}^K$ is zero in coordinates outside $\mathcal{C}_k$. This setup is standard when tasks are subsets of a larger taxonomy (e.g., CIFAR-100 has $K{=}100$ classes and each task uses a subset). The disjoint-support implications below rely on masked softmax (logits set to $-\infty$ outside $\mathcal{C}_k$); with an unmasked $K$-way softmax, probability mass can appear on non-task classes, so the errors need not be sparse.

### F.2. Implications

Under this embedding:

$$\langle \Gamma_{e,i}, \Gamma_{e,j} \rangle_F = \langle \Gamma_{e,i}[\mathcal{C}_{ij}], \Gamma_{e,j}[\mathcal{C}_{ij}] \rangle_F, \qquad S_{\mathrm{cov}}(\Gamma_{e,i}, \Gamma_{e,j}) = \frac{\langle \Gamma_{e,i}[\mathcal{C}_{ij}], \Gamma_{e,j}[\mathcal{C}_{ij}] \rangle_F}{\|\Gamma_{e,i}\|_F \|\Gamma_{e,j}\|_F}, \tag{25}$$

where $\mathcal{C}_{ij} := \mathcal{C}_i \cap \mathcal{C}_j$ and $\Gamma_{e,k}[\mathcal{C}_{ij}]$ denotes the submatrix restricted to rows/columns in $\mathcal{C}_{ij}$. If $\|\Gamma_{e,i}\|_F$ or $\|\Gamma_{e,j}\|_F$ is zero, then $S_{\mathrm{cov}}$ is undefined (we $\varepsilon$-regularize in practice).

- **Disjoint tasks** (no shared classes, $\mathcal{C}_i \cap \mathcal{C}_j = \emptyset$): Error covariances have disjoint support, so $\langle \Gamma_{e,i}, \Gamma_{e,j} \rangle_F = 0$ and $S_{\mathrm{cov}}(\Gamma_{e,i}, \Gamma_{e,j}) = 0$ when the norms are nonzero.

- **Partially overlapping tasks**: The overlap-restricted inner product above governs $S_{\mathrm{cov}}(\Gamma_{e,i}, \Gamma_{e,j})$. In a stylized toy model that ignores the simplex constraint $\mathbf{1}^\top e = 0$, assume $\Gamma_{e,k} = \sigma_k^2 I_{\mathcal{C}_k}$,[3]

$$S_{\mathrm{cov}}(\Gamma_{e,i}, \Gamma_{e,j}) = \frac{|\mathcal{C}_{ij}|}{\sqrt{|\mathcal{C}_i| |\mathcal{C}_j|}}.$$

- **Identical tasks**: If $\Gamma_{e,i} = c \Gamma_{e,j}$ with $c > 0$, then $S_{\mathrm{cov}}(\Gamma_{e,i}, \Gamma_{e,j}) = 1$.

---

[3] For softmax errors, $\mathbf{1}^\top e = 0$ implies $\Gamma_{e,k} \mathbf{1} = 0$ (rank at most $|\mathcal{C}_k| - 1$), so a realizable isotropic model on the task support is $\Gamma_{e,k} \propto I_{\mathcal{C}_k} - \frac{1}{|\mathcal{C}_k|} \mathbf{1}\mathbf{1}^\top$.

**Implementation sanity checks.** It is helpful to verify that (i) $e_k$ is zero outside $\mathcal{C}_k$ under masked softmax (logits set to $-\infty$ outside $\mathcal{C}_k$), (ii) Eq. (25) holds numerically within floating-point tolerance, and (iii) $\Gamma_{e,k}\mathbf{1} \approx 0$ for softmax errors.

This framework applies when tasks are subsets of a larger taxonomy with a fixed label alignment (e.g., selecting animal vs. vehicle classes from CIFAR-100). The resulting $S_{\text{cov}}(\Gamma_{e,i}, \Gamma_{e,j})$ depends on the chosen embedding. For different label vocabularies, any cross-task embedding is a modeling choice. For tasks with fundamentally different output spaces (e.g., classification vs. regression), error-covariance comparison requires an explicit embedding, while representation-only metrics remain agnostic to gradient compatibility.

## G. CKA–Fisher Gap Decomposition (Head Layer)

This appendix provides a self-contained deviation bound that relates *empirical* linear CKA (on a paired probe set) to *population* head Fisher alignment. The bound is primarily diagnostic: it separates (i) finite-sample CKA estimation error, (ii) CKA's inherent cross-vs.-auto covariance mismatch, (iii) centering (mean) effects, (iv) error-geometry misalignment, and (v) activation–error dependence (i.e., Kronecker-factorization violation).

**Theorem G.1** (CKA–Fisher deviation decomposition, head layer). *Fix a probe input distribution $\mathcal{D}_0$ and two tasks indexed by $i$ and $j$. Let $a^{(i)}(x), a^{(j)}(x) \in \mathbb{R}^d$ denote the* head-input *(penultimate) activations of the two models for an input $x$. Let $Z_i, Z_j \in \mathbb{R}^{n \times d}$ be the corresponding* centered *probe matrices evaluated on* paired *probe inputs $x_t \sim \mathcal{D}_0$; row $t$ of $Z_i$ equals $a^{(i)}(x_t) - \hat{\mu}_i$ and row $t$ of $Z_j$ equals $a^{(j)}(x_t) - \hat{\mu}_j$.*

**CKA terms.** *Let $\text{CKA}(Z_i, Z_j)$ denote empirical linear CKA on this paired probe set. Define the population quantity and deviation*

$$S_{\text{cross}}(\Sigma_{ij}; \Sigma_i, \Sigma_j) := \frac{\|\Sigma_{ij}\|_F^2}{\|\Sigma_i\|_F \|\Sigma_j\|_F}, \qquad \gamma_n := \big| \text{CKA}(Z_i, Z_j) - S_{\text{cross}}(\Sigma_{ij}; \Sigma_i, \Sigma_j) \big|,$$

*where $\Sigma_i := \text{Cov}_{x \sim \mathcal{D}_0}[a^{(i)}(x)]$, $\Sigma_j := \text{Cov}_{x \sim \mathcal{D}_0}[a^{(j)}(x)]$, and $\Sigma_{ij} := \text{Cov}_{x \sim \mathcal{D}_0}[a^{(i)}(x), a^{(j)}(x)]$ (all covariances are centered).*

*Define the* cross-vs-auto mismatch

$$\kappa_{ij} := \big| S_{\text{cross}}(\Sigma_{ij}; \Sigma_i, \Sigma_j) - S_{\text{cov}}(\Sigma_i, \Sigma_j) \big|.$$

*We estimate $\kappa_{ij}$ using the Procrustes proxy $\hat{\kappa}$ in Appendix I.*

**Head Fisher terms.** *For each task $k \in \{i, j\}$, let $e^{(k)}(x, y) \in \mathbb{R}^K$ denote the backpropagated head error (i.e., the loss gradient with respect to logits). Define $g^{(k)}(x, y) := \text{vec}(\nabla_W \ell^{(k)}(x, y)) = a^{(k)}(x) \otimes e^{(k)}(x, y)$ (Lemma 4.1).[4] Define the head Fisher blocks*

$$\Sigma_{\psi,k} := \mathbb{E}_{(x,y) \sim \mathcal{D}_k}\big[ g^{(k)}(x, y) \, g^{(k)}(x, y)^\top \big], \qquad A_F^{\text{head}}(i, j) := S_{\text{cov}}(\Sigma_{\psi,i}, \Sigma_{\psi,j}).$$

*Also define activation and error second moments*

$$M_{a,k} := \mathbb{E}[a^{(k)} a^{(k)\top}], \quad \Gamma_{e,k} := \mathbb{E}[e^{(k)} e^{(k)\top}], \quad \mu_{a,k} := \mathbb{E}[a^{(k)}], \quad \Sigma_{a,k} := \text{Cov}(a^{(k)}),$$

*(all expectations under $\mathcal{D}_k$). Define the diagnostics*

$$\delta_k := \frac{\|\Sigma_{\psi,k} - M_{a,k} \otimes \Gamma_{e,k}\|_F}{\|\Sigma_{\psi,k}\|_F}, \qquad \varepsilon_e := 1 - S_{\text{cov}}(\Gamma_{e,i}, \Gamma_{e,j}), \qquad \varepsilon_{\mu,k} := \frac{\|\mu_{a,k}\|_2^2}{\|\Sigma_{a,k}\|_F}.$$

*If a denominator is zero, we follow the paper-wide $\varepsilon$-regularized convention; in practice we set the corresponding ratio/alignment to $0$.*

---

[4]The Kronecker order $a \otimes e$ matches our column-stacking vec convention (so $\text{vec}(uv^\top) = v \otimes u$). Using row-major vectorization swaps the Kronecker factors; all alignment scores are unchanged by this fixed permutation, but residuals such as $\delta_k$ must be computed with the matching order.

**Deviation bound under matched probe/task input marginals.** *Assume the probe input marginal matches each task's input marginal, $\mathcal{D}_0 = \mathcal{D}_i^x = \mathcal{D}_j^x$, so that $\Sigma_k = \Sigma_{a,k}$ for $k \in \{i, j\}$. Then*

$$\left| \text{CKA}(Z_i, Z_j) - A_F^{\text{head}}(i, j) \right| \leq \gamma_n + \kappa_{ij} + 2(\varepsilon_{\mu,i} + \varepsilon_{\mu,j}) + \varepsilon_e + 2(\delta_i + \delta_j). \tag{26}$$

*Proof.* Define the (separable) proxy

$$A_{\text{proxy}}(i, j) := S_{\text{cov}}(M_{a,i}, M_{a,j}) \cdot S_{\text{cov}}(\Gamma_{e,i}, \Gamma_{e,j}) = S_{\text{cov}}(M_{a,i} \otimes \Gamma_{e,i}, \, M_{a,j} \otimes \Gamma_{e,j}),$$

The last equality follows from $\langle A \otimes B, \, C \otimes D \rangle_F = \langle A, C \rangle_F \langle B, D \rangle_F$ and $\|A \otimes B\|_F = \|A\|_F \|B\|_F$.

By adding and subtracting intermediate terms and applying the triangle inequality,

$$\begin{aligned}
|\text{CKA} - A_F^{\text{head}}| \leq \; & |\text{CKA} - S_{\text{cross}}| + |S_{\text{cross}} - S_{\text{cov}}(\Sigma_i, \Sigma_j)| \\
& + |S_{\text{cov}}(\Sigma_i, \Sigma_j) - S_{\text{cov}}(M_{a,i}, M_{a,j})| + |S_{\text{cov}}(M_{a,i}, M_{a,j}) - A_{\text{proxy}}| \\
& + |A_{\text{proxy}} - A_F^{\text{head}}|.
\end{aligned}$$

The first two terms are $\gamma_n$ and $\kappa_{ij}$, respectively.

For the third term, since $\Sigma_i = \Sigma_{a,i}$ and $\Sigma_j = \Sigma_{a,j}$ under the matched-marginal assumption, Lemma M.1 yields

$$|S_{\text{cov}}(\Sigma_i, \Sigma_j) - S_{\text{cov}}(M_{a,i}, M_{a,j})| \leq 2(\varepsilon_{\mu,i} + \varepsilon_{\mu,j}).$$

For the fourth term, write $A_{\text{proxy}} = S_{\text{cov}}(M_{a,i}, M_{a,j}) \cdot S_{\text{cov}}(\Gamma_{e,i}, \Gamma_{e,j})$. Since $0 \leq S_{\text{cov}}(M_{a,i}, M_{a,j}) \leq 1$ for PSD matrices,

$$|S_{\text{cov}}(M_{a,i}, M_{a,j}) - A_{\text{proxy}}| = S_{\text{cov}}(M_{a,i}, M_{a,j})\big(1 - S_{\text{cov}}(\Gamma_{e,i}, \Gamma_{e,j})\big) \leq \varepsilon_e.$$

Finally, write $\Sigma_{\psi,k} = M_{a,k} \otimes \Gamma_{e,k} + R_k$ and note that $\delta_k = \|R_k\|_F / \|\Sigma_{\psi,k}\|_F$. Apply Corollary L.2 with $(\Sigma_i, \Sigma_j) = (\Sigma_{\psi,i}, \Sigma_{\psi,j})$ and $(\Sigma_i^{(0)}, \Sigma_j^{(0)}) = (M_{a,i} \otimes \Gamma_{e,i}, \, M_{a,j} \otimes \Gamma_{e,j})$ to obtain

$$|A_{\text{proxy}} - A_F^{\text{head}}(i, j)| = \left| S_{\text{cov}}(M_{a,i} \otimes \Gamma_{e,i}, \, M_{a,j} \otimes \Gamma_{e,j}) - S_{\text{cov}}(\Sigma_{\psi,i}, \Sigma_{\psi,j}) \right| \leq 2(\delta_i + \delta_j).$$

Combining the bounds yields (26). $\qquad \square$

**Probe mismatch.** If the probe input marginal $\mathcal{D}_0$ differs from the task input marginals $\mathcal{D}_i^x, \mathcal{D}_j^x$, the only change is in the mean-correction step. Insert the additional term $\left| S_{\text{cov}}(\Sigma_i, \Sigma_j) - S_{\text{cov}}(\Sigma_{a,i}, \Sigma_{a,j}) \right|$ to account for replacing probe covariances by task covariances. The error term $\varepsilon_e$ and coupling residuals $\delta_k$ are often the dominant obstructions in many-class settings, explaining why CKA can decouple from head Fisher when label/error structures differ.

**Estimating $\kappa_{ij}$.** When $d$ is moderate and we form explicit sample covariances, $\kappa_{ij}$ can be estimated directly as $\left| S_{\text{cross}}(\hat{\Sigma}_{ij}; \hat{\Sigma}_i, \hat{\Sigma}_j) - S_{\text{cov}}(\hat{\Sigma}_i, \hat{\Sigma}_j) \right|$. For high-dimensional representations where it is undesirable to interpret $S_{\text{cov}}(\hat{\Sigma}_i, \hat{\Sigma}_j)$ in a fixed coordinate basis, we additionally report the rotation-invariant Procrustes diagnostic $\hat{\kappa}$ from Appendix I as a *proxy* for cross-vs.-auto mismatch.

## H. Relationship to K-FAC and Prior Work

This appendix clarifies how our work relates to prior work on Kronecker-factored approximations to the Fisher information matrix, particularly K-FAC (Martens & Grosse, 2015; Grosse & Martens, 2016). Subsequent work has extended K-FAC to recurrent networks, eigenbasis approximations, and distributed training (Martens et al., 2018; George et al., 2018; Ba et al., 2017).

### H.1. What K-FAC Established

K-FAC (Kronecker-Factored Approximate Curvature; Martens & Grosse 2015) introduced a Kronecker-factored approximation of layer-wise Fisher information matrices for efficient second-order optimization. Key components include:

**Exact gradient structure.** K-FAC observes that for layer $l$ with weight matrix $W_l$, the per-example gradient has an exact Kronecker structure:

$$\text{vec}(\nabla_{W_l}\ell) = a_{l-1} \otimes \delta_l$$

Here $a_{l-1}$ is the layer input and $\delta_l$ is the backpropagated error. This identity follows directly from the chain rule.

**Independence approximation.** K-FAC's central approximation is that activations and errors are approximately independent, yielding the approximation:

$$\mathbb{E}[(a \otimes \delta)(a \otimes \delta)^\top] \approx \mathbb{E}[aa^\top] \otimes \mathbb{E}[\delta\delta^\top]$$

This factorization enables efficient Fisher inversion (two small matrix inversions instead of one giant one).

**Block-diagonal approximation.** K-FAC further approximates the full Fisher as block-diagonal across layers, ignoring cross-layer correlations.

## H.2. Our Novel Contributions Beyond K-FAC

Our work builds on K-FAC's structural insights but addresses a different question. K-FAC asks: "How can we efficiently approximate the natural gradient for optimization?" We ask: "Why do representation similarity metrics fail to predict gradient alignment?"

**1. Error covariance alignment as the critical factor.** While K-FAC uses the approximation $\Sigma_{\psi,k} \approx M_{a,k} \otimes \Gamma_{e,k}$ (Lemma B.2) for computational efficiency, we identify error-covariance alignment $S_{\text{cov}}(\Gamma_{e,i}, \Gamma_{e,j})$ as the missing factor that explains why CKA can decouple from Fisher alignment. K-FAC does not study cross-task alignment or representation metrics.

**2. Formal characterization of CKA's limitation.** We prove that CKA (and all representation-only metrics) structurally cannot, in general, access error covariance information (Theorem 3.2). This is a fundamental limitation independent of any approximation quality.

**3. Quantitative diagnostics for the CKA–Fisher gap.** Our analysis exposes measurable drivers of the gap, including the coupling correction $\rho$ (Theorem 4.2), the Kronecker residual $\delta$ (Remark 4.4), and the coupling-misalignment term $\kappa$ (Appendix I). K-FAC focuses on optimization convergence, not metric reliability.

**4. Label structure dependence.** We show that $S_{\text{cov}}(\Gamma_{e,i}, \Gamma_{e,j})$ drops to zero for disjoint $K$-class tasks under the shared output-space embedding (Appendix F), illustrating a failure mode for CKA when label structures differ. This is orthogonal to K-FAC's optimization focus.

**5. Practical estimator (FA-CKA).** FA-CKA provides a forward-pass-only estimator of head Fisher alignment without computing gradients or Fisher matrices. K-FAC requires gradient computation for natural gradient steps.

## H.3. Independence Assumption: Our Analysis vs. K-FAC

K-FAC relies on approximate independence between activations and backpropagated errors (denoted $\delta$ in H.1 and $e$ elsewhere in this paper). Under this assumption, $\Sigma_{\psi,k} = \mathbb{E}[(a \otimes e)(a \otimes e)^\top] \approx M_{a,k} \otimes \Gamma_{e,k}$ (Lemma B.2). Our analysis is *unconditional*: Theorem 4.2 is an exact identity for head Fisher alignment with no independence assumption.

**In K-FAC:** Violating independence reduces optimization efficiency but does not fundamentally break the method; even with imperfect factorization, K-FAC can remain a reasonable preconditioner.

**In our work:** Head Fisher alignment decomposes *exactly* as $A_{\text{F}}^{\text{head}}(i,j) = A_{\text{proxy}}(i,j) \cdot \rho_{ij}$ (Theorem 4.2). Here $A_{\text{proxy}} := S_{\text{cov}}(M_a) \cdot S_{\text{cov}}(\Gamma_e)$, and $\rho_{ij}$ is defined when the denominator is nonzero. Independence implies $\rho_{ij} = 1$ (Corollary 4.3), but $\rho \neq 1$ does not invalidate the theory—it is simply a measurable correction. The Kronecker residual $\delta$ yields a worst-case alignment bound (Appendix L); whether $\rho \approx 1$ is architecture-dependent (on ViT, $\rho$ varies from 0.35 to 1.11; see Appendix J).

Crucially, even with perfect independence ($\rho = 1$), CKA can fail when error covariances are misaligned (small $S_{\mathrm{cov}}(\Gamma_{e,i}, \Gamma_{e,j})$), e.g., for disjoint label subsets under the shared output-space embedding (Appendix F).

## H.4. Summary of Attribution

- **Gradient Kronecker structure** (Eq. (3)): Exact identity, noted by K-FAC (Martens & Grosse, 2015)

- **Fisher independence factorization** (Lemma B.2): K-FAC's foundational approximation

- **Error covariance alignment factor**: Our contribution

- **CKA limitation theorem** (Theorem 3.2): Our contribution

- **Quantitative diagnostics for the CKA–Fisher gap** ($\rho, \delta, \kappa$): Our contribution

- **FA-CKA estimator**: Our contribution

# I. Estimation and Diagnostic Experiments

This appendix provides implementation details for estimating the structural quantities in our theory and presents diagnostic experiments that validate the theory across different architectures.

## I.1. Estimating Structural Quantities

Each term in our decomposition can be estimated from finite samples.

**Representation misalignment ($\kappa$).** Define the population mismatch $\kappa_{ij} := \left| S_{\mathrm{cross}}(\Sigma_{ij}; \Sigma_i, \Sigma_j) - S_{\mathrm{cov}}(\Sigma_i, \Sigma_j) \right|$. **Direct estimator (when covariances are materialized).** When the representation dimension $d$ is shared across models and we form explicit sample covariances, we estimate $\kappa_{ij}$ directly via

$$\hat{\kappa}_{ij}^{\mathrm{dir}} := \left| \widehat{S}_{\mathrm{cross}}(\hat{\Sigma}_{ij}; \hat{\Sigma}_i, \hat{\Sigma}_j) - S_{\mathrm{cov}}(\hat{\Sigma}_i, \hat{\Sigma}_j) \right|, \qquad \widehat{S}_{\mathrm{cross}}(\hat{\Sigma}_{ij}; \hat{\Sigma}_i, \hat{\Sigma}_j) := \frac{\|\hat{\Sigma}_{ij}\|_F^2}{\|\hat{\Sigma}_i\|_F \|\hat{\Sigma}_j\|_F}.$$

This is the plug-in estimator that matches the definition of $\kappa_{ij}$.

**Rotation-invariant diagnostic (Procrustes proxy).** For high-dimensional representations, it can be undesirable to interpret $S_{\mathrm{cov}}(\hat{\Sigma}_i, \hat{\Sigma}_j)$ in a fixed coordinate basis. We therefore also report a rotation-invariant Procrustes diagnostic $\hat{\kappa}$, computed from the regularized whitened cross-covariance (below). This $\hat{\kappa}$ is a *proxy* for cross- and auto-covariance mismatch rather than an unbiased estimator of $\kappa_{ij}$. Let $Z_i \in \mathbb{R}^{n_i \times d}$ and $Z_j \in \mathbb{R}^{n_j \times d}$ be centered representations on probe inputs, with a shared feature dimension $d$. Unless stated otherwise, we use the same probe set as CKA; if sample counts differ, we form a paired subset of size $n_{ij}$ with matched inputs. Let $Z_i^{(ij)}, Z_j^{(ij)} \in \mathbb{R}^{n_{ij} \times d}$ denote those paired rows (in our experiments $n_i = n_j = n_{ij}$). Choose $r_{\mathrm{sub}}$ as the smallest $r$ such that the top-$r$ eigenvalues of both $\hat{\Sigma}_i$ and $\hat{\Sigma}_j$ explain at least 90% of the trace (equivalently, take the larger of the two 90% ranks). Define $O(r_{\mathrm{sub}}) := \{Q \in \mathbb{R}^{r_{\mathrm{sub}} \times r_{\mathrm{sub}}} : Q^\top Q = I\}$.

1. Compute sample covariances with regularization: $\hat{\Sigma}_i^{(\lambda)} = Z_i^\top Z_i / n_i + \lambda I$, $\hat{\Sigma}_j^{(\lambda)} = Z_j^\top Z_j / n_j + \lambda I$, $\hat{\Sigma}_{ij} = (Z_i^{(ij)})^\top Z_j^{(ij)} / n_{ij}$. We use $\lambda \approx 10^{-6}$ for numerical stability.

2. Compute eigendecompositions and extract the leading $r_{\mathrm{sub}}$-dimensional subspace bases: take $\hat{U}_i, \hat{U}_j \in \mathbb{R}^{d \times r_{\mathrm{sub}}}$ from the top-$r_{\mathrm{sub}}$ eigenvectors of $\hat{\Sigma}_i$ and $\hat{\Sigma}_j$.

3. Compute regularized whitened cross-covariance: $\hat{\tilde{C}} = (\hat{\Sigma}_i^{(\lambda)})^{-1/2} \hat{\Sigma}_{ij} (\hat{\Sigma}_j^{(\lambda)})^{-1/2}$.

4. Project onto the shared subspace: $\hat{C}_r = \hat{U}_i^\top \hat{\tilde{C}} \hat{U}_j$.

5. Solve the orthogonal Procrustes problem $Q^\star = \arg\min_{Q \in O(r_{\mathrm{sub}})} \|\hat{C}_r - Q\|_F$ (via SVD). Define

$$\hat{\kappa}(i, j) = \frac{\|\hat{C}_r - Q^\star\|_F}{\sqrt{r_{\mathrm{sub}}}}.$$

If the spectrum is simple, sign-aligning the eigenvectors yields $Q^\star = I$, so $\hat{\kappa}$ reduces to $\|\hat{C}_r - I\|_F / \sqrt{r_{\text{sub}}}$. Because $Q^\star$ solves the Procrustes problem, $\hat{\kappa}$ depends only on the singular values of $\hat{C}_r$ and is invariant to orthogonal basis rotations within the chosen subspaces. Small $\hat{\kappa}$ indicates aligned coupling in the whitened cross-covariance. With the normalization above, uncorrelated representations (whitened cross-covariance near zero in the shared subspace) yield $\hat{\kappa} \approx 1$.

**Kronecker independence violation ($\delta$).** For head gradients, we use the column-stacking vec convention so $g = a \otimes e$. With row-major vectorization, $g = e \otimes a$ and the Kronecker factors swap. We estimate the Kronecker approximation error per task $k$ by comparing the empirical gradient second moment $\hat{\Sigma}_{\psi,k} = \frac{1}{n_k} \sum_i g_i g_i^\top$ to the Kronecker product $\hat{M}_{a,k} \otimes \hat{\Gamma}_{e,k}$ (with $\hat{M}_{a,k} = \frac{1}{n_k} \sum_i a_i a_i^\top$):

$$\hat{\delta}_k = \frac{\|\hat{\Sigma}_{\psi,k} - \hat{M}_{a,k} \otimes \hat{\Gamma}_{e,k}\|_F}{\|\hat{\Sigma}_{\psi,k}\|_F}.$$

We set $\hat{\delta}_k := 0$ when $\|\hat{\Sigma}_{\psi,k}\|_F = 0$.

**Block dominance ($r$).** We compute layer-wise Fisher blocks $\hat{\mathcal{F}}_{ll}$ and cross-layer blocks $\hat{\mathcal{F}}_{lm}$ from gradient samples, then estimate

$$\hat{r} = \sqrt{\frac{\sum_{l \neq m} \|\hat{\mathcal{F}}_{lm}\|_F^2}{\sum_l \|\hat{\mathcal{F}}_{ll}\|_F^2}}.$$

**Error covariance alignment ($S_{\text{cov}}(\Gamma_e)$).** For $K$-class classification, define the error vector $e^{(k)} = \hat{p} - \mathbf{1}_y \in \mathbb{R}^K$ and the error second moment $\Gamma_{e,k} = \mathbb{E}[e^{(k)} e^{(k)\top}]$, a $K \times K$ matrix. To estimate $\Gamma_e$: (i) run forward passes on $n$ samples to obtain softmax outputs $\hat{p}$ and targets $y$; (ii) compute $e = \hat{p} - \mathbf{1}_y$ per sample; (iii) estimate $\hat{\Gamma}_e = \frac{1}{n} \sum_i e_i e_i^\top$; and (iv) compute

$$S_{\text{cov}}(\hat{\Gamma}_{e,i}, \hat{\Gamma}_{e,j}) = \frac{\langle \hat{\Gamma}_{e,i}, \hat{\Gamma}_{e,j} \rangle_F}{\|\hat{\Gamma}_{e,i}\|_F \|\hat{\Gamma}_{e,j}\|_F}.$$

Computational cost is $O(nK^2)$, tractable for ImageNet ($K$=1000) with $n \approx 1000$ samples. We set $S_{\text{cov}}(\hat{\Gamma}_{e,i}, \hat{\Gamma}_{e,j}) := 0$ when $\|\hat{\Gamma}_{e,i}\|_F = 0$ or $\|\hat{\Gamma}_{e,j}\|_F = 0$. For very large $K$, a diagonal proxy $S_{\text{cov}}(\text{diag}(\Gamma_{e,i}), \text{diag}(\Gamma_{e,j}))$ provides a cheaper $O(nK)$ approximation, though it is not a general lower bound.

### I.2. Diagnostic Experiments

**Setup.** We evaluate feedforward networks of varying depth and width on classification tasks. We measure: (i) penultimate-layer CKA, (ii) empirical head Fisher alignment $A_{\mathcal{F}}^{\text{head}}$, (iii) full-network Fisher alignment $A_{\mathcal{F}}^{\text{full}}$, and (iv) the structural quantities $\kappa$ (reported as $\hat{\kappa}$), $\delta$, $r$, and $w_{\text{head}}$ (head dominance; see (44)).

**CKA–Fisher correlation.** Across 30 task pairs with varying similarity, we observe strong correlation between CKA and full-network Fisher alignment (Pearson corr. $= 0.90$, Fisher-z 95% CI $[0.80, 0.95]$) and moderate correlation with head Fisher alignment (Pearson corr. $= 0.77$, Fisher-z 95% CI $[0.57, 0.88]$). The correlation between task similarity and CKA (Pearson corr. $= 0.86$, Fisher-z 95% CI $[0.72, 0.93]$) confirms that CKA captures meaningful task relationships. When the structural conditions in our theory degrade (e.g., larger $\hat{\kappa}$ or $\delta$, or weaker head dominance), the CKA–Fisher gap increases from a minimum of 0.04 to a maximum of 0.51.

**Coupling strength validation.** We verify that $\hat{\kappa}$ tracks representation coupling: for highly similar networks $\hat{\kappa} \approx 0$ (strong coupling), while for uncorrelated representations $\hat{\kappa} \approx 1$ after normalization (weak coupling). Intermediate task similarities yield intermediate $\hat{\kappa}$ values as expected.

**Block dominance across architectures.** We observe that block dominance $r$ varies dramatically with architecture:

- **Shallow-wide** (depth 1–2, width $\geq 32$): $r \approx 0.2$–$0.5$, $w_{\text{head}} \approx 0.94$–$0.99$.

- **Deep networks** (depth $\geq 3$): $r > 1$, $w_{\text{head}} < 0.5$.

*Table 12.* Independence violation ($\delta$) across architectures. All networks violate the Kronecker independence assumption, with deeper/narrower architectures exhibiting larger $\delta$.

| Architecture | Depth $\times$ Width | $\delta$ |
|---|---|---|
| Shallow-Wide | $1 \times 256$ | 0.36 |
| Shallow-Medium | $1 \times 128$ | 0.36 |
| Medium | $2 \times 128$ | 0.26 |
| Deep-Medium | $3 \times 64$ | 0.24 |
| Deep-Narrow | $4 \times 32$ | 0.31 |
| Very Deep-Narrow | $5 \times 16$ | 0.49 |
| *Average* | | 0.34 |

These ranges suggest that head dominance (large $w_{\text{head}}$) holds reliably only for shallow-wide architectures; deeper networks require the full-network diagnostics in Appendix O.

### I.3. Independence Violation Across Architectures

The Kronecker independence assumption ($\delta = 0$) is violated across all architectures tested ($\delta \approx 0.34$, Table 12). By Theorem 4.2, head Fisher alignment satisfies $A_{\text{F}}^{\text{head}}(i,j) = A_{\text{proxy}}(i,j) \cdot \rho_{ij}$ *exactly*, where $\rho_{ij}$ is the coupling correction.

This has two consequences:

1. **Empirical validation**: Head-level validation uses exact Fisher alignment via the kernel identity (Appendix A), while full-network LLM validation uses parameter-subsampled gradients across all layers for computational tractability.

2. **Negative proxy rank correlation on ViT**: On ViT-B/16 ($n = 40$ same-dataset pairs), Pearson $\text{Corr}(\hat{\rho}, A_{\text{proxy}}) = -0.85$, and the proxy is negatively correlated with exact Fisher (Pearson corr. $= -0.34$, Spearman corr. $= -0.30$; Table 5), indicating an inverted ranking signal. FA-CKA succeeds (Pearson corr. $= 0.79$, Spearman corr. $= 0.90$) because it is validated against exact Fisher.

**Architecture-dependent coupling.** Independence implies $\rho_{ij} = 1$ (Corollary 4.3), but $\rho \neq 1$ for both CNNs and transformers. The key difference is Pearson $\text{Corr}(\hat{\rho}, A_{\text{proxy}})$: positive on ResNet ($+0.88$, $n = 75$) and negative on ViT ($-0.85$, $n = 40$). Consistent with this, the proxy correlates positively with exact Fisher on ResNet (Pearson corr. $= 0.98$, Spearman corr. $= 0.69$; $n = 75$) and negatively on ViT (Pearson corr. $= -0.34$, Spearman corr. $= -0.30$; $n = 40$; Table 5). FA-CKA works regardless because it is validated against exact Fisher, not the proxy.

## J. Empirical Distribution of Coupling Correction $\hat{\rho}$

We compute the empirical coupling correction

$$\hat{\rho}_{ij} := \frac{A_{\text{F}}^{\text{exact}}(i,j)}{A_{\text{proxy}}(i,j)}, \qquad A_{\text{proxy}}(i,j) = S_{\text{cov}}(M_{a,i}, M_{a,j}) \cdot S_{\text{cov}}(\Gamma_{e,i}, \Gamma_{e,j}),$$

using exact head Fisher alignment $A_{\text{F}}^{\text{exact}}$ (Appendix A), as a plug-in estimator of $\rho_{ij}$. We retain pairs satisfying $A_{\text{F}}^{\text{exact}}(i,j) > \varepsilon$ and $A_{\text{proxy}}(i,j) > \varepsilon$, using $\varepsilon = 10^{-6}$. In our ViT/ResNet experiments, this filter keeps same-dataset pairs and excludes cross-dataset pairs under Assumption F.1. By Theorem 4.2, $\rho_{ij} = 1$ under independence. When coupling does not cancel, $\rho_{ij}$ deviates from 1; $\hat{\rho}$ captures that deviation empirically.

For ResNet-50, we use five datasets, three seeds, and two training modes (fine-tuned and linear probe), yielding six tasks per dataset and 15 within-dataset task pairs (total $n = 75$).

**Interpretation.** On ViT ($n = 40$), $\hat{\rho}$ ranges from 0.35 to 1.11, and Pearson $\text{Corr}(\hat{\rho}, A_{\text{proxy}}) = -0.85$. $\hat{\rho}$ is not constrained to be at most 1; in our runs it reaches 1.11. Empirically, high-proxy pairs tend to have lower $\hat{\rho}$. Consistent with this, the proxy is negatively correlated with exact Fisher (Pearson corr. $= -0.34$; Spearman corr. $= -0.30$; Table 5). On ResNet-50 ($n = 75$), $\hat{\rho}$ ranges from 0.15 to 0.99 and Pearson $\text{Corr}(\hat{\rho}, A_{\text{proxy}}) = +0.88$. The proxy correlates positively with exact Fisher (Pearson corr. $= +0.98$; Spearman corr. $= +0.69$), indicating moderate rank agreement.

*Table 13.* Distribution of coupling correction $\hat{\rho}_{ij}$ on ViT-B/16 and ResNet-50 (same-dataset pairs with nonzero proxy denominator). ViT shows adversarial coupling (Pearson $\text{Corr}(\hat{\rho}, A_{\text{proxy}}) = -0.85$); ResNet shows positive coupling (Pearson $\text{Corr}(\hat{\rho}, A_{\text{proxy}}) = +0.88$).

| Architecture | $n$ pairs | Mean $\hat{\rho}$ | Std | Min | Max |
|---|---|---|---|---|---|
| ViT-B/16 | 40 | 0.60 | 0.22 | 0.35 | 1.11 |
| ResNet-50 | 75 | 0.51 | 0.28 | 0.15 | 0.99 |

**Why FA-CKA works despite $\hat{\rho}$ variation.** FA-CKA (defined as $\text{CKA} \times S_{\text{cov}}(\Gamma_e)$) correlates well with exact Fisher (Pearson corr. $= 0.79$; Spearman corr. $= 0.90$; $n = 40$; Table 5). By contrast, $A_{\text{proxy}} = S_{\text{cov}}(M_a) \times S_{\text{cov}}(\Gamma_e)$ is negatively correlated (Pearson corr. $= -0.34$). Since $A_{\text{F}}^{\text{exact}} = S_{\text{cov}}(M_a) \cdot S_{\text{cov}}(\Gamma_e) \cdot \hat{\rho}$, we can write

$$\frac{\text{FA-CKA}(i,j)}{A_{\text{F}}^{\text{exact}}(i,j)} = \frac{\text{CKA}(i,j)}{S_{\text{cov}}(M_{a,i}, M_{a,j})} \cdot \frac{1}{\hat{\rho}_{ij}}.$$

Two factors explain this. First, on ViT, CKA differs from $S_{\text{cov}}(M_a)$: centering contributes (Appendix M), and representation mismatch $\kappa_{ij}$ can also drive the gap (Theorem G.1). Across same-dataset pairs ($n = 40$), the ratio $\text{CKA}(i,j)/S_{\text{cov}}(M_{a,i}, M_{a,j})$ averages about 2.28 and is strongly negatively correlated with $1/\hat{\rho}_{ij}$ (Pearson corr. $= -0.91$), which partially offsets coupling effects. Second, $S_{\text{cov}}(\Gamma_e)$ alone already correlates strongly with $A_{\text{F}}^{\text{exact}}$ (Pearson corr. $= 0.78$; $n = 40$), suggesting that the $\Gamma_e$ component drives much of the signal. FA-CKA is validated empirically against exact Fisher, rather than justified solely by the $\rho = 1$ proxy assumption.

### J.1. Proof of Remark 4.5: Proxy Approximation Bound

We prove that when the coupling tensor $C_k$ is small relative to the rank-1 part, the separable proxy is guaranteed to be accurate.

Let $x_k := \mu_{ae,k}$, $u_k := \mu_{a,k} \otimes \mu_{e,k}$, and $v_k := C_k$. Then $x_k = u_k + v_k$, where $u_k$ is the rank-1 component and $v_k$ is the coupling residual. Define $\varepsilon_k := \|v_k\|/\|u_k\|$.

**Lemma J.1** (Normalization perturbation). *If $\varepsilon_k < 1$, then $\|\hat{x}_k - \hat{u}_k\| \leq 2\varepsilon_k/(1 - \varepsilon_k)$, where $\hat{z} := z/\|z\|$ denotes unit normalization.*

*Proof.* We have $\|x_k\| = \|u_k + v_k\| \geq \|u_k\| - \|v_k\| = (1 - \varepsilon_k)\|u_k\|$. Thus,

$$\|\hat{x}_k - \hat{u}_k\| \leq \left\| \frac{v_k}{\|x_k\|} \right\| + \|u_k\| \cdot \left| \frac{1}{\|x_k\|} - \frac{1}{\|u_k\|} \right|$$
$$\leq \frac{\|v_k\|}{\|x_k\|} + \frac{\|u_k\| \cdot |\|x_k\| - \|u_k\||}{\|x_k\|\|u_k\|} \leq \frac{\|v_k\|}{\|x_k\|} + \frac{\|v_k\|}{\|x_k\|} = \frac{2\|v_k\|}{\|x_k\|} \leq \frac{2\varepsilon_k}{1 - \varepsilon_k}.$$

□

**Lemma J.2** (Cosine perturbation). *If $\|\hat{x}_i - \hat{u}_i\| \leq \eta_i$ and $\|\hat{x}_j - \hat{u}_j\| \leq \eta_j$, then for any $(i,j)$, $|\langle \hat{x}_i, \hat{x}_j \rangle - \langle \hat{u}_i, \hat{u}_j \rangle| \leq \eta_i + \eta_j + \eta_i\eta_j$.*

*Proof.* Write $\hat{x}_i = \hat{u}_i + \delta_i$ with $\|\delta_i\| \leq \eta_i$. Expanding the inner product yields

$$\langle \hat{x}_i, \hat{x}_j \rangle = \langle \hat{u}_i + \delta_i, \hat{u}_j + \delta_j \rangle = \langle \hat{u}_i, \hat{u}_j \rangle + \langle \delta_i, \hat{u}_j \rangle + \langle \hat{u}_i, \delta_j \rangle + \langle \delta_i, \delta_j \rangle.$$

By Cauchy-Schwarz, $|\langle \delta_i, \hat{u}_j \rangle| \leq \eta_i$, $|\langle \hat{u}_i, \delta_j \rangle| \leq \eta_j$, $|\langle \delta_i, \delta_j \rangle| \leq \eta_i\eta_j$. □

**Proof of Remark 4.5.** Combining Lemmas J.1 and J.2 with $\eta_k = 2\varepsilon_k/(1 - \varepsilon_k)$ yields the stated proxy approximation bound. Note that $A_{\text{proxy}}(i,j) = \cos(\mu_{a,i} \otimes \mu_{e,i}, \mu_{a,j} \otimes \mu_{e,j}) = \langle \hat{u}_i, \hat{u}_j \rangle$. Also, $A_F^{\text{head}}(i,j) = \langle \hat{x}_i, \hat{x}_j \rangle$. □

## K. RSA and SVCCA are Representation-Only Metrics

This appendix proves that RSA and SVCCA, like CKA, are representation-only metrics (Definition 3.1). Therefore, they inherit the same fundamental limitation (Theorem 3.2).

### K.1. RSA (Representational Similarity Analysis)

**Proposition K.1** (RSA is representation-only)**.** *RSA (Kriegeskorte et al., 2008) depends only on the representation Gram matrices $K_i = Z_i Z_i^\top$ and $K_j = Z_j Z_j^\top$ (equivalently, on all pairwise distances in $Z_i$ and $Z_j$).*

*Proof.* RSA is the Spearman rank correlation between representational dissimilarity matrices (RDMs):

$$\mathrm{RSA}(Z_i, Z_j) := \mathrm{Corr}_{\mathrm{Spearman}}(\mathrm{RDM}_i, \mathrm{RDM}_j)$$

where $\mathrm{RDM}_i$ is the vector of pairwise Euclidean distances $\{\|z_a^{(i)} - z_b^{(i)}\|_2 : a < b\}$ computed from $Z_i$. We use Euclidean RDMs; other common dissimilarities (e.g., correlation distance) are also representation-only by the same reasoning. We assume $Z_i$ and $Z_j$ are evaluated on the same probe set so the RDMs are comparable. We further assume $\mathrm{RDM}_i$ and $\mathrm{RDM}_j$ are nonconstant so Spearman correlation is well-defined. Because Spearman correlation is invariant to monotone transforms, we can work with squared distances.

For any two samples $z_a^{(i)}, z_b^{(i)} \in Z_i$, the squared distance is:

$$\|z_a^{(i)} - z_b^{(i)}\|^2 = \|z_a^{(i)}\|^2 + \|z_b^{(i)}\|^2 - 2(z_a^{(i)})^\top z_b^{(i)}$$
$$= (K_i)_{aa} + (K_i)_{bb} - 2(K_i)_{ab}$$

where $K_i := Z_i Z_i^\top$ is the Gram matrix.

Define the (squared) distance matrix $D_i$ with entries $D_{i,ab} = \|z_a^{(i)} - z_b^{(i)}\|_2^2$. Then

$$D_i = \mathbf{1}\,\mathrm{diag}(K_i)^\top + \mathrm{diag}(K_i)\,\mathbf{1}^\top - 2K_i$$

where $\mathbf{1}$ is the all-ones vector. Thus all pairwise distances are determined by $K_i$. Since $\mathrm{RDM}_i$ is the vector of entries $D_{i,ab}$ for $a < b$, it is a deterministic function of $K_i$ (and similarly for $j$). Therefore

$$\mathrm{RSA}(Z_i, Z_j) = \mathrm{Corr}_{\mathrm{Spearman}}(f(K_i), f(K_j)),$$

which depends only on $(Z_i, Z_j)$ through $(K_i, K_j)$ and not on errors or gradients. Hence RSA is representation-only. $\qquad\square$

**Corollary K.2** (RSA-Fisher gap)**.** *For tasks with shared backbone ($Z_i = Z_j$), nonconstant RDMs, and disjoint label subsets under the shared output-space embedding (Assumption F.1):*

$$RSA(Z_i, Z_j) = 1, \quad while \quad A_{\mathrm{F}}^{\mathrm{head}}(i, j) = 0.$$

This follows directly from Theorem 3.2 since RSA is a representation-only metric (Proposition K.1).

### K.2. SVCCA (Singular Vector CCA)

**Proposition K.3** (SVCCA is representation-only)**.** *SVCCA (Raghu et al., 2017a) depends only on covariances of the SVD-reduced representations (i.e., on $(\tilde{\Sigma}_i, \tilde{\Sigma}_j, \tilde{\Sigma}_{ij})$ computed from $Z_i, Z_j$).*

*Proof.* SVCCA is defined as:

$$\mathrm{SVCCA}(Z_i, Z_j) := \frac{1}{k} \sum_{r=1}^{k} \rho_r$$

where $\rho_1, \ldots, \rho_k$ are the canonical correlation coefficients between the SVD-reduced representations (with $k = \min(r_i, r_j)$).

Let $Z_i, Z_j$ be centered over the probe samples ($n$ rows). Compute SVDs $Z_i = U_i S_i V_i^\top$ and $Z_j = U_j S_j V_j^\top$. Retain the top components (e.g., enough to explain a fixed variance fraction), yielding reduced representations $\tilde{Z}_i := U_i(:, 1:r_i)S_i(1:r_i, 1:r_i)$ and $\tilde{Z}_j := U_j(:, 1:r_j)S_j(1:r_j, 1:r_j)$. CCA is then computed on the reduced representations using only their covariance blocks.

Define the (regularized) covariances

$$\tilde{\Sigma}_i := \tfrac{1}{n}\tilde{Z}_i^\top \tilde{Z}_i + \lambda I, \quad \tilde{\Sigma}_j := \tfrac{1}{n}\tilde{Z}_j^\top \tilde{Z}_j + \lambda I, \quad \tilde{\Sigma}_{ij} := \tfrac{1}{n}\tilde{Z}_i^\top \tilde{Z}_j,$$

with a small $\lambda \geq 0$ for numerical stability, following the official implementation (Raghu et al., 2017b). The squared canonical correlations are the eigenvalues of $\tilde{\Sigma}_i^{-1} \tilde{\Sigma}_{ij} \tilde{\Sigma}_j^{-1} \tilde{\Sigma}_{ji}$, where $\tilde{\Sigma}_{ji} = \tilde{\Sigma}_{ij}^\top$ (or an equivalent generalized eigenproblem). Hence $\{\rho_r\}$ depend only on $(\tilde{\Sigma}_i, \tilde{\Sigma}_j, \tilde{\Sigma}_{ij})$.

All subsequent CCA computations operate exclusively on these covariance blocks; the raw activations are never accessed after forming them. Therefore, SVCCA depends only on covariances of the reduced representations and is a representation-only metric. □

**Implementation details (Appendix K.3).** Our experiments use the official SVCCA implementation (Raghu et al., 2017a). It mean-centers activations (via `np.cov`), prunes low-variance directions below $\epsilon = 10^{-10}$ (equivalently $\lambda = 10^{-10}$ in Appendix K.2), and reports the mean canonical correlation up to the default truncation threshold 0.98. A standard 99% variance SVD truncation (Raghu et al., 2017a) yields the same representation-only conclusion.

**Corollary K.4** (SVCCA-Fisher gap). *For tasks with shared backbone ($Z_i = Z_j$) and disjoint label subsets under the shared output-space embedding (Assumption F.1):*

$$SVCCA(Z_i, Z_j) = 1, \quad while \quad A_{\mathrm{F}}^{\mathrm{head}}(i, j) = 0.$$

### K.3. Empirical Verification

We verify these theoretical results numerically:

**Implementation details.** Test 1 uses $n = 200$, $d = 64$, $K = 10$, seed $= 100$. Test 2 (table) uses $n = 500$, $d = 128$, seed $= 42$ (reset for each $K$) with $K \in \{5, 10, 20, 50, 100\}$. Following the official implementation, SVCCA uses $\epsilon = 10^{-10}$ (equivalently $\lambda = 10^{-10}$) and the default truncation threshold 0.98.

**Test 1: Invariance to error structure.** For two task pairs with *identical representations* but different error covariances:

- Pair 1: $S_{\mathrm{cov}}(\Gamma_{e,1}) = 0.999$

- Pair 2: $S_{\mathrm{cov}}(\Gamma_{e,2}) = 0.000$ (disjoint classes)

Result: $|\mathrm{RSA}(\text{pair 1}) - \mathrm{RSA}(\text{pair 2})| < 10^{-8}$ and $|\mathrm{SVCCA}(\text{pair 1}) - \mathrm{SVCCA}(\text{pair 2})| < 10^{-8}$.

Both metrics are *completely invariant* to error structure, as predicted by the theory.

**Test 2: RSA-Fisher and SVCCA-Fisher gaps.** For disjoint $K$-class tasks with shared representations:

| $K$ | RSA | SVCCA | $A_{\mathrm{F}}^{\mathrm{head}}$ | Gap |
|-----|-----|-------|------|-----|
| 5 | 1.000 | 1.000 | 0.000 | 1.000 |
| 10 | 1.000 | 1.000 | 0.000 | 1.000 |
| 20 | 1.000 | 1.000 | 0.000 | 1.000 |
| 50 | 1.000 | 1.000 | 0.000 | 1.000 |
| 100 | 1.000 | 1.000 | 0.000 | 1.000 |

Both RSA and SVCCA exhibit the same pathological gap as CKA: perfect similarity despite orthogonal gradients.

## L. Explicit Perturbation Bound for Alignment

This appendix states a unit-normalization perturbation bound (with constant 2) used in later alignment bounds.

**Lemma L.1** (Normalization perturbation). *Let $x = x_0 + r$, where $x \neq 0$ and $x_0 \neq 0$. Let $\| \cdot \|$ be any norm (e.g., the Frobenius norm for matrices), and define $\delta := \|r\|/\|x\|$. Then*

$$\left\| \frac{x}{\|x\|} - \frac{x_0}{\|x_0\|} \right\| \leq 2\delta.$$

*Proof.* Observe that $\frac{x}{\|x\|} - \frac{x_0}{\|x_0\|} = \frac{r}{\|x\|} + x_0 \left( \frac{1}{\|x\|} - \frac{1}{\|x_0\|} \right)$. Taking norms and applying the reverse triangle inequality $|\|x\| - \|x_0\|| \leq \|r\|$, we obtain

$$\left\| \frac{x}{\|x\|} - \frac{x_0}{\|x_0\|} \right\| \leq \frac{\|r\|}{\|x\|} + \frac{|\|x\| - \|x_0\||}{\|x\|}$$

$$\leq 2 \frac{\|r\|}{\|x\|} = 2\delta.$$

$\square$

If we instead define $\varepsilon := \|r\|/\|x_0\| < 1$, then $\|x\| = \|x_0 + r\| \geq \|x_0\| - \|r\| = (1 - \varepsilon)\|x_0\|$. Consequently,

$$\left\| \frac{x}{\|x\|} - \frac{x_0}{\|x_0\|} \right\| \leq \frac{2\varepsilon}{1 - \varepsilon}.$$

**Corollary L.2** (Alignment error bound). *Let* $\Sigma_i = \Sigma_i^{(0)} + R_i$ *and* $\Sigma_j = \Sigma_j^{(0)} + R_j$. *Assume* $\|\Sigma_i\|_F, \|\Sigma_j\|_F > 0$ *and* $\|\Sigma_i^{(0)}\|_F, \|\Sigma_j^{(0)}\|_F > 0$. *For* $k \in \{i, j\}$, *define* $\varepsilon_{R,k} := \|R_k\|_F/\|\Sigma_k\|_F$. *Then*

$$\left| S_{\mathrm{cov}}(\Sigma_i, \Sigma_j) - S_{\mathrm{cov}}(\Sigma_i^{(0)}, \Sigma_j^{(0)}) \right| \leq 2(\varepsilon_{R,i} + \varepsilon_{R,j}).$$

*If all four matrices are positive semidefinite (PSD), then* $S_{\mathrm{cov}}(\cdot, \cdot) \in [0, 1]$ *and the bound can be clipped as*

$$\left| S_{\mathrm{cov}}(\Sigma_i, \Sigma_j) - S_{\mathrm{cov}}(\Sigma_i^{(0)}, \Sigma_j^{(0)}) \right| \leq \min\{1, 2(\varepsilon_{R,i} + \varepsilon_{R,j})\}.$$

*Proof.* Let $u = \Sigma_i/\|\Sigma_i\|_F$, $u_0 = \Sigma_i^{(0)}/\|\Sigma_i^{(0)}\|_F$, and define $v = \Sigma_j/\|\Sigma_j\|_F$, $v_0 = \Sigma_j^{(0)}/\|\Sigma_j^{(0)}\|_F$. Then

$$\left| S_{\mathrm{cov}}(\Sigma_i, \Sigma_j) - S_{\mathrm{cov}}(\Sigma_i^{(0)}, \Sigma_j^{(0)}) \right| = |\langle u, v \rangle_F - \langle u_0, v_0 \rangle_F| \leq \|u - u_0\|_F + \|v - v_0\|_F.$$

By Lemma L.1 with $x = \Sigma_i$, $x_0 = \Sigma_i^{(0)}$, and $r = R_i$, we have $\|u - u_0\|_F \leq 2\|R_i\|_F/\|\Sigma_i\|_F = 2\varepsilon_{R,i}$, and likewise $\|v - v_0\|_F \leq 2\varepsilon_{R,j}$. The claim follows. $\square$

**Interpretation.** In the head-Fisher setting, set $\Sigma_k^{(0)} := M_{a,k} \otimes \Gamma_{e,k}$. Then $R_k$ is the coupling residual from Remark 4.4. Under this identification, $\varepsilon_{R,k}$ matches the reported $\delta \approx 0.34$. The corollary then yields a worst-case bound of $2(0.34 + 0.34) = 1.36$. Because $S_{\mathrm{cov}}(\cdot, \cdot) \in [0, 1]$ for PSD matrices, we may clip the bound at 1. The worst-case term $2(\varepsilon_{R,i} + \varepsilon_{R,j})$ can be vacuous. Empirically, observed deviations are typically much smaller (see Appendix J).

## M. Proof of the Mean-Term Correction Lemma

This appendix proves Lemma M.1, which quantifies the gap between covariance alignment computed from centered and uncentered second moments.

**Lemma M.1** (Mean-term correction). *Consider two tasks with activation second moments* $M_{a,i}, M_{a,j}$ *and centered covariances* $\Sigma_{a,i}, \Sigma_{a,j}$, *with means* $\mu_{a,i}, \mu_{a,j}$. *Assume* $\Sigma_{a,i}, \Sigma_{a,j} \succeq 0$ *and* $\|\Sigma_{a,i}\|_F, \|\Sigma_{a,j}\|_F > 0$ *(hence* $\|M_{a,i}\|_F, \|M_{a,j}\|_F > 0$*). Then the covariance alignment gap is bounded as:*

$$\left| S_{\mathrm{cov}}(M_{a,i}, M_{a,j}) - S_{\mathrm{cov}}(\Sigma_{a,i}, \Sigma_{a,j}) \right| \leq 2(\varepsilon_{\mu,i} + \varepsilon_{\mu,j})$$

*where* $\varepsilon_{\mu,k} := \|\mu_{a,k}\|_2^2/\|\Sigma_{a,k}\|_F$ *is the ratio of squared-mean magnitude to covariance Frobenius norm.*

*Proof.* By definition, $M_{a,k} = \Sigma_{a,k} + \mu_{a,k}\mu_{a,k}^\top$. Let $A = \Sigma_{a,i}$, $B = \Sigma_{a,j}$, $\Delta_A = \mu_{a,i}\mu_{a,i}^\top$, and $\Delta_B = \mu_{a,j}\mu_{a,j}^\top$. Define the normalized matrices $u = \frac{A + \Delta_A}{\|A + \Delta_A\|_F}$ and $u_0 = \frac{A}{\|A\|_F}$. Similarly, define $v = \frac{B + \Delta_B}{\|B + \Delta_B\|_F}$ and $v_0 = \frac{B}{\|B\|_F}$. Viewing matrices as vectors under the Frobenius inner product, we have

$$S_{\mathrm{cov}}(M_{a,i}, M_{a,j}) = \langle u, v \rangle_F, \quad S_{\mathrm{cov}}(\Sigma_{a,i}, \Sigma_{a,j}) = \langle u_0, v_0 \rangle_F.$$

Therefore,
$$\left|\langle u, v\rangle_F - \langle u_0, v_0\rangle_F\right| = \left|\langle u - u_0, v\rangle_F + \langle u_0, v - v_0\rangle_F\right| \leq \|u - u_0\|_F + \|v - v_0\|_F.$$

The last inequality uses Cauchy-Schwarz with $\|u\|_F = \|u_0\|_F = \|v\|_F = \|v_0\|_F = 1$. Applying Lemma L.1 with $x = A + \Delta_A$, $x_0 = A$, and $r = \Delta_A$, we obtain

$$\|u - u_0\|_F \leq 2\frac{\|\Delta_A\|_F}{\|A + \Delta_A\|_F}.$$

Because $A, \Delta_A \succeq 0$, we have $\langle A, \Delta_A\rangle_F \geq 0$, and thus $\|A + \Delta_A\|_F^2 = \|A\|_F^2 + \|\Delta_A\|_F^2 + 2\langle A, \Delta_A\rangle_F \geq \|A\|_F^2$. Substituting into the previous bound yields $\|u - u_0\|_F \leq 2\|\Delta_A\|_F/\|A\|_F$. Noting that $\|\Delta_A\|_F = \|\mu_{a,i}\|_2^2$, this gives $\|u - u_0\|_F \leq 2\varepsilon_{\mu,i}$. The same argument yields $\|v - v_0\|_F \leq 2\varepsilon_{\mu,j}$. Therefore,

$$\left|S_{\mathrm{cov}}(M_{a,i}, M_{a,j}) - S_{\mathrm{cov}}(\Sigma_{a,i}, \Sigma_{a,j})\right| \leq 2(\varepsilon_{\mu,i} + \varepsilon_{\mu,j}),$$

as claimed. $\qquad\square$

**Remark (tighter denominator).** Lemma L.1 directly gives $\|u - u_0\|_F \leq 2\|\Delta_A\|_F/\|A + \Delta_A\|_F$ and similarly for $v$. Define $\tilde{\varepsilon}_{\mu,k} := \|\mu_{a,k}\|_2^2/\|M_{a,k}\|_F$. Then the always-valid bound $\left|S_{\mathrm{cov}}(M_{a,i}, M_{a,j}) - S_{\mathrm{cov}}(\Sigma_{a,i}, \Sigma_{a,j})\right| \leq 2(\tilde{\varepsilon}_{\mu,i} + \tilde{\varepsilon}_{\mu,j})$. Since $\|M_{a,k}\|_F \geq \|\Sigma_{a,k}\|_F$ under $\Sigma_{a,k} \succeq 0$, this bound is never looser than the stated bound.

**Interpretation.** If $\varepsilon_{\mu,k}$ is large, the bound can be vacuous (i.e., greater than 1), but it still quantifies a worst-case impact of mean terms; reporting $\varepsilon_{\mu,k}$ keeps the approximation auditable.

**Empirical mean-term sizes (ViT-B/16).** Using the ViT-B/16 validation data from Appendix J (20 tasks; 40 same-dataset pairs), we find $\varepsilon_{\mu,k} \in [0.78, 1.35]$ with a median of 0.96. Accordingly, the worst-case bound $2(\varepsilon_{\mu,i} + \varepsilon_{\mu,j})$ ranges from 3.16 to 5.29 across same-dataset pairs (vacuous). The observed gap $|S_{\mathrm{cov}}(M_{a,i}, M_{a,j}) - S_{\mathrm{cov}}(\Sigma_{a,i}, \Sigma_{a,j})|$ ranges from 0.007 to 0.147 (median 0.085).

## N. Proof of Theorem 3.2

This appendix provides a full proof of Theorem 3.2, showing that representation-only metrics are fundamentally blind to error structure.

*Proof of Theorem 3.2.* Consider two classification tasks $\mathcal{T}_i, \mathcal{T}_j$ sharing an encoder $f_\theta : \mathcal{X} \to \mathbb{R}^d$ but using disjoint label sets $\mathcal{Y}_i \cap \mathcal{Y}_j = \emptyset$.

**Setup.** Both tasks share a head parameter matrix $W \in \mathbb{R}^{K \times d}$ where $K = |\mathcal{Y}_i| + |\mathcal{Y}_j|$. Task $i$ uses rows $1, \ldots, |\mathcal{Y}_i|$ (with masked softmax over this block), while task $j$ uses rows $|\mathcal{Y}_i| + 1, \ldots, K$. This shared-output embedding with masked softmax (equivalently logits set to $-\infty$ outside each task's label block) ensures the label spaces are orthogonally embedded in the shared output space (Assumption F.1; see Appendix F). This construction is regime-specific: without masked softmax (i.e., with a full $K$-way softmax even if weights are zero-padded), disjoint label sets do not necessarily imply orthogonal errors.

**Gradient structure.** By Lemma 4.1 (affine head) and cross-entropy loss, the gradient w.r.t. $\mathrm{vec}(W)$ at sample $(x, y)$ is:

$$g_k = a \otimes e_k, \quad \text{where } a = f_\theta(x) \in \mathbb{R}^d, \ e_k = \hat{p}_k - \mathbf{1}_y \in \mathbb{R}^K.$$

Here $\hat{p}_k$ is the masked softmax over the task's label block (equivalently logits set to $-\infty$ outside the block; Assumption F.1) and $\mathbf{1}_y := P_k \tilde{y}_k$ (with $P_k, \tilde{y}_k$ as in Assumption F.1) is the one-hot target embedded in $\mathbb{R}^K$.

**Disjoint support.** Since task $i$ only activates rows $1, \ldots, |\mathcal{Y}_i|$ and task $j$ only activates rows $|\mathcal{Y}_i| + 1, \ldots, K$:

$$\mathrm{supp}(e_i) \subseteq \{1, \ldots, |\mathcal{Y}_i|\}, \quad \mathrm{supp}(e_j) \subseteq \{|\mathcal{Y}_i| + 1, \ldots, K\}.$$

These supports are disjoint, so $e_i^\top e_j = 0$ for any pair of samples.

**Fisher inner product.** Let $\mathcal{F}_k^{\text{head}} := \mathbb{E}[g_k g_k^\top]$ be the head Fisher. By Lemma D.2 with independent samples from $\mathcal{D}_i$ and $\mathcal{D}_j$:

$$\langle \mathcal{F}_i^{\text{head}}, \mathcal{F}_j^{\text{head}} \rangle_F = \mathbb{E}_{(x_i,y_i)\sim\mathcal{D}_i,(x_j,y_j)\sim\mathcal{D}_j}\left[(g_i^\top g_j)^2\right].$$

Using the Kronecker structure:

$$g_i^\top g_j = (a_i \otimes e_i)^\top (a_j \otimes e_j) = (a_i^\top a_j)(e_i^\top e_j) = 0,$$

since $e_i^\top e_j = 0$ by disjoint support. Therefore $\langle \mathcal{F}_i^{\text{head}}, \mathcal{F}_j^{\text{head}} \rangle_F = 0$, which implies:

$$A_{\text{F}}^{\text{head}}(i,j) = \frac{\langle \mathcal{F}_i^{\text{head}}, \mathcal{F}_j^{\text{head}} \rangle_F}{\|\mathcal{F}_i^{\text{head}}\|_F \|\mathcal{F}_j^{\text{head}}\|_F} = 0.$$

Since $\mathcal{F}_k^{\text{head}} \succeq 0$, we have $\mathcal{F}_k^{\text{head}} = 0$ iff $\text{tr}(\mathcal{F}_k^{\text{head}}) = 0$. Moreover $\text{tr}(\mathcal{F}_k^{\text{head}}) = \mathbb{E}\|g_k\|_2^2$. Thus if $\mathbb{P}(\|a\|_2 > 0 \wedge \|e_k\|_2 > 0) > 0$ (equivalently $\mathbb{E}\|g_k\|_2^2 > 0$), then $\|\mathcal{F}_k^{\text{head}}\|_F > 0$ for $k \in \{i,j\}$, hence $A_{\text{F}}^{\text{head}}(i,i) = A_{\text{F}}^{\text{head}}(j,j) = 1$.

**Representation metric blindness.** On the shared probe set, the encoder is shared, so $Z_i = Z_j$. Any representation-only metric $M$ therefore satisfies $M(Z_i, Z_j) = M(Z_i, Z_i)$ (and equals 1 if $M$ is normalized). Yet $A_{\text{F}}^{\text{head}}(i,j) = 0$ while $A_{\text{F}}^{\text{head}}(i,i) = 1$ for nondegenerate tasks.

The gap arises because representation metrics access only the marginal distribution of activations $a$, while Fisher alignment depends on the joint distribution of $(a, e)$. The error vectors $e_i, e_j$—which encode label structure—are invisible to any representation-only metric. $\square$

### N.1. Full Proofs for CKA Non-Identifiability

This section provides the complete proofs of the linear CKA identity (Eq. (2)) and Corollary 3.3, which appear in the main text.

*Proof of Eq. (2).* Let $Z_i \in \mathbb{R}^{n \times d_i}$ and $Z_j \in \mathbb{R}^{n \times d_j}$ be representation matrices evaluated on a shared probe set of $n$ inputs, with columns centered: $Z_i^\top \mathbf{1} = 0$ and $Z_j^\top \mathbf{1} = 0$. Define the (centered) empirical covariances $\hat{\Sigma}_i := \frac{1}{n} Z_i^\top Z_i$, $\hat{\Sigma}_j := \frac{1}{n} Z_j^\top Z_j$, $\hat{\Sigma}_{ij} := \frac{1}{n} Z_i^\top Z_j$. Since $Z_i^\top Z_j = n\hat{\Sigma}_{ij}$ and $Z_i^\top Z_i = n\hat{\Sigma}_i$ (and similarly for $j$), we have

$$\frac{\|Z_i^\top Z_j\|_F^2}{\|Z_i^\top Z_i\|_F \|Z_j^\top Z_j\|_F} = \frac{n^2 \|\hat{\Sigma}_{ij}\|_F^2}{(n\|\hat{\Sigma}_i\|_F)(n\|\hat{\Sigma}_j\|_F)} = \frac{\|\hat{\Sigma}_{ij}\|_F^2}{\|\hat{\Sigma}_i\|_F \|\hat{\Sigma}_j\|_F}. \qquad \square$$

*Proof of Corollary 3.3.* By Theorem 3.2, there exist tasks $i \neq j$ with $Z_i = Z_j$ and $A_{\text{F}}^{\text{head}}(i,j) = 0$, while $A_{\text{F}}^{\text{head}}(i,i) = 1$ for the self-pair. Since $Z_i = Z_j$, we have $\text{CKA}(Z_i, Z_j) = \text{CKA}(Z_i, Z_i)$ (and equals 1 when $\|\hat{\Sigma}_i\|_F > 0$). Thus no single-valued function $g$ can map that one CKA value to both 0 and 1. The same argument applies to $S_{\text{cov}}(\Gamma_{e,i}, \Gamma_{e,j})$ since in the construction $\Gamma_{e,i}$ and $\Gamma_{e,j}$ have disjoint support, yielding $S_{\text{cov}} = 0$. $\square$

## O. Extension to Full Networks

The main text establishes a rigorous bridge between CKA and *head* (final layer) Fisher alignment. This appendix provides **exact structural decompositions** that relate full-network Fisher alignment to head Fisher alignment, with measurable diagnostics that explain when and why the reduction works.

**Scope.** The block-diagonal/off-diagonal decompositions below are purely algebraic and hold for any parameter partition. The exact three-factor identity in Section O.6 additionally assumes that per-layer gradients are rank-1 (affine layers without parameter sharing). For convolutional or attention layers, the identity changes because per-example gradients are sums of outer products across positions. Appendix C gives the exact shared-parameter kernel identity and an unbiased sketch. Our empirical validation in this appendix relies on the block-decomposition diagnostics.

*Table 14.* **Full-network diagnostic summary (Corollary O.8).** We report the profile cosine $c_r$ (mean/min across pairs), the off-diagonal discrepancy $\Delta_{\text{off}}$, and the observed proxy gap $|A_F^{\text{full}} - A_F^{\text{blk}}|$. The 67–72B row aggregates ranges across 8 models.

| Model | $c_r$ | $\Delta_{\text{off}}$ | Gap |
|---|---|---|---|
| ViT-B/16 | 1.00/0.99 | 0.09 | 0.08 |
| Llama-8B | 1.00/0.98 | 0.39 | 0.33 |
| Qwen-14B | 0.99/0.96 | 0.50 | 0.24 |
| 67–72B (8) | .95–1.0 | .30–.75 | .16–.46 |

## O.1. Full-Network Fisher Block Structure

Consider an $L$-layer network with parameters $\theta = (\theta_1, \ldots, \theta_L)$ where $\theta_l = \text{vec}(W_l)$. The full-network Fisher matrix has block structure:

$$
\mathcal{F} = \begin{bmatrix} \mathcal{F}_{11} & \mathcal{F}_{12} & \cdots & \mathcal{F}_{1L} \\ \mathcal{F}_{21} & \mathcal{F}_{22} & \cdots & \mathcal{F}_{2L} \\ \vdots & \vdots & \ddots & \vdots \\ \mathcal{F}_{L1} & \mathcal{F}_{L2} & \cdots & \mathcal{F}_{LL} \end{bmatrix}, \tag{27}
$$

where $\mathcal{F}_{ll} = \mathbb{E}[g_l g_l^\top]$ are the layer-wise Fisher blocks and $\mathcal{F}_{lm} = \mathbb{E}[g_l g_m^\top]$ for $l \neq m$ are cross-layer blocks.

**Exact algebraic split.** Define, for each task $k$:

- **Block-diagonal part**: $D_k := \text{blkdiag}(\mathcal{F}_{11}^{(k)}, \ldots, \mathcal{F}_{LL}^{(k)})$
- **Off-diagonal part**: $O_k := \mathcal{F}^{(k)} - D_k$

These have **disjoint support** in block coordinates: $D_k$ is nonzero only on diagonal blocks $(\ell, \ell)$, while $O_k$ is nonzero only on off-diagonal blocks $(\ell, m)$ with $\ell \neq m$. This implies $\langle D_k, O_k \rangle_F = 0$ and $\|\mathcal{F}^{(k)}\|_F^2 = \|D_k\|_F^2 + \|O_k\|_F^2$.

**Assumption O.1** (Nondegeneracy). We assume $\|D_k\|_F > 0$ (nontrivial block-diagonal Fisher) and, when $A_{\text{off}}$ is defined, $\|O_k\|_F > 0$. When head/rest quantities are defined below, we also assume $\|H_k\|_F > 0$ and $\|R_k\|_F > 0$.

**Degenerate cases.** If any denominator is zero, we use $\varepsilon$-regularized norms $\|X\|_{F,\varepsilon} := \max(\|X\|_F, \varepsilon)$ and set the corresponding alignment to $\langle X, Y \rangle_F / (\|X\|_{F,\varepsilon} \|Y\|_{F,\varepsilon})$ (which yields 0 when $X = 0$ or $Y = 0$). Weights are computed directly from norms via the first equalities in (32) and (44); ratios $r_k, \tau_k$ are used only when denominators are nonzero.

**Definition O.2** (Alignment diagnostics). Define the following measurable quantities for tasks $i, j$:

$$
A_F^{\text{full}}(i, j) := \frac{\langle \mathcal{F}_i, \mathcal{F}_j \rangle_F}{\|\mathcal{F}_i\|_F \|\mathcal{F}_j\|_F} \qquad \text{(full-network alignment)}, \tag{28}
$$

$$
A_{\text{blk}}(i, j) := \frac{\langle D_i, D_j \rangle_F}{\|D_i\|_F \|D_j\|_F} \qquad \text{(block-diagonal alignment)}, \tag{29}
$$

$$
A_{\text{off}}(i, j) := \frac{\langle O_i, O_j \rangle_F}{\|O_i\|_F \|O_j\|_F} \qquad \text{(off-diagonal alignment)}, \tag{30}
$$

$$
r_k := \frac{\|O_k\|_F}{\|D_k\|_F} \qquad \text{(cross-layer energy ratio; defined when } \|D_k\|_F > 0\text{)}. \tag{31}
$$

## O.2. Step 1: Exact Decomposition for Full-to-Block-Diagonal

The key insight is that mixed terms vanish by disjoint support, yielding an **exact weighted sum** with nonnegative weights.

**Definition O.3** (Decomposition weights). Define the following weights:

$$
w_{\text{blk}} := \frac{\|D_i\|_F \|D_j\|_F}{\|\mathcal{F}_i\|_F \|\mathcal{F}_j\|_F} = \frac{1}{\sqrt{1 + r_i^2} \sqrt{1 + r_j^2}}, \tag{32}
$$

$$
w_{\text{off}} := \frac{\|O_i\|_F \|O_j\|_F}{\|\mathcal{F}_i\|_F \|\mathcal{F}_j\|_F} = \frac{r_i r_j}{\sqrt{1 + r_i^2} \sqrt{1 + r_j^2}} = r_i r_j \cdot w_{\text{blk}}. \tag{33}
$$

The equalities involving $r_i, r_j$ assume $\|D_k\|_F > 0$; otherwise use the norm-based first equality.

*Remark* O.4 (Weights do not sum to 1 in general). By Cauchy–Schwarz on vectors $(1, r_i)$ and $(1, r_j)$ (for nondegenerate $r_i, r_j$):

$$w_{\text{blk}} + w_{\text{off}} = \frac{1 + r_i r_j}{\sqrt{1 + r_i^2}\sqrt{1 + r_j^2}} \leq 1, \tag{34}$$

with equality if and only if $r_i = r_j$. Thus we have a nonnegative weighted sum that is a true convex combination only when tasks have identical cross-layer energy profiles.

**Definition O.5** (Profile cosine). Define the **profile cosine**:

$$c_r := w_{\text{blk}} + w_{\text{off}} = \frac{1 + r_i r_j}{\sqrt{1 + r_i^2}\sqrt{1 + r_j^2}} \in (0, 1]. \tag{35}$$

This measures the alignment of the energy profiles $(\|D_k\|_F, \|O_k\|_F)$ between tasks.

**Lemma O.6** (Exact full-to-block-diagonal decomposition). *For any two tasks $i, j$ satisfying Assumption O.1:*

$$\boxed{A_F^{\text{full}}(i, j) = w_{\text{blk}} \cdot A_{\text{blk}}(i, j) + w_{\text{off}} \cdot A_{\text{off}}(i, j)} \tag{36}$$

*This is an **exact identity**.*

*Proof.* By disjoint block support, the cross terms vanish identically:

$$\langle D_i, O_j \rangle_F = 0, \quad \langle O_i, D_j \rangle_F = 0.$$

This follows from block structure: $D_i$ has nonzero entries only on diagonal blocks $(\ell, \ell)$, while $O_j$ has nonzero entries only on off-diagonal blocks $(\ell, m)$ with $\ell \neq m$.

Therefore:

$$\langle \mathcal{F}_i, \mathcal{F}_j \rangle_F = \langle D_i, D_j \rangle_F + \langle O_i, O_j \rangle_F.$$

Expanding $A_F^{\text{full}}$:

$$\begin{aligned}
A_F^{\text{full}} &= \frac{\langle D_i, D_j \rangle_F + \langle O_i, O_j \rangle_F}{\|\mathcal{F}_i\|_F \|\mathcal{F}_j\|_F} \\
&= \frac{\|D_i\|_F \|D_j\|_F}{\|\mathcal{F}_i\|_F \|\mathcal{F}_j\|_F} \cdot A_{\text{blk}} + \frac{\|O_i\|_F \|O_j\|_F}{\|\mathcal{F}_i\|_F \|\mathcal{F}_j\|_F} \cdot A_{\text{off}} \\
&= w_{\text{blk}} \cdot A_{\text{blk}} + w_{\text{off}} \cdot A_{\text{off}}. \qquad \square
\end{aligned}$$

**Exact gap identity.** Rewriting the weighted sum using the profile cosine:

$$A_F^{\text{full}} = (w_{\text{blk}} + w_{\text{off}})A_{\text{blk}} + w_{\text{off}}(A_{\text{off}} - A_{\text{blk}}) = c_r \cdot A_{\text{blk}} + w_{\text{off}}(A_{\text{off}} - A_{\text{blk}}). \tag{37}$$

Thus the gap has the exact form:

$$\boxed{A_F^{\text{full}} - A_{\text{blk}} = (c_r - 1)A_{\text{blk}} + w_{\text{off}}(A_{\text{off}} - A_{\text{blk}})} \tag{38}$$

**Definition O.7** (Off-diagonal non-adversariality). Define the **off-diagonal discrepancy**:

$$\Delta_{\text{off}} := |A_{\text{off}} - A_{\text{blk}}|. \tag{39}$$

This measures how "adversarial" the off-diagonal alignment is relative to the block-diagonal alignment. Note that $A_{\text{off}}$ can be negative since off-diagonal blocks $O_k$ are not PSD, but $|A_{\text{off}}| \leq 1$ still holds by Cauchy–Schwarz on the Frobenius inner product.

**Corollary O.8** (Sharp bound via profile cosine and non-adversariality). *Since $|A_{\text{blk}}| \leq 1$:*

$$\boxed{|A_F^{\text{full}} - A_{\text{blk}}| \leq (1 - c_r) + w_{\text{off}} \cdot \Delta_{\text{off}}} \tag{40}$$

*This bound does not require the energy ratios $r_i$ or $r_j$ to be less than 1.*

**Why this works for deep nets.** The bound has two measurable terms; each is small when:

- **Profile mismatch**: $(1 - c_r)$ is small when $r_i \approx r_j$ (tasks have similar cross-layer energy profiles).

- **Adversarial off-diagonal**: $w_{\text{off}} \cdot \Delta_{\text{off}}$ is small when $\Delta_{\text{off}}$ is small (off-diagonal alignment tracks block-diagonal).

When tasks share similar architectures and training regimes, both conditions typically hold.

### O.3. Step 2: Block-Diagonal to Head Fisher

Now split the block-diagonal part into head vs. rest:

- **Head block**: $H_k := \mathcal{F}_{LL}^{(k)}$

- **Non-head blocks**: $R_k := \mathrm{blkdiag}(\mathcal{F}_{11}^{(k)}, \ldots, \mathcal{F}_{L-1,L-1}^{(k)})$

Again, these have disjoint support: $\langle H_k, R_k \rangle_F = 0$ and $\|D_k\|_F^2 = \|H_k\|_F^2 + \|R_k\|_F^2$.

**Definition O.9** (Head dominance diagnostics). Define:

$$A_{\text{head}}(i,j) := \frac{\langle H_i, H_j \rangle_F}{\|H_i\|_F \|H_j\|_F} \qquad \text{(head alignment)}, \tag{41}$$

$$A_{\text{rest}}(i,j) := \frac{\langle R_i, R_j \rangle_F}{\|R_i\|_F \|R_j\|_F} \qquad \text{(non-head alignment)}, \tag{42}$$

$$\tau_k := \frac{\|R_k\|_F}{\|H_k\|_F} \qquad \text{(non-head energy ratio; defined when } \|H_k\|_F > 0). \tag{43}$$

Since $H_k$ is the head block, $A_{\text{head}}(i,j)$ coincides with the head Fisher alignment $A_F^{\text{head}}(i,j)$.

**Definition O.10** (Head/rest weights and profile cosine). Define the following weights:

$$w_{\text{head}} := \frac{\|H_i\|_F \|H_j\|_F}{\|D_i\|_F \|D_j\|_F} = \frac{1}{\sqrt{1 + \tau_i^2}\sqrt{1 + \tau_j^2}}, \tag{44}$$

$$w_{\text{rest}} := \frac{\|R_i\|_F \|R_j\|_F}{\|D_i\|_F \|D_j\|_F} = \frac{\tau_i \tau_j}{\sqrt{1 + \tau_i^2}\sqrt{1 + \tau_j^2}} = \tau_i \tau_j \cdot w_{\text{head}}, \tag{45}$$

$$c_\tau := w_{\text{head}} + w_{\text{rest}} = \frac{1 + \tau_i \tau_j}{\sqrt{1 + \tau_i^2}\sqrt{1 + \tau_j^2}} \in (0, 1]. \tag{46}$$

As with $c_r$, we have $c_\tau \leq 1$ by Cauchy–Schwarz, with equality iff $\tau_i = \tau_j$. The equalities involving $\tau_i, \tau_j$ assume $\|H_k\|_F > 0$; otherwise use the norm-based first equality.

**Lemma O.11** (Exact block-diagonal to head decomposition). *For any two tasks $i, j$ satisfying Assumption O.1:*

$$\boxed{A_{\text{blk}}(i,j) = w_{\text{head}} \cdot A_{\text{head}}(i,j) + w_{\text{rest}} \cdot A_{\text{rest}}(i,j)} \tag{47}$$

*This is an **exact identity**.*

*Proof.* Identical to Lemma O.6, with $(D, O, r, w_{\text{blk}}, w_{\text{off}}) \to (H, R, \tau, w_{\text{head}}, w_{\text{rest}})$ and using that $\langle H_i, R_j \rangle_F = 0$ by disjoint block support. $\square$

**Exact gap identity.**

$$\boxed{A_{\text{blk}} - A_{\text{head}} = (c_\tau - 1)A_{\text{head}} + w_{\text{rest}}(A_{\text{rest}} - A_{\text{head}})} \tag{48}$$

**Definition O.12** (Non-head discrepancy). Define the **non-head discrepancy**:

$$\Delta_{\text{rest}} := |A_{\text{rest}} - A_{\text{head}}|. \tag{49}$$

**Corollary O.13** (Sharp bound via profile cosine and non-adversariality). *Since $|A_{\text{head}}| \leq 1$:*

$$\boxed{|A_{\text{blk}} - A_{\text{head}}| \leq (1 - c_\tau) + w_{\text{rest}} \cdot \Delta_{\text{rest}}} \tag{50}$$

*This bound does not require $\tau_i$ or $\tau_j$ to be less than 1.*

## O.4. Combined Full-Network Theorem

**Theorem O.14** (CKA to full-network Fisher: sharp bound). *Under Assumption O.1 and finite fourth moments (so all alignments are well-defined, with $\varepsilon$-regularization in degenerate cases):*

$$|\text{CKA} - A_F^{\text{full}}| \leq \underbrace{|\text{CKA} - A_{\text{head}}|}_{\text{head-layer term}} + \underbrace{\left[(1 - c_\tau) + w_{\text{rest}}\Delta_{\text{rest}}\right]}_{\text{non-head contribution}} + \underbrace{\left[(1 - c_r) + w_{\text{off}}\Delta_{\text{off}}\right]}_{\text{cross-layer contribution}} \tag{51}$$

*where all terms are measurable or can be bounded from activations and gradient samples.*

Whenever a head-layer bound of the form $|\text{CKA} - A_{\text{head}}| \leq \varepsilon_{\text{head}}$ is available, it plugs directly into (51) to yield a closed-form CKA-to-full-network bound (e.g., Theorem G.1 in Appendix G under matched probe/input distributions).

*Proof.* By triangle inequality and the bounds from Corollaries O.8 and O.13:

$$|\text{CKA} - A_F^{\text{full}}| \leq |\text{CKA} - A_{\text{head}}| + |A_{\text{head}} - A_{\text{blk}}| + |A_{\text{blk}} - A_F^{\text{full}}|$$
$$\leq |\text{CKA} - A_{\text{head}}| + \left[(1 - c_\tau) + w_{\text{rest}}\Delta_{\text{rest}}\right] + \left[(1 - c_r) + w_{\text{off}}\Delta_{\text{off}}\right]. \qquad \square$$

**Why this theorem "works" for deep nets.** Unlike worst-case bounds that become vacuous when $r, \tau > 1$, this bound remains informative because it has four measurable terms that are small when:

- **Cross-layer profile mismatch**: $(1 - c_r)$ is small when $r_i \approx r_j$.

- **Head/rest profile mismatch**: $(1 - c_\tau)$ is small when $\tau_i \approx \tau_j$.

- **Adversarial off-diagonal**: $w_{\text{off}} \cdot \Delta_{\text{off}}$ is small when $\Delta_{\text{off}}$ is small.

- **Adversarial non-head**: $w_{\text{rest}} \cdot \Delta_{\text{rest}}$ is small when $\Delta_{\text{rest}}$ is small.

When tasks share similar architectures/training (so $r_i \approx r_j$, $\tau_i \approx \tau_j$), the profile cosines $c_r, c_\tau \approx 1$. When the network exhibits consistent alignment patterns across blocks, $\Delta_{\text{off}}, \Delta_{\text{rest}}$ are small. Both conditions typically hold in practice.

## O.5. Estimating Diagnostics from Gradient Samples

All quantities can be estimated from mini-batch gradient samples:

**Weights and profile cosines.** For each layer $l$, collect per-example gradients $g_l^{(t)}$. Estimate Fisher block norms:

$$\|\hat{\mathcal{F}}_{lm}\|_F^2 \approx \left\| \frac{1}{n} \sum_{t=1}^n g_l^{(t)} g_m^{(t)\top} \right\|_F^2.$$

Then compute $w_{\text{blk}}, w_{\text{off}}, c_r$ from (32)–(35) and $w_{\text{head}}, w_{\text{rest}}, c_\tau$ from (44)–(46). This plug-in estimator is biased for finite $n$; an unbiased alternative is the U-statistic $\frac{1}{n(n-1)} \sum_{t \neq t'} \langle g_l^{(t)}, g_l^{(t')} \rangle \langle g_m^{(t)}, g_m^{(t')} \rangle$.

**Discrepancies $\Delta_{\text{off}}, \Delta_{\text{rest}}$.** Compute cross-task alignments $A_{\text{off}}, A_{\text{rest}}, A_{\text{blk}}, A_{\text{head}}$ from gradient inner products, then take differences.

## O.6. Exact Full-Network Decomposition with Depth Coupling

While the previous subsections provide bounds relating full-network Fisher to head Fisher, here we establish an **exact identity** analogous to Theorem 4.2 but for the full network.

**Definition O.15** (Depth-profile vectors). For independent samples $x \sim \mathcal{D}_i$ and $x' \sim \mathcal{D}_j$, with layer-wise activations $a_{\ell-1}^{(i)}, a_{\ell-1}^{(j)}$ and backpropagated errors $\delta_\ell^{(i)}, \delta_\ell^{(j)}$, define:

$$u \in \mathbb{R}^L, \quad u_\ell := \langle a_{\ell-1}^{(i)}, a_{\ell-1}^{(j)} \rangle \quad \text{(forward similarity profile)}, \tag{52}$$

$$v \in \mathbb{R}^L, \quad v_\ell := \langle \delta_\ell^{(i)}, \delta_\ell^{(j)} \rangle \quad \text{(backward similarity profile)}. \tag{53}$$

**Rank-1 per-layer gradients.** For affine layers applied to a single activation vector (no parameter sharing), the per-example gradient satisfies $g_\ell = \text{vec}(\delta_\ell a_{\ell-1}^\top)$ (Lemma 4.1). In this rank-1 setting, the full gradient inner product factorizes across depth:

$$g^{(i)\top} g^{(j)} = \sum_{\ell=1}^L \langle a_{\ell-1}^{(i)}, a_{\ell-1}^{(j)} \rangle \cdot \langle \delta_\ell^{(i)}, \delta_\ell^{(j)} \rangle = \sum_{\ell=1}^L u_\ell v_\ell = u^\top v. \tag{54}$$

Biases can be incorporated by augmenting activations with a constant 1, as in the head-layer analysis.

**Definition O.16** (Stacked second moments). For task $k$, define block-diagonal matrices of stacked second moments:

$$M_{A,k} := \text{blkdiag}\big(\mathbb{E}[a_0 a_0^\top], \ldots, \mathbb{E}[a_{L-1} a_{L-1}^\top]\big), \tag{55}$$

$$\Gamma_{\Delta,k} := \text{blkdiag}\big(\mathbb{E}[\delta_1 \delta_1^\top], \ldots, \mathbb{E}[\delta_L \delta_L^\top]\big). \tag{56}$$

**Definition O.17** (Full-network coupling coefficient). Let $\mathbb{E}_{i,j}$ denote expectation over independent draws $x \sim \mathcal{D}_i$ and $x' \sim \mathcal{D}_j$ (and the resulting depth-profile vectors $u, v$ from Definition O.15). Define:

$$\chi_{ij}^{\text{full}} := \frac{\mathbb{E}_{i,j}[(u^\top v)^2]}{\mathbb{E}_{i,j}[\|u\|^2] \cdot \mathbb{E}_{i,j}[\|v\|^2]}, \quad \rho_{ij}^{\text{full}} := \frac{\chi_{ij}^{\text{full}}}{\sqrt{\chi_{ii}^{\text{full}} \chi_{jj}^{\text{full}}}}. \tag{57}$$

**Theorem O.18** (Full-network three-factor decomposition). *For an $L$-layer feedforward network with affine weight matrices and rank-1 per-layer gradients (no parameter sharing), full-network Fisher alignment satisfies the exact identity:*

$$\boxed{A_F^{\text{full}}(i,j) = S_{\text{cov}}(M_{A,i}, M_{A,j}) \cdot S_{\text{cov}}(\Gamma_{\Delta,i}, \Gamma_{\Delta,j}) \cdot \rho_{ij}^{\text{full}}} \tag{58}$$

*This is structurally identical to Theorem 4.2 (head layer), with the single-layer quantities replaced by depth-stacked versions.*

*Proof.* The proof follows the same structure as Theorem 4.2.

From (54), the Fisher inner product becomes:

$$\langle \mathcal{F}_i, \mathcal{F}_j \rangle_F = \mathbb{E}_{i,j}[(g^{(i)\top} g^{(j)})^2] = \mathbb{E}_{i,j}[(u^\top v)^2]. \tag{59}$$

For the stacked second moments (with independent draws):

$$\langle M_{A,i}, M_{A,j} \rangle_F = \sum_{\ell=1}^L \text{tr}\big(\mathbb{E}[a_{\ell-1}^{(i)} a_{\ell-1}^{(i)\top}] \cdot \mathbb{E}[a_{\ell-1}^{(j)} a_{\ell-1}^{(j)\top}]\big) \tag{60}$$

$$= \sum_{\ell=1}^L \mathbb{E}_{i,j}[u_\ell^2] = \mathbb{E}_{i,j}[\|u\|^2], \tag{61}$$

$$\langle \Gamma_{\Delta,i}, \Gamma_{\Delta,j} \rangle_F = \sum_{\ell=1}^L \mathbb{E}_{i,j}[v_\ell^2] = \mathbb{E}_{i,j}[\|v\|^2]. \tag{62}$$

Define $\chi_{ij}^{\text{full}}$ such that $\mathbb{E}_{i,j}[(u^\top v)^2] = \chi_{ij}^{\text{full}} \cdot \mathbb{E}_{i,j}[\|u\|^2] \cdot \mathbb{E}_{i,j}[\|v\|^2]$.

Similarly, $\|\mathcal{F}_i\|_F^2 = \chi_{ii}^{\text{full}} \cdot \mathbb{E}_{i,i}[\|u\|^2] \cdot \mathbb{E}_{i,i}[\|v\|^2]$ and likewise $\|\mathcal{F}_j\|_F^2 = \chi_{jj}^{\text{full}} \cdot \mathbb{E}_{j,j}[\|u\|^2] \cdot \mathbb{E}_{j,j}[\|v\|^2]$. Normalizing to alignment form and defining $\rho_{ij}^{\text{full}} = \chi_{ij}^{\text{full}} / \sqrt{\chi_{ii}^{\text{full}} \chi_{jj}^{\text{full}}}$ yields (58). $\square$

*Remark* O.19 (Beyond rank-1 layers). For convolutional or attention layers with parameter sharing, per-example gradients are sums of outer products across positions. In that regime (54) and Theorem O.18 require modification; Appendix C gives the shared-parameter kernel identity and an unbiased sketch, while the block-decomposition identities here remain exact.

**Geometric interpretation.** The coupling coefficient $\rho^{\text{full}}$ measures **depth interference**: whether forward and backward similarity profiles align in depth space. Since $(u^\top v)^2 = \|u\|^2 \|v\|^2 \cos^2 \theta$ where $\theta$ is the angle between $u$ and $v$:

$$\chi_{ij}^{\text{full}} = \frac{\mathbb{E}_{i,j}[\|u\|^2 \|v\|^2 \cos^2 \theta]}{\mathbb{E}_{i,j}[\|u\|^2] \cdot \mathbb{E}_{i,j}[\|v\|^2]}. \tag{63}$$

When forward and backward similarities co-localize in depth (all layers contribute similarly), $\cos \theta \approx 1$ yields constructive interference. Misalignment across depth yields destructive interference.

**Corollary O.20** (Constant depth profiles imply head = full). *Assume the rank-1 per-layer gradient setting of Theorem O.18. Suppose that for every pair of samples $(x, x')$ used in the Fisher inner products, the forward and backward similarity profiles are constant across depth:*

$$u_\ell(x, x') \equiv u_\star(x, x') \quad \text{and} \quad v_\ell(x, x') \equiv v_\star(x, x') \qquad \forall \ell \in \{1, \ldots, L\}.$$

*Equivalently, for each sample pair the per-layer gradient inner products are identical:*

$$\langle g_\ell(x), g_\ell(x') \rangle \equiv \langle g_L(x), g_L(x') \rangle \qquad \forall \ell.$$

*Then for all task pairs $(i, j)$,*

$$A_F^{\text{full}}(i, j) = A_{\text{head}}(i, j).$$

*Moreover, in this regime the depth-coupling term in Theorem O.18 reduces to the head-layer coupling term (it need not equal 1).*

*Remark* O.21 (A sufficient special case: deep linear orthogonal networks). In a deep linear network $a_\ell = W_\ell a_{\ell-1}$ with no biases and orthogonal weights $W_\ell^\top W_\ell = I$, we have $\langle a_\ell(x), a_\ell(x') \rangle = \langle a_0(x), a_0(x') \rangle$ for all $\ell$. Backpropagated errors satisfy an analogous invariance, yielding constant depth profiles and hence $A_F^{\text{full}} = A_{\text{head}}$. Such networks exhibit perfect dynamical isometry (input–output Jacobian singular values all 1).

**Relation to profile diagnostics.** The profile cosine $c_r$ (Definition O.5) and discrepancy $\Delta_{\text{off}}$ (Definition O.7) from the bound analysis are *indirect* measures of depth coupling. Corollary O.8 bounds the gap between full and block-diagonal Fisher; the exact theorem above shows the gap is controlled by $\rho^{\text{full}}$ deviating from the head-layer coupling.

**Computational note.** The head theorem (Theorem 4.2, $L = 1$) is a special case: the depth profiles reduce to scalars $(u, v \in \mathbb{R}^1)$, so $u^\top v = uv$ and $\rho^{\text{full}} = \rho$.

### O.7. Controlled Transfer Validation (CIFAR-100)

We validate that $S_{\text{cov}}(\Gamma_e)$ predicts transfer in a controlled setting where class overlap can be precisely manipulated.

**Setup.** Using ResNet-18 on CIFAR-100, we train a source task on classes 0–49 (fixed), then measure *linear probe* transfer to target tasks with varying class overlap (0%, 24%, 50%, 74%, 100%; i.e., 0/50, 12/50, 25/50, 37/50, 50/50 shared). Linear probe transfer—training only the classification head while freezing the backbone—matches the head-layer theory assumptions. Because the backbone is shared, representation similarity is constant (CKA $\approx 1$), so any variation in transfer must come from error geometry. We compare $S_{\text{cov}}(\Gamma_e)$ against standard transferability metrics (LogME, LEEP, H-score, NCE) (You et al., 2021; Nguyen et al., 2020; Bao et al., 2019; Tran et al., 2019) that require explicit label information.

**Results.** Table 15 shows that $S_{\text{cov}}(\Gamma_e)$ tracks class overlap (0.00 at 0% overlap, 1.00 at 100%) and predicts linear probe transfer gain as well as dedicated transferability metrics.

**Why CKA fails.** With a frozen backbone, CKA is constant by construction (CKA $\approx 1.00$), so it cannot discriminate transfer quality: all source-target pairs appear identical to representation-only metrics regardless of label compatibility. $S_{\text{cov}}(\Gamma_e)$ captures the error structure that CKA misses, enabling meaningful source ranking.

*Table 15.* Controlled transfer validation (CIFAR-100, ResNet-18, linear probe). $S_{\mathrm{cov}}(\Gamma_e)$ achieves correlation 0.977 with transfer gain, matching LogME (0.976) and exceeding LEEP (0.964).

| Class Overlap | $S_{\mathrm{cov}}(\Gamma_e)$ | CKA | Probe Gain | Probe Acc |
|---|---|---|---|---|
| 100% | 1.00 | 1.00 | 41.8% | 60.8% |
| 74% | 0.63 | 1.00 | 35.8% | 57.3% |
| 50% | 0.44 | 1.00 | 31.1% | 51.4% |
| 24% | 0.20 | 1.00 | 30.2% | 51.3% |
| 0% | 0.00 | 1.00 | 27.8% | 49.5% |

**Baseline comparison.** All metrics achieve high correlation with transfer gain ($\rho > 0.96$) because overlap is monotonically related to transfer. The key advantage of $S_{\mathrm{cov}}(\Gamma_e)$ is scalability: standard LogME/LEEP-style finite-label implementations require explicit label maps or class statistics and are not designed to estimate full vocabulary-scale error covariance; FisherSketch estimates the corresponding joint update geometry in one pass without materializing $\Gamma_e$, using $O(d + K + m)$ working memory plus projection state.

**Limitation.** With only 5 overlap levels, correlation comparisons have limited statistical power. The vocab-scale experiment (100 domains, 24 candidates/target, 4,188 transfer pairs) provides stronger evidence.

## O.8. Vision Transformer Validation

We validate the full-network decomposition (Corollary O.8) on Vision Transformers to address concerns about modern attention-based architectures.

**Experimental setup.** We fine-tune ViT-B/16 (Dosovitskiy et al., 2021) (pretrained on ImageNet-21k) on four downstream classification tasks: CIFAR-100 (100 classes), Flowers102 (102 classes), Oxford Pets (37 classes), and DTD (47 classes). Each task is trained with 5 random seeds, yielding 20 models total. We analyze 50 pairwise task comparisons (40 same-dataset pairs with different seeds, 10 cross-dataset pairs, subsampled). Images are resized to $224 \times 224$, representing a substantial scale increase from the CIFAR-32 experiments in the main text.

**Full parameter partition.** Unlike prior work that tracks only the classification head, we collect per-example gradients across the *complete* parameter partition: patch embedding, all 12 transformer blocks, final layer norm, and classification head (14 parameter groups total). This ensures our validation matches the theorem's assumptions—no layers are omitted. Gradients are computed in float64 precision with 1000 samples per task to ensure numerical stability.

**Head gradient comparability across datasets.** To compare head gradients across datasets with different class counts ($K_{\mathrm{CIFAR}} = 100$, $K_{\mathrm{Flowers}} = 102$, $K_{\mathrm{Pets}} = 37$, $K_{\mathrm{DTD}} = 47$), we embed all error and prediction vectors into a shared $K_{\mathrm{shared}} = 286$ dimensional space. Each dataset occupies a contiguous, disjoint block: CIFAR-100 uses coordinates $[0, 100)$, Flowers102 uses $[100, 202)$, Oxford Pets uses $[202, 239)$, and DTD uses $[239, 286)$. Predictions use masked softmax (logits set to $-\infty$ outside their respective blocks) and one-hot labels are embedded via the shared-output injection, yielding 286-dimensional error vectors with zeros outside their respective blocks. Head gradients thus have uniform dimension $K_{\mathrm{shared}} \times d + K_{\mathrm{shared}} = 286 \times 768 + 286 = 219\,934$ parameters. For cross-dataset pairs, this construction yields *exactly* zero Fisher alignment: the error vectors have disjoint support, so $\langle e_i, e_j \rangle = 0$ for all sample pairs, making head gradient inner products zero.

**Key finding: profile cosines remain near-unity.** Despite the attention-based architecture and larger scale, profile cosines $c_r$ remain extremely high (Table 16):

$$c_r = 0.9995 \pm 0.0011 \quad (\text{mean} \pm \text{std}, n = 50). \tag{64}$$

This confirms that ViT models trained on related fine-tuning tasks share similar cross-layer energy profiles, even though the absolute energy ratio $r \approx 2.75$ substantially exceeds unity (off-diagonal energy exceeds diagonal energy by $\sim 2.7\times$). The deterministic bound is satisfied for all 50 pairs (numerical check), and the mean gap (0.077) closely matches the mean bound (0.078; Table 16).

*Table 16.* ViT-B/16 validation of full-network decomposition (Corollary O.8). 20 models (4 datasets × 5 seeds), 50 task pairs, 224 × 224 resolution. Full parameter partition: patch embedding + 12 transformer blocks + layer norm + head (14 groups). Per-example gradients in float64. Bound satisfaction is reported as a numerical sanity check; tightness is reflected by the gap vs. bound rows.

| Diagnostic | Mean | Range | Interpretation |
|---|---|---|---|
| $c_r$ (profile cosine) | 0.9995 | [0.995, 1.00] | Near-perfect alignment |
| $r$ (cross-layer ratio) | 2.75 | [2.03, 2.97] | Off-diag exceeds diagonal |
| $\Delta_{\text{off}}$ | 0.088 | [0.049, 0.150] | Moderate discrepancy |
| Gap $\lvert A_F^{\text{full}} - A_{\text{blk}} \rvert$ | 0.077 | [0.042, 0.132] | Measured deviation |
| Bound $(1 - c_r) + w_{\text{off}} \Delta_{\text{off}}$ | 0.078 | [0.042, 0.132] | Theoretical bound |
| Bound satisfied (sanity check) | | | 50/50 (100%) |

**Constructed orthogonality under shared-output embedding (sanity check).** Table 4 (main text) provides an illustrative regime on ViT: for cross-dataset pairs (e.g., CIFAR-100 vs. Flowers102), the error structures are orthogonal because disjoint label spaces are embedded via the masked-softmax shared-output embedding (logits set to $-\infty$ outside each block). CKA remains moderately high ($\approx 0.57$) because representations share similar geometry, but $S_{\text{cov}}(\Gamma_e) = 0$ and FA-CKA drops to zero, as required by the construction. (This orthogonality is with respect to the disjoint-block embedding of label spaces; alternative embeddings would define different cross-task error geometry.)

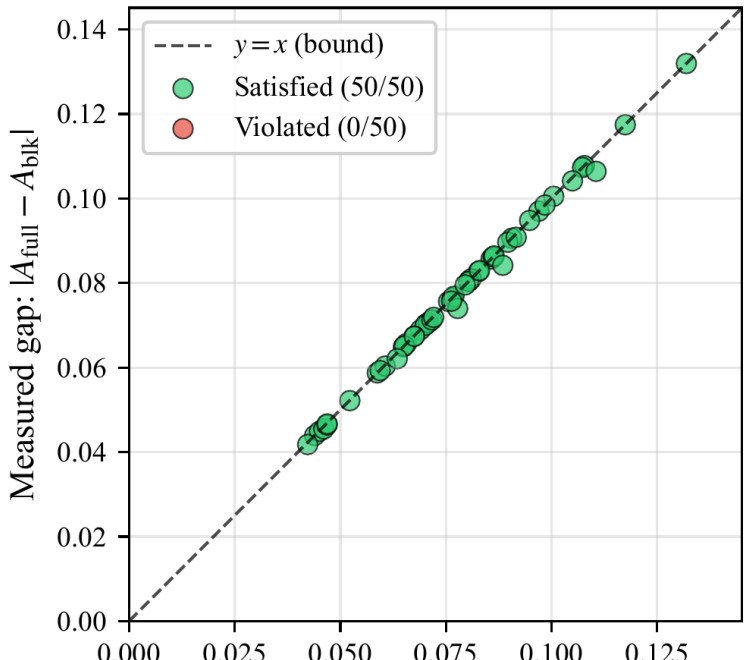

*Figure 3.* ViT-B/16 bound tightness. Each point is a task pair; $x$-axis: theoretical bound $(1 - c_r) + w_{\text{off}} \Delta_{\text{off}}$; $y$-axis: measured gap $\lvert A_F^{\text{full}} - A_{\text{blk}} \rvert$. All 50 points fall below $y = x$ as required by the deterministic bound; proximity to the diagonal indicates tightness.

**Kronecker proxy vs. exact head Fisher.** On the 40 same-dataset pairs (where exact Fisher is nonzero), we compare the Kronecker proxy with exact head Fisher computed via the kernel trick (Appendix A). The Kronecker proxy is *anti-correlated* with exact Fisher ($r = -0.34$, $\rho = -0.30$), while FA-CKA maintains strong correlation ($r = 0.79$, $\rho = 0.90$); see Table 5. This confirms that FA-CKA's validity does not depend on the Kronecker approximation.

*Table 17.* LLM validation of full-network decomposition (Corollary O.8). 30 MMLU subjects, 100 task pairs per model, 100 samples per subject. Parameter-subsampled gradient collection enables tractable gradients across ~8–14B parameters. Bound % is reported as a numerical sanity check; tightness is reflected in gap magnitudes.

| Model | $c_r$ (mean/min) | $\Delta_{\text{off}}$ (mean) | Gap (mean) | Bound % |
|---|---|---|---|---|
| Llama-3.1-8B | 0.998 / 0.978 | 0.392 | 0.332 | 100% |
| Qwen2.5-14B | 0.992 / 0.956 | 0.495 | 0.243 | 99% |

*Table 18.* LLM validation at 70B scale. Same protocol as Table 17: 30 MMLU subjects, 100 task pairs per model, 100 samples per subject. Parameter-subsampled gradient collection enables tractable gradients across 67–72B parameters. Bound % is reported as a numerical sanity check; tightness is reflected in gap magnitudes.

| Model | $c_r$ (mean/min) | $\Delta_{\text{off}}$ | Gap | Bound % |
|---|---|---|---|---|
| Llama-2-70b-hf | 0.982 / 0.920 | 0.527 | 0.344 | 100% |
| Llama-2-70b-chat-hf | 0.990 / 0.952 | 0.594 | 0.464 | 100% |
| Llama-3.1-70B | 0.992 / 0.946 | 0.556 | 0.318 | 100% |
| Llama-3.1-70B-Instruct | 0.963 / 0.725 | 0.579 | 0.408 | 99% |
| Qwen2.5-72B | 0.996 / 0.972 | 0.559 | 0.158 | 100% |
| Qwen2.5-72B-Instruct | 0.993 / 0.935 | 0.298 | 0.278 | 100% |
| deepseek-llm-67b-base | 0.978 / 0.858 | 0.521 | 0.196 | 100% |
| deepseek-llm-67b-chat | 0.954 / 0.796 | 0.754 | 0.436 | 100% |

## O.9. Large Language Model Validation

We extend the full-network decomposition validation to Large Language Models, demonstrating that Corollary O.8 generalizes beyond vision architectures to modern autoregressive transformers.

**Experimental setup.** We evaluate on two architectures: **Llama-3.1-8B** (Grattafiori et al., 2024) (32 layers, 4096 hidden dim) and **Qwen2.5-14B** (Qwen Team, 2024) (48 layers, 5120 hidden dim). Using 30 MMLU subjects as tasks provides a natural multi-task setting with shared $K = 4$ output space (4-way multiple choice), eliminating masked-softmax block-embedding confounds. For each model, we sample 100 task pairs from the 435 possible combinations, collecting 100 gradient samples per subject across the full parameter partition: embeddings, all transformer layers, final layer norm, and a 4-way classification head.

**Empirical tightness of the deterministic bound.** Despite architectural differences (32 vs. 48 layers, different training objectives), both models show $c_r \approx 0.99$ and mean gaps 0.332/0.243 (Table 17). The inequality itself is deterministic, so satisfaction is a numerical check rather than a discovery:

$$|A_F^{\text{full}} - A_{\text{blk}}| \leq (1 - c_r) + w_{\text{off}}\Delta_{\text{off}} \quad \text{for all 100 pairs.} \tag{65}$$

High profile cosines indicate consistent depth profiles in residual-stream architectures.

**Scale validation: 70B parameters.** To validate that the decomposition extends to frontier-scale models, we evaluate on 8 models at 67–72B parameters across three architectures (Table 18). Profile cosines remain high ($c_r \approx 0.98$) and gaps range 0.158–0.464 across models; bound satisfaction is a numerical check rather than a discovery. This suggests the residual stream geometry remains consistent even at $10\times$ scale.

**Comparison with ViT.** LLMs exhibit higher $\Delta_{\text{off}}$ than ViT (0.25–0.75 vs. 0.088), reflecting more cross-layer interaction variance in deeper models (32–80 layers vs. 12 layers). However, the bound remains tight: mean gap tracks mean bound closely across all 10 models tested. Bound satisfaction is expected from the inequality; the key insight is that profile cosines $c_r \approx 0.98$–0.99 dominate the bound, making the block-diagonal reduction reliable even when $\Delta_{\text{off}}$ is elevated.

**Vocabulary-scale transfer validation.** To validate FisherSketch at vocabulary scale, we measure correlation between FisherSketch-estimated Fisher alignment and actual LoRA transfer success. **Protocol**: (1) Compute base perplexity for

each domain; (2) fine-tune LoRA adapter on source domain (rank=8, $\alpha$=16, up to 50 steps with early stopping on source-dev loss, patience=3, lr=$5 \times 10^{-5}$); (3) evaluate on target domain; (4) compute normalized transfer score in loss space: $(\log \text{base}_{\text{ppl}} - \log \text{transfer}_{\text{ppl}})/(\log \text{base}_{\text{ppl}} - \log \text{self}_{\text{ppl}})$, with a small denominator floor (0.01) for stability. Negative transfer (transfer PPL > base PPL) can occur; the normalized score handles this naturally. **Setup**: 100 domains constructed from eight HF datasets (Appendix B.11), 500 samples/domain for FisherSketch ($m$=4096). LoRA transfer uses 200 train / 100 eval samples per domain; we evaluate 4,188 cross-domain pairs (union of uniform + stratified candidates) and report top-1 under the uniform 24-candidate protocol. Sketching and LoRA splits are disjoint by construction; candidate sets and sketch sample indices are saved for audit. **Result**: FisherSketch with SRHT error projections achieves 45.7% $\pm$ 5.0 top-1 source selection ($\approx$10.9$\times$ random) with max regret 0.119 $\pm$ 0.013 (mean $\pm$ std over seeds 43–45), confirming that FisherSketch predicts cross-domain transfer at vocabulary scale ($K$=128,256) where storing $\Gamma_e$ scales as $O(K^2)$. Per-target Spearman corr. $= 0.47 \pm 0.02$; global off-diagonal Spearman corr. $= 0.47 \pm 0.01$ is lower because it mixes heterogeneous task families. A diagonal error-geometry proxy (cosine between $\text{diag}(\Gamma_e)$ vectors; LEEP-style) is weaker: 44.0% $\pm$ 2.2 top-1 and max regret 0.210 $\pm$ 0.011 under the same uniform-24 protocol, indicating off-diagonal error covariance carries signal at vocab scale. Sketch-only stability with fixed LoRA outcomes is moderate: across SRHT seeds 43–45, pairwise top-1 agreement is 0.51 $\pm$ 0.02 and off-diagonal Spearman is 0.982 $\pm$ 0.003.

## O.10. Empirical Findings

**What we measure.**

- **Shallow-wide networks** (depth 1–2): $c_r \approx 0.95$–$0.99$, $c_\tau \approx 0.95$–$0.99$. Profile cosines close to 1 mean tasks have similar energy profiles.

- **Deep networks** (depth $\geq 3$): $c_r, c_\tau$ can be lower when tasks have different $r, \tau$ values, but this does *not* imply the bound is vacuous.

- **Vision Transformers** (ViT-B/16, Table 16): $c_r = 0.9995$, with mean gap 0.077 vs bound 0.078 across 50 task pairs. Full 14-layer partition (patch embedding through classification head) confirms profile alignment persists in modern attention-based architectures.

- **Large Language Models** (8B–70B, Tables 17–18): $c_r \approx 0.95$–$1.00$, with mean gaps 0.158–0.464 across 8B–70B models. The residual stream architecture maintains geometric consistency from 32 to 80 transformer layers.

**The key finding: discrepancies are small.** Even when $c_r, c_\tau < 1$ (deep nets with varying energy profiles), we observe:

- $\Delta_{\text{rest}} \approx 0.10$–$0.20$: non-head alignment tracks head alignment.

- **ViT-B/16**: $\Delta_{\text{off}} \approx 0.05$–$0.15$ (mean 0.088), with mean gap 0.077 vs bound 0.078 across the full 14-layer parameter partition (Figure 3).

- **LLMs** (8B–70B): $\Delta_{\text{off}} = 0.25$–$0.75$ (higher due to deeper architectures), with gaps 0.158–0.464; high profile cosines ($c_r \approx 0.95$–$1.00$) keep the bound tight.

This explains why, in residual-stream architectures with aligned profile cosines, CKA can correlate strongly with full-network Fisher ($r > 0.89$) in our validation suite (Vision Transformers and Large Language Models). This does not contradict non-identifiability for transfer: when error geometry differs, CKA can remain high while head alignment (and transfer) collapse.

**Key takeaway.** The exact decompositions (36) and (47) show that what matters is not "$r < 1$" but "$c_r \approx 1$ and $\Delta_{\text{off}}$ small." Our theory predicts *exactly what to measure*: the profile cosines $c_r, c_\tau$ and discrepancies $\Delta_{\text{off}}, \Delta_{\text{rest}}$. When these are favorable, the head-layer analysis transfers to full-network Fisher—and empirically, they are.

## P. Why Error Geometry Ranks While Activation Geometry Screens

This appendix provides a stylized explanation for the empirical asymmetry observed in Section 6.1: the error-geometry score $S_{\text{cov}}(\Gamma_e)$ can yield higher top-1 source selection accuracy, while incorporating activation geometry reduces worst-case regret. We model source selection for a fixed target task as follows.

Let $\mathcal{S}$ be a finite set of candidate sources. For each $s \in \mathcal{S}$ define a nonnegative *error score* $E_s \in (0,1]$ (e.g., $E_s := S_{\mathrm{cov}}(\Gamma_{e,s}, \Gamma_{e,t})$). Define also a nonnegative *compatibility factor* $A_s \geq 0$ (e.g., an activation/coupling term such as $S_{\mathrm{cov}}(M_{a,s}, M_{a,t})$, optionally folding $\rho$ into $A_s$). The product score is $P_s := A_s E_s$. We compare the error-only selector $\hat{s}_E := \arg\max_{s \in \mathcal{S}} E_s$ and the product selector $\hat{s}_P := \arg\max_{s \in \mathcal{S}} P_s$.

**Lemma P.1** (Tight-race flips under multiplicative universality noise)**.** *Let $\mathcal{C}$ be a finite set and let $\{E_s\}_{s \in \mathcal{C}}$ be deterministic scores with a unique maximizer $s^\star := \arg\max_{s \in \mathcal{C}} E_s$ and $E_s > 0$ for all $s \in \mathcal{C}$. Let $\{Z_s\}_{s \in \mathcal{C}}$ be independent mean-zero $\sigma$-subGaussian random variables, i.e., $\mathbb{E}[\exp(\lambda Z_s)] \leq \exp(\lambda^2 \sigma^2/2)$ for all $\lambda \in \mathbb{R}$. Define $A_s := \exp(Z_s)$ and $P_s := A_s E_s$.*

*Then the probability that the multiplicative factors change the maximizer satisfies*

$$\mathbb{P}\left(\arg\max_{s \in \mathcal{C}} P_s \neq s^\star\right) \leq \sum_{s \in \mathcal{C} \setminus \{s^\star\}} \exp\left(-\frac{\log^2\left(\frac{E_{s^\star}}{E_s}\right)}{4\sigma^2}\right).$$

*In particular, letting $E_{(2)} := \max_{s \in \mathcal{C} \setminus \{s^\star\}} E_s$ and $\Delta := \log(E_{s^\star}/E_{(2)})$, we have*

$$\mathbb{P}\left(\arg\max_{s \in \mathcal{C}} P_s \neq s^\star\right) \leq (|\mathcal{C}| - 1) \exp\left(-\frac{\Delta^2}{4\sigma^2}\right).$$

*Proof.* Fix $s \neq s^\star$. If $P_s \geq P_{s^\star}$ then

$$Z_s - Z_{s^\star} \geq \log\left(\frac{E_{s^\star}}{E_s}\right).$$

Since $Z_s$ and $Z_{s^\star}$ are independent $\sigma$-subGaussian, $Z_s - Z_{s^\star}$ is $\sqrt{2}\sigma$-subGaussian, hence $\mathbb{P}(Z_s - Z_{s^\star} \geq t) \leq \exp(-t^2/(4\sigma^2))$. Applying this with $t = \log(E_{s^\star}/E_s)$ and union bounding over $s \in \mathcal{C} \setminus \{s^\star\}$ yields the first inequality. The second follows because $\log(E_{s^\star}/E_s) \geq \Delta$ for all $s \neq s^\star$. □

**Theorem P.2** (When the product screens catastrophes and can lose top-1 in tight races)**.** *Let $\mathcal{S}$ be a finite candidate set and fix a target. Let $U_s \in \mathbb{R}$ be the (unknown) transfer utility of selecting source $s$, and let $r(s) := U_{s^\star} - U_s$ denote regret, where $s^\star := \arg\max_{s \in \mathcal{S}} U_s$ is the oracle source.*

*Assume there exists a subset $\mathcal{C} \subseteq \mathcal{S}$ of* compatible *sources such that:*

*(A1)* *(Oracle is compatible)* $s^\star \in \mathcal{C}$.

*(A2)* *($E$ ranks within $\mathcal{C}$) There exist deterministic scores $E_s \in (0,1]$ with a unique maximizer over $\mathcal{C}$ at $s^\star$, i.e., $s^\star = \arg\max_{s \in \mathcal{C}} E_s$.*

*(A3)* *(Universality within $\mathcal{C}$) For $c \in \mathcal{C}$, the compatibility factors satisfy $A_c = \exp(Z_c)$ where $\{Z_c\}_{c \in \mathcal{C}}$ are independent, mean-zero, $\sigma$-subGaussian random variables.*

*(A4)* *(Screening outside $\mathcal{C}$) For all $b \notin \mathcal{C}$, $A_b \leq \beta$ for some $\beta \in (0,1)$.*

*(A5)* *(Catastrophic incompatibility) There exists $R > 0$ such that $r(c) < R$ for all $c \in \mathcal{C}$ and $r(b) \geq R$ for all $b \notin \mathcal{C}$ (i.e., catastrophes occur iff an incompatible source is selected).*

*Define the selectors $\hat{s}_E := \arg\max_{s \in \mathcal{S}} E_s$ and $\hat{s}_P := \arg\max_{s \in \mathcal{S}}(A_s E_s)$.*

**(1) Product screening (tail safety).** *If $E_{s^\star} > \beta$, then*

$$\mathbb{P}(\hat{s}_P \notin \mathcal{C}) \leq \exp\left(-\frac{\log^2\left(\frac{E_{s^\star}}{\beta}\right)}{2\sigma^2}\right).$$

*By (A5),*

$$\mathbb{P}(r(\hat{s}_P) \geq R) = \mathbb{P}(\hat{s}_P \notin \mathcal{C}) \leq \exp\left(-\frac{\log^2\left(\frac{E_{s^\star}}{\beta}\right)}{2\sigma^2}\right).$$

**(2) Tight-race degradation (top-1 loss).** *Let $\Delta := \min_{c \in \mathcal{C} \setminus \{s^\star\}} \log(E_{s^\star}/E_c)$. Then*

$$\mathbb{P}(\hat{s}_P \neq s^\star, \, \hat{s}_P \in \mathcal{C}) \ \leq \ (|\mathcal{C}| - 1) \exp\left(-\frac{\Delta^2}{4\sigma^2}\right).$$

*In particular, when $\Delta \lesssim \sigma$ (tight races), the product selector can mis-rank within $\mathcal{C}$ with non-negligible probability.*

**(3) Top-1 comparison from bounds.** *Let $p_{\text{bad}} := \mathbb{P}(\hat{s}_E \notin \mathcal{C})$, where the probability is over any randomness in candidate-set generation or score estimation (if $E$ is deterministic and $\mathcal{S}$ is fixed, then $p_{\text{bad}} \in \{0, 1\}$). Under (A1)–(A2), $\mathbb{P}(\hat{s}_E = s^\star) = 1 - p_{\text{bad}}$. Moreover,*

$$\mathbb{P}(\hat{s}_P = s^\star) \ \geq \ 1 - \exp\left(-\frac{\log^2\left(\frac{E_{s^\star}}{\beta}\right)}{2\sigma^2}\right) - (|\mathcal{C}| - 1) \exp\left(-\frac{\Delta^2}{4\sigma^2}\right).$$

*Define*

$$B \ := \ \exp\left(-\frac{\log^2\left(\frac{E_{s^\star}}{\beta}\right)}{2\sigma^2}\right) + (|\mathcal{C}| - 1) \exp\left(-\frac{\Delta^2}{4\sigma^2}\right).$$

*If $p_{\text{bad}} > B$, then the product rule is guaranteed to have higher top-1 selection probability than the error-only rule. If $p_{\text{bad}} < B$, these bounds do not determine which rule has higher top-1; either ordering is possible depending on the distribution of the multiplicative factors.*

*Proof.* (1) For any $b \notin \mathcal{C}$, $A_b E_b \leq \beta \cdot 1 = \beta$ since $E_b \in (0, 1]$. Thus $\hat{s}_P \notin \mathcal{C}$ implies $\max_{c \in \mathcal{C}} A_c E_c \leq \beta$, which in particular implies $A_{s^\star} E_{s^\star} \leq \beta$. Equivalently $Z_{s^\star} \leq -\log(E_{s^\star}/\beta)$. A subGaussian lower-tail bound gives $\mathbb{P}(Z_{s^\star} \leq -t) \leq \exp(-t^2/(2\sigma^2))$, yielding the first inequality. By (A5), $r(\hat{s}_P) \geq R$ iff $\hat{s}_P \notin \mathcal{C}$, so $\mathbb{P}(r(\hat{s}_P) \geq R) = \mathbb{P}(\hat{s}_P \notin \mathcal{C})$.

(2) Conditional on $\hat{s}_P \in \mathcal{C}$, the event $\hat{s}_P \neq s^\star$ implies that for some $c \in \mathcal{C} \setminus \{s^\star\}$ we have $A_c E_c \geq A_{s^\star} E_{s^\star}$, i.e., $Z_c - Z_{s^\star} \geq \log(E_{s^\star}/E_c) \geq \Delta$. Applying Lemma P.1 (or directly union bounding the same tail event) yields the stated bound.

(3) If $\hat{s}_E \in \mathcal{C}$, then by (A2) the unique maximizer of $E$ on $\mathcal{C}$ is $s^\star$, so $\hat{s}_E = s^\star$. Hence $\mathbb{P}(\hat{s}_E = s^\star) = 1 - \mathbb{P}(\hat{s}_E \notin \mathcal{C}) = 1 - p_{\text{bad}}$. For the product, $\mathbb{P}(\hat{s}_P \neq s^\star)$ is at most the probability of selecting outside $\mathcal{C}$ plus the probability of selecting a suboptimal element within $\mathcal{C}$; combine (1) and (2). Comparing $\mathbb{P}(\hat{s}_E = s^\star) = 1 - p_{\text{bad}}$ with $\mathbb{P}(\hat{s}_P = s^\star) \geq 1 - B$ yields the stated sufficient condition $p_{\text{bad}} > B$ for the product rule to be more accurate in top-1. For $p_{\text{bad}} < B$, no ordering follows from these bounds alone. $\square$

## Q. LEEP as a Diagonal Error-Geometry Proxy

We formalize the claim that LEEP (Nguyen et al., 2020) leverages error geometry. The statement below is deliberately about LEEP, not LogME: LogME is used elsewhere as a finite-label evidence baseline, while this appendix proves only a stylized LEEP–error-geometry connection. Empirically, the diagonal proxy underperforms at vocabulary scale (top-1 accuracy $44.0\% \pm 2.2$; max regret $0.210 \pm 0.011$; Appendix B.11), consistent with the theory below. Let $(X, Y) \sim \mathcal{D}$ be the target distribution with $Y \in [K]$, and let $p(\cdot \mid x) \in \Delta^{K-1}$ be a source classifier's predictive distribution. Let $Z \mid X \sim \text{Cat}(p(\cdot \mid X))$. Define the soft confusion matrix $C_{y,z} := \mathbb{E}[\mathbf{1}\{Y = y\} \cdot p(z \mid X)]$. Define the marginal $q_z := \sum_y C_{y,z}$. Then $C_{y,z} = \mathbb{P}(Y = y, Z = z)$ and $q_z = \mathbb{P}(Z = z)$. Assume $q_z > 0$, and define $p(y \mid z) := C_{y,z}/q_z = \mathbb{P}(Y = y \mid Z = z)$. Assume $p(y \mid z) > 0$ whenever $\mathbb{P}(Y = y, Z = z) > 0$ (or define LEEP $= -\infty$ otherwise). The population LEEP score is

$$\text{LEEP}(p; \mathcal{D}) := \mathbb{E}_{X,Y} \log \sum_{z=1}^{K} p(Y \mid z) \, p(z \mid X) = \mathbb{E}_{X,Y} \log \mathbb{E}_{Z|X}[p(Y \mid Z)]. \tag{66}$$

**Lemma Q.1** (Negative conditional entropy lower bounds LEEP). *Let $Z \mid X \sim \text{Cat}(p(\cdot \mid X))$ be the sampled prediction. Then*

$$\text{LEEP}(p; \mathcal{D}) \ \geq \ -H(Y \mid Z),$$

*with equality when predictions are hard (deterministic).*

*Proof.* By Jensen's inequality on the concave log: $\log \mathbb{E}_{Z|X}[p(Y \mid Z)] \geq \mathbb{E}_{Z|X}[\log p(Y \mid Z)]$. Taking $\mathbb{E}_{X,Y}$ yields the bound; equality holds when $Z$ is deterministic given $X$. $\qquad \square$

**Proposition Q.2** (Hard-prediction LEEP is monotone in diagonal $\Gamma_e^{\mathrm{disc}}$ under uniform confusion)**.** *Assume hard predictions (Z deterministic given X), uniform prior* $\mathbb{P}(Y = y) = 1/K$, $K \geq 2$, *and diagonal confusion:* $\mathbb{P}(Y = y \mid Z = z) = \alpha$ *if* $y = z$, *else* $(1 - \alpha)/(K - 1)$. *This specifies the confusion in the reverse direction—*$\mathbb{P}(Y \mid Z)$ *(conditioned on predictions)— rather than the more common* $\mathbb{P}(Z \mid Y)$. *Let* $\mathbf{1}_Z, \mathbf{1}_Y \in \mathbb{R}^K$ *denote the one-hot vectors for Z and Y, and let* $e^{\mathrm{disc}} := \mathbf{1}_Z - \mathbf{1}_Y$ *and define* $\Gamma_e^{\mathrm{disc}} := \mathbb{E}[e^{\mathrm{disc}} e^{\mathrm{disc}\top}]$. *This is the hard-prediction error; it differs from the soft error* $e = \mathrm{softmax}(Wa) - y$ *used elsewhere. Let* $d := (\Gamma_e^{\mathrm{disc}})_{cc} = 2(1 - \alpha)/K$ *be the diagonal error moment. Then*

$$-H(Y \mid Z) = (1 - \tfrac{K}{2}d)\log(1 - \tfrac{K}{2}d) + \tfrac{K}{2}d \log \tfrac{Kd/2}{K-1}, \qquad (67)$$

*and* $\mathrm{LEEP}(p; \mathcal{D}) = -H(Y \mid Z)$ *is strictly decreasing in d for accuracies above chance* $(\alpha > 1/K)$.

*Proof.* Let $M_{y,z} := \mathbb{P}(Y = y \mid Z = z)$. With the stated symmetry, $M$ is doubly stochastic and has eigenvalues 1 and $\alpha - \frac{1-\alpha}{K-1}$. Viewing $\mathbb{P}(Y)$ and $\mathbb{P}(Z)$ as $K$-vectors, the system $\mathbb{P}(Y) = M\mathbb{P}(Z)$ implies $\mathbb{P}(Z) = \mathbf{1}/K$ for $\alpha \neq 1/K$. At $\alpha = 1/K$, $M$ is rank-1 and $\mathbb{P}(Z)$ is not identifiable from $\mathbb{P}(Y)$ alone; we set $\mathbb{P}(Z) = \mathbf{1}/K$ by convention. Thus $(\Gamma_e^{\mathrm{disc}})_{cc} = \mathbb{E}[e_c^2] = \mathbb{P}(Z = c) + \mathbb{P}(Y = c) - 2\mathbb{P}(Y = c, Z = c) = 2(1 - \alpha)/K$. The conditional entropy is

$$H(Y \mid Z) = -\alpha \log \alpha - (1 - \alpha) \log\left(\frac{1 - \alpha}{K - 1}\right), \qquad (68)$$

so substituting $\alpha = 1 - \frac{K}{2}d$ yields the displayed expression. Differentiating gives

$$\frac{d}{dd}\big(-H(Y \mid Z)\big) = -\frac{K}{2} \log\left(\frac{\alpha(K-1)}{1 - \alpha}\right) = \frac{K}{2} \log\left(\frac{Kd}{2(K-1)(1 - \frac{K}{2}d)}\right),$$

which is negative for $\alpha > 1/K$. Hard predictions imply $\mathrm{LEEP}(p; \mathcal{D}) = -H(Y \mid Z)$ by Lemma Q.1. $\qquad \square$

This shows the conditional-entropy lower bound (and LEEP in the hard-prediction limit) is a monotone function of a diagonal error second moment under the uniform-confusion model. In this stylized regime, LEEP is a "diagonal proxy for error geometry" in a formal sense. This formal connection should not be read as a derivation of LogME; LogME remains a finite-label evidence baseline in our comparisons.

## R. Task Signatures and Retrieval

Algorithm 1 produces per-task mean embeddings $\hat{\mu}_{ae,k} \in \mathbb{R}^m$ as a direct byproduct of computing pairwise Fisher alignments. These embeddings, which we call *task signatures*, enable efficient indexing, retrieval, and dynamic updates without recomputing all pairwise alignments.

### R.1. Signature Storage

**Storage footprint.** Each task signature consists of the joint embedding $\hat{\mu}_{ae} \in \mathbb{R}^m$ stored in float32:

$$\text{Storage per task} = m \times 4 \,\text{bytes} = 16 \,\text{KB at } m{=}4096. \qquad (69)$$

Optionally, the factorized components $(\hat{\mu}_a, \hat{\mu}_e)$ can be stored for $S_{\mathrm{cov}}(\Gamma_e)$ screening, yielding 48 KB per task.

**Compute-time footprint.** During streaming, our implementation maintains six float64 accumulators per task for split-sample estimation (Appendix I); Algorithm 1 shows the simplified single-embedding update. This requires:

$$\text{Streaming state per task} = 6 \times m \times 8 \,\text{bytes} = 192 \,\text{KB at } m{=}4096. \qquad (70)$$

This is transient during signature construction; once signatures are computed, retrieval only requires the float32 $\hat{\mu}_{ae}$ vectors (16 KB per task). If component embeddings $(\hat{\mu}_a, \hat{\mu}_e)$ are also stored (as in FisherAtlas by default), persistent per-task storage is 48 KB, and the FAISS index stores another copy of $\hat{\mu}_{ae}$ for fast search.

**Comparison to Task2Vec.** Task2Vec (Achille et al., 2019) stores the diagonal of the Fisher information matrix, requiring $O(p)$ storage. Here $p$ is the number of parameters. For a 70B-parameter model, this is approximately 280 GB per task (float32). FisherSketch compresses this to 16 KB—a factor of $\sim 10^7$ reduction—while preserving pairwise alignment structure via the product-kernel embedding.

### R.2. Retrieval Procedure

**Prompt signature.** Given a held-out prompt, we compute its signature by streaming over token positions using next-token prediction as the self-supervised objective. Specifically, for a prompt with $T$ tokens, we extract $(a_t, e_t)$ pairs for $t = 1, \ldots, T-1$. Here $a_t$ is the hidden state before the head, and $e_t = \mathrm{softmax}(W_{\mathrm{head}} a_t) - \mathbf{1}_{y_{t+1}}$ is the prediction error. The prompt signature is:

$$\hat{\mu}_{\mathrm{prompt}} = \frac{1}{T-1} \sum_{t=1}^{T-1} \Psi_m(a_t, e_t), \tag{71}$$

where $\Psi_m$ is the factored Random Maclaurin feature map (Section B).

**Nearest-neighbor retrieval.** Given an index of $K$ task signatures $\{\hat{\mu}_{ae,k}\}_{k=1}^K$, retrieval returns:

$$k^* = \arg \max_{k \in [K]} \cos(\hat{\mu}_{\mathrm{prompt}}, \hat{\mu}_{ae,k}) = \arg \max_{k \in [K]} \hat{\mu}_{\mathrm{prompt}}^\top \hat{\mu}_{ae,k}. \tag{72}$$

Since all embeddings are $\ell_2$-normalized, retrieval reduces to a single matrix-vector product.

### R.3. Retrieval Evaluation

We evaluate training-free task retrieval on 700 domains from Natural Instructions (Wang et al., 2022) using Llama-3.1-8B signatures at $m=4096$.

**Protocol.** For each domain, we compute a signature from 200 training samples. Evaluation uses held-out prompts (10 per domain, 6974 total). For each held-out prompt, we compute its signature and retrieve the nearest domain.

*Table 19.* Training-free task retrieval on 700 domains (Natural Instructions, Llama-3.1-8B).

| Method | Top-1 Acc. | Top-3 Acc. | Training |
|---|---|---|---|
| FisherSketch (joint) | 55.5% | 80.4% | None |
| Mean hidden state | 51.7% | 78.6% | None |
| Last token embedding | 54.1% | 77.3% | None |
| Learned linear probe | 62.9% | 86.0% | 345s |
| Learned MLP | 58.9% | 85.2% | 73s |

**Results.** FisherSketch achieves 55.5% top-1 accuracy without any training, compared to 51.7% for mean-hidden baselines. Learned probes achieve higher accuracy (62.9%) but require labeled training data and cannot support open-set addition. A pairwise diagnostic over all 244,650 domain pairs shows substantial activation-dark structure: activation similarity and Fisher alignment have Spearman 0.525, and one nearest-neighbor pair has $S_a = 0.904$ but Fisher alignment 0.088. Under the fixed high-activation/low-Fisher nearest-neighbor diagnostic, 121 of 700 domains have an activation-confusable but update-distinct neighbor, while only 11 show the reverse pattern.

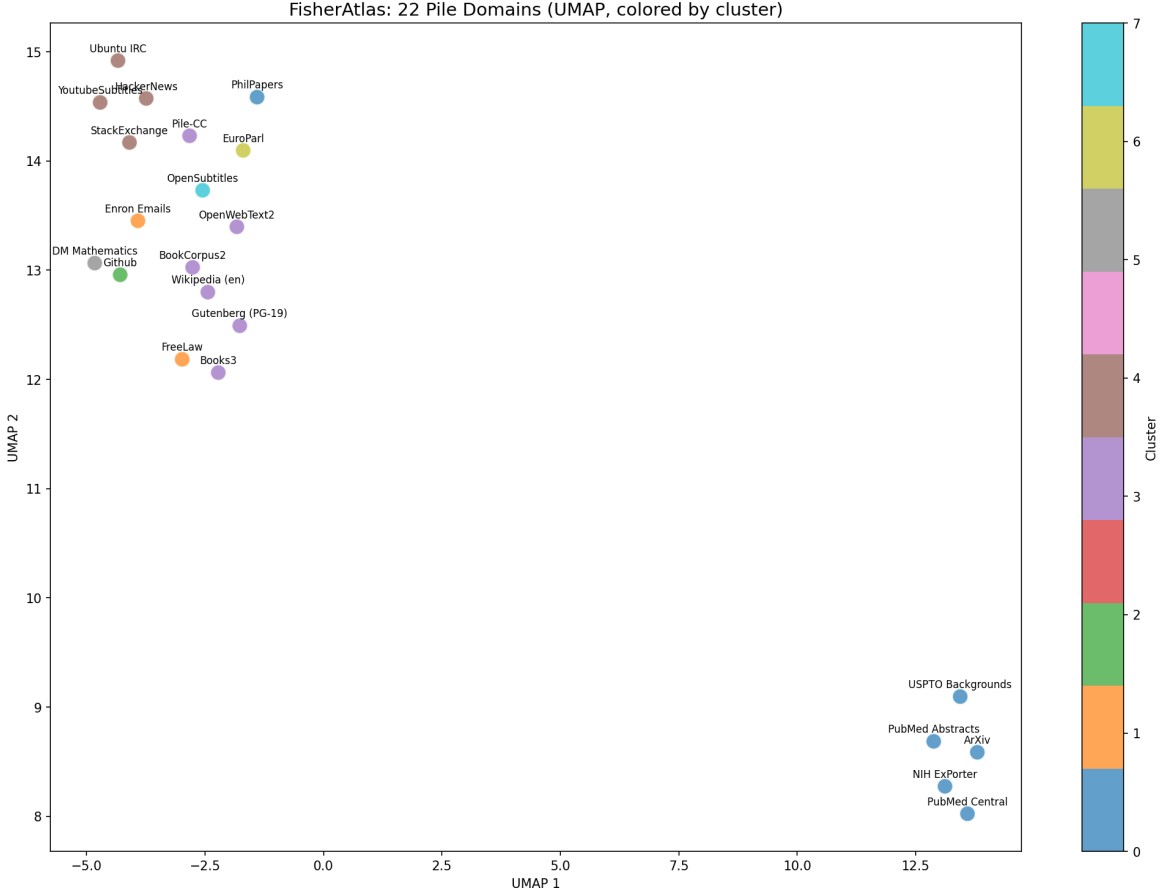

*Figure 4.* FisherAtlas UMAP visualization of 22 Pile domains using FisherSketch task signatures (colored by cluster).

### R.4. Molecular SMILES Proof of Concept

We include a small scientific-sequence check because shared-vocabulary source selection is especially natural for molecular strings. Using Llama-3.1-8B, we form nine molecular SMILES domains (brominated, chlorinated, fluorinated, highly aromatic, large complex, multi-halogen, nitrogen-heavy, simple aliphatic, and sulfur-containing). FisherSketch signatures use $m = 2048$ and 180 molecules per domain; transfer is measured by cross-domain perplexity reduction under a 20-example in-context adaptation protocol, averaged over seeds 42–44 with 50 evaluation molecules per target.

*Table 20.* Molecular SMILES proof of concept. Spearman/Mantel correlations compare pairwise similarity scores to cross-domain perplexity reduction over the 36 off-diagonal domain pairs.

| Score | $\rho_s$ | $p$ |
|---|---|---|
| FisherSketch joint score | 0.530 | 0.0064 |
| Activation-only score | 0.370 | 0.0807 |
| Error-only score | 0.516 | 0.0051 |

The point of this experiment is not to claim a complete chemistry benchmark. It checks the same activation-dark mechanism in a scientific string setting: activation similarity is high and not significant as a transfer predictor at this scale, while the joint FisherSketch score and the error marginal correlate with held-out cross-domain perplexity reduction.

### R.5. Dynamic Addition

**Protocol.** To add a new domain to an existing index:

1. Collect 200 samples from the new domain.

2. Compute the domain signature via a single forward pass (4.7 seconds on an A100).

3. Append the signature to the index.

No retraining is required—signatures are simply appended.

**Evaluation.**    We evaluate on four held-out domains (code, medical, legal, math), each added to an existing 674-domain index. Retrieval accuracy on 30 held-out prompts from each new domain:

*Table 21.* Dynamic addition: retrieval accuracy on held-out prompts from newly added domains.

| New Domain | Held-out Acc. | Time to Add |
|---|---|---|
| code | 100% (30/30) | 4.7s |
| medical | 100% (30/30) | 4.8s |
| legal | 100% (30/30) | 4.6s |
| math | 100% (30/30) | 4.7s |

This is $64\times$ faster than retraining a learned classifier on the expanded domain set.

### R.6. Error Geometry Tracks Label Overlap

The non-identifiability witness in the main text uses the zero-overlap endpoint for clarity. To check that the effect is not merely a corner case, we run a fixed-representation label-overlap sweep in which activation geometry is held constant while the shared-output label support varies.

*Table 22.* Label-overlap continuum. CKA/activation geometry remains fixed, while Fisher alignment falls smoothly as error support diverges. The relative gap is $(\text{CKA} - A_F)/\text{CKA}$.

| Overlap | CKA | Fisher $A_F$ | $S_{\text{cov}}(\Gamma_e)$ | Rel. gap |
|---|---|---|---|---|
| 100% | 1.000 | 0.7881 | 0.9388 | 21.2% |
| 75% | 1.000 | 0.3655 | 0.6473 | 63.4% |
| 50% | 1.000 | 0.3473 | 0.5716 | 65.3% |
| 25% | 1.000 | 0.2536 | 0.3625 | 74.6% |
| 0% | 1.000 | 0.0556 | 0.0558 | 94.4% |

The gap grows continuously as overlap decreases. Thus the theorem isolates a clean endpoint of a graded structural phenomenon: representation metrics see the unchanged activation geometry, whereas Fisher alignment also sees label-conditioned error geometry.

## S. Additional Validation Diagnostics

### S.1. Rho/Delta Calibration for Proxy Reliability

The coupling diagnostic should be used as a warning signal, not as a universal threshold. In the calibration below, lower pairwise $\rho$ and higher $\delta$ coincide with larger gaps between exact Fisher alignment and the separable proxy. High $\rho$ is not a guarantee, but low $\rho$ is a practical red flag that separable activation/error proxies may be unreliable.

*Table 23.* ViT-B/16 calibration of separable-proxy risk. Proxy gap is mean $|A_F - A_{\text{proxy}}|$ over same-dataset pairs.

| Dataset | Mean $\rho$ | Mean $\delta$ | Proxy gap |
|---|---|---|---|
| CIFAR-100 | 0.262 | 0.978 | 0.265 |
| DTD | 0.492 | 0.879 | 0.139 |
| Flowers102 | 0.582 | 0.872 | 0.083 |
| Oxford Pets | 0.914 | 0.798 | 0.026 |

## S.2. Gradient and Diagonal-Fisher Baselines

Table 24 compares FisherSketch with first-moment and diagonal-Fisher baselines in the conditional 100-domain LLM regime. We separate the SRHT row used for the vocabulary-scale backend from a dense-projection check to avoid mixing estimator provenance. FisherSketch improves top-1 over these gradient baselines, while activation-only, error-only, and product proxies remain competitive on rank correlation; this is why the main text avoids unconditional dominance claims.

*Table 24.* Conditional-regime gradient baselines. Top-1/Top-3 are within the fixed candidate set; Spearman is mean target-wise Spearman. The SRHT row matches the vocabulary-scale backend; the dense row is a same-seed projection check and is reported separately because near ties can change Top-1.

| Method | Top-1 | Top-3 | Mean Spearman |
|---|---|---|---|
| FisherSketch (SRHT, EB-shrunk) | 50% | 89% | 0.437 |
| FisherSketch (dense, EB-shrunk check) | 53% | 88% | 0.468 |
| Mean-gradient cosine | 48% | 88% | 0.338 |
| Diagonal-Fisher cosine | 47% | 88% | 0.340 |
| Activation-only | 49% | 87% | 0.411 |
| Error-only | 47% | 87% | 0.492 |
| Product proxy | 46% | 88% | 0.491 |

## S.3. Sketch-Dimension Sweep

Table 25 gives empirical guidance for the 100-domain LLM regime. It should not be overinterpreted as a monotone top-1 law: larger sketches stabilize the estimator, but source-selection accuracy also depends on transfer noise, candidate composition, and near ties. The $m = 4096$ sweep row comes from the SRHT sweep extraction, while Table 24 reports the SRHT EB-shrunk row for the gradient-baseline comparison.

*Table 25.* Sketch-dimension sweep in the 100-domain LLM regime.

| $m$ | F Top-1 | F Top-3 | F Spear. | Proxy Top-1 | Proxy Top-3 | Proxy Spear. |
|---|---|---|---|---|---|---|
| 256 | 50% | 85% | 0.438 | 46% | 91% | 0.532 |
| 512 | 52% | 89% | 0.476 | 41% | 87% | 0.464 |
| 1024 | 45% | 92% | 0.414 | 51% | 87% | 0.479 |
| 2048 | 47% | 86% | 0.467 | 49% | 88% | 0.485 |
| 4096 | 49% | 89% | 0.437 | 47% | 90% | 0.487 |
| 8192 | 51% | 89% | 0.479 | 52% | 88% | 0.487 |

# T. Heterogeneous Outputs and Internal-Layer Directional Fisher

Head-level Fisher alignment requires a shared output basis. When label spaces or heads differ, head Fisher is undefined for cross-dataset pairs rather than merely weak. We therefore treat internal-layer Fisher descriptors as gradient diagnostics rather than replacements for representation metrics. Table 26 reports per-target Spearman correlations from the heterogeneous-output ViT-B/16 validation. We use the target-normalized directional descriptors for the internal Fisher rows; raw symmetric block Fisher was near zero or negative in this experiment and should not be interpreted as the successful signal.

*Table 26.* Heterogeneous-output validation with per-target Spearman only. Internal directional Fisher descriptors are complementary diagnostics; CKA remains competitive in this validation.

| Metric | Linear probe | LoRA |
|---|---|---|
| CKA | 0.671 | 0.664 |
| LogME | 0.357 | 0.243 |
| Directed internal Fisher, MLP6 | 0.221 | 0.286 |
| Directed internal Fisher, QKV/block | 0.214 | 0.321 |
| Directed block descriptor, 4 layers | 0.121 | 0.243 |
| Raw symmetric block Fisher | $-0.071$ | $-0.064$ |

**One-step LoRA tangent bridge.** For a layer $W \in \mathbb{R}^{d_{\text{out}} \times d_{\text{in}}}$ with loss gradient $G = \nabla_W \ell$, a LoRA parameterization writes the update as $BA$ with $B \in \mathbb{R}^{d_{\text{out}} \times r}$ and $A \in \mathbb{R}^{r \times d_{\text{in}}}$ (Hu et al., 2022). At the common initialization used in many LoRA implementations ($B = 0$, $A$ fixed), one infinitesimal gradient step gives $B^+ = -\eta G A^\top$ and hence the induced weight update

$$\Delta W = B^+ A = -\eta G(A^\top A).$$

Therefore, for a fixed LoRA tangent subspace, one-step update inner products are

$$\langle \Delta W_i, \Delta W_j \rangle_F = \eta^2 \operatorname{tr}\!\big((A^\top A) G_i^\top G_j (A^\top A)\big),$$

i.e., a gradient/Fisher alignment after a fixed low-rank projection. This is the intended bridge between internal Fisher descriptors and LoRA transfer. It does not characterize multi-step nonlinear LoRA training, and Table 26 should be read as diagnostic evidence rather than a dominance claim over CKA.

The stored descriptor is still $m$ floats per task per layer after pooling, but internal shared-token collection has a larger transient state. At $n = 500$ and $m = 4096$, the transient sketch-feature state is about 8 MiB per task per layer in fp32; in this run, four internal layers across eight models used about 250 MiB of transient feature state and peaked at 4.19 GiB GPU memory during Fisher collection. For the QKV validation, pooled sketch descriptors store about $432\times$ fewer values than saving all per-example QKV gradients.

## U. Verbalizer-Shift Experiment Details

This section provides full experimental details for the verbalizer-shift controlled experiment (Section 6.2 of the main text). This experiment demonstrates that FisherSketch predicts transfer in a regime where representation-only metrics are maximally uninformative.

### U.1. Experimental Design

**Core insight.** For prompt-based binary classification with a shared prefix, the hidden state at the answer position depends only on the prefix tokens due to causal attention. Therefore, changing only the *label verbalizer tokens* (e.g., from `Yes/No` to `True/False`) yields:

- **Identical activations**: $M_a$ is the same across all verbalizers $\Rightarrow S_{\text{cov}}(M_a) \approx 1$ for all pairs.

- **Different errors**: The error vector $e = \operatorname{softmax}(\ell) - \operatorname{onehot}(y)$ changes because the target token ID differs $\Rightarrow S_{\text{cov}}(\Gamma_e)$ varies across pairs.

This isolates error geometry as the only source of variation, providing a clean test of whether FisherSketch can predict transfer when representation-only metrics are degenerate.

**Connection to Theorem 3.2.** This provides a dense-vocabulary instantiation of the non-identifiability result. Even when representations are identical ($Z_i = Z_j$), head Fisher alignment is not identified because error geometry can differ. Unlike the masked-softmax witness construction, this uses the *full* vocabulary softmax ($K$=128,256) with standard cross-entropy loss.

### U.2. Verbalizer Configuration

We use six single-token verbalizers on Llama-3.1-8B (vocabulary size 128,256). All tokens were verified to be single tokens using the Llama-3.1 tokenizer.

**Flipped verbalizer (`no_yes`).** The `no_yes` verbalizer inverts the label mapping: "No" indicates true, "Yes" indicates false. This tests whether FisherSketch captures output-space structure beyond semantic intuition.

### U.3. Datasets and Prompt Template

**Datasets.** We evaluate on three binary classification datasets:

- **BoolQ** (Clark et al., 2019): Yes/no reading comprehension (9,427 train examples).

*Table 27.* Verbalizer tokens and IDs (Llama-3.1-8B tokenizer). All tokens include a leading space and are single tokens.

| Name | True Token | Token ID | False Token | Token ID |
|---|---|---|---|---|
| yes_no | Yes | 7566 | No | 2360 |
| true_false | True | 3082 | False | 3641 |
| correct_wrong | Correct | 41070 | Wrong | 41856 |
| right_wrong | Right | 10291 | Wrong | 41856 |
| positive_negative | Positive | 45003 | Negative | 51957 |
| no_yes (flipped) | No | 2360 | Yes | 7566 |

- **RTE** (Wang et al., 2019a): Textual entailment (2,490 train examples).

- **CB** (Wang et al., 2019a): Commitment bank, binarized to entailment vs. not (250 train examples).

**Prompt template.** All datasets use the same template (prefix only, no label token in input):

```
Passage: {passage}

Question: {question}

The answer is
```

For RTE/CB, we map hypothesis/premise to the passage/question fields. The model predicts the next token (the verbalizer) given this prefix.

**Data splits.**

- **Sketch**: 500 samples for FisherSketch computation (seed 42/43/44).

- **Train**: 200 samples for LoRA fine-tuning.

- **Eval**: 100 samples for transfer evaluation (50 early-stopping, 50 final eval).

For CB (250 total examples), splits are scaled proportionally: sketch = 118, train = 94, eval = 38. Sketch and LoRA splits are disjoint.

**U.4. FisherSketch Configuration**

- **Sketch dimension**: $m = 4096$

- **Sketch samples**: $n = 500$ per verbalizer (scaled for CB)

- **Error projection**: SRHT on the full $K = 128{,}256$ vocabulary error side; dense Rademacher/Gaussian projections are used only in controlled small-dimensional estimator comparisons

- **Activation extraction**: Last layer hidden state at answer position

- **Error computation**: $e = \mathrm{softmax}(\ell) - \mathrm{onehot}(y)$ over full 128K vocabulary

**U.5. LoRA Fine-Tuning Configuration**

- **LoRA rank**: $r = 8$

- **LoRA alpha**: $\alpha = 16$

- **Target modules**: q_proj, k_proj, v_proj, o_proj

- **Training steps**: 100

- **Learning rate**: $5 \times 10^{-5}$

- **Batch size**: 2 (gradient accumulation 4, effective batch 8)

- **Loss**: Full-vocabulary cross-entropy on the single target token (not restricted to the two verbalizer tokens)

- **Early stopping**: Patience 5 on held-out 50 samples

**Critical implementation detail.**   The loss is computed over the *full* vocabulary:

$$\mathcal{L} = -\log p(y_{\text{verbalizer}} \mid \text{prefix})$$

where $p$ is the full softmax over 128K tokens. We do *not* renormalize over only the two verbalizer tokens—this avoids reintroducing masked-softmax artifacts.

## U.6. Transfer Metric

We use the same normalized transfer metric as the main text (Table 1):

$$U_{s \to t} = \frac{\ell_t^{\text{base}} - \ell_{s \to t}}{\max(\ell_t^{\text{base}} - \ell_t^{\text{self}}, 0.01)}$$

where:

- $\ell_t^{\text{base}}$: Base model loss on target verbalizer (no fine-tuning)

- $\ell_t^{\text{self}}$: Loss after fine-tuning on target verbalizer itself

- $\ell_{s \to t}$: Loss on target after fine-tuning on source verbalizer

**Source selection metrics.**   For each target verbalizer $t$, we select the best source from the other 5 verbalizers:

- **Top-1 accuracy**: Does the predicted source match the oracle (argmax transfer)?

- **% Oracle**: Mean $U_{\text{predicted}}/U_{\text{oracle}}$ across targets.

- **Max regret**: $\max_t (U_{\text{oracle},t} - U_{\text{predicted},t})$.

## U.7. Per-Run Results

*Table 28.* Verbalizer-shift results: per-run breakdown (3 seeds $\times$ 3 datasets = 9 runs).

| Seed | Dataset | Top-1 | % Oracle | Max Regret | $S_a$ Mean |
|------|---------|-------|----------|------------|------------|
| 42 | BoolQ | 83.3% | 100% | 0.560 | 1.0018 |
| 42 | RTE | 66.7% | 100% | 1.169 | 1.0006 |
| 42 | CB | 66.7% | 100% | 0.264 | 1.0018 |
| 43 | BoolQ | 50.0% | 61% | 0.732 | 1.0016 |
| 43 | RTE | 66.7% | 100% | 0.667 | 1.0007 |
| 43 | CB | 83.3% | 100% | 0.795 | 1.0016 |
| 44 | BoolQ | 66.7% | 100% | 1.008 | 1.0015 |
| 44 | RTE | 66.7% | 100% | 0.770 | 1.0004 |
| 44 | CB | 50.0% | 100% | 2.607 | 1.0018 |
| **Aggregate** | | **66.7%** | **95.7%** | 0.952 | 1.0013 |
| Random baseline | | 20% | – | – | – |

**Key observations.**

1. **Activation invariance confirmed**: $S_{\text{cov}}(M_a)$ mean is $1.0013 \pm 0.0006$ across all runs, confirming that activations are identical.

2. **Error divergence confirmed**: $S_{\text{cov}}(\Gamma_e)$ ranges from near-zero to 0.99 across verbalizer pairs.

3. **FisherSketch predicts transfer**: 66.7% top-1 ($3.3\times$ random), 95.7% of oracle.

## U.8. Flipped Verbalizer Analysis

A surprising finding: the flipped verbalizer (no_yes $\to$ yes_no) shows *positive* transfer (U $\approx 0.7$), not negative transfer as naive semantic intuition would predict.

**Explanation.** The base Llama-3.1-8B does not concentrate probability on Yes/No tokens—it spreads mass across the full vocabulary. Fine-tuning on no_yes teaches the model that "the answer is one of {Yes, No}"—this *output space learning* dominates over the semantic mapping.

After fine-tuning on no_yes:

- $p(\text{Yes} \mid \text{true}) \approx 0.30$ (up from $\approx 0.05$ in base model)

- $p(\text{No} \mid \text{true}) \approx 0.60$

Even though the mapping is flipped, $p(\text{correct})$ increased from 5% to 30%, so the loss decreases.

**FisherSketch correctly predicts this.** $S_{\text{cov}}(\Gamma_e)$ between yes_no and no_yes is $\approx 0.9$ (they share the same output tokens), so FisherSketch predicts high transfer—matching the empirical result.

This demonstrates that FisherSketch captures *what fine-tuning actually learns* (output-space structure), not what we semantically assume it learns.

## U.9. Error Similarity Matrix

Table 29 shows $S_{\text{cov}}(\Gamma_e)$ for seed 42, BoolQ. The matrix exhibits interpretable structure:

- Same-token pairs (yes_no $\leftrightarrow$ no_yes) have high similarity ($\approx 0.9$).

- Semantically related pairs (correct_wrong $\leftrightarrow$ right_wrong) have moderate similarity ($\approx 0.3$).

- Unrelated pairs have near-zero similarity.

*Table 29.* $S_{\text{cov}}(\Gamma_e)$ matrix (seed 42, BoolQ). Y/N = yes_no, T/F = true_false, C/W = correct_wrong, R/W = right_wrong, P/N = positive_negative, N/Y = no_yes.

|       | Y/N  | T/F  | C/W  | R/W  | P/N  | N/Y  |
|-------|------|------|------|------|------|------|
| Y/N   | 1.00 | 0.00 | 0.00 | 0.01 | 0.01 | **0.91** |
| T/F   | 0.00 | 1.00 | 0.00 | 0.02 | 0.01 | 0.00 |
| C/W   | 0.00 | 0.00 | 1.00 | **0.30** | 0.01 | 0.00 |
| R/W   | 0.01 | 0.02 | **0.30** | 1.00 | 0.00 | 0.00 |
| P/N   | 0.01 | 0.01 | 0.01 | 0.00 | 1.00 | 0.01 |
| N/Y   | **0.91** | 0.00 | 0.00 | 0.00 | 0.01 | 1.00 |

## U.10. Compute and Reproducibility

**Hardware.** The experiments reported in Appendix U were run on a single NVIDIA A100 (40GB).

**Timing.**

- FisherSketch computation (Phase 1): $\approx$30 minutes per seed (3 datasets $\times$ 6 verbalizers)

- LoRA fine-tuning + transfer evaluation (Phase 2): $\approx$2 hours per seed

- Total: $\approx$7.5 hours for all 3 seeds

**Reproducibility.**    Sketch seeds (42, 43, 44) and LoRA seeds are fixed and released. All indices (sketch/train/eval splits) are deterministic given the seed.

