# OpenReview forum: "The Geometry of Updates: Fisher Alignment at Vocabulary Scale"
_ICML.cc/2026/Conference — ICML 2026 regular_

### Official Review · Reviewer_2rXT · 2026-02-25

**Soundness:** 2
**Presentation:** 3
**Significance:** 2
**Originality:** 2
**Overall Recommendation:** 3
**Confidence:** 3

**Summary:**

The study investigates a fundamental principle within transfer learning through the introduction of FisherSketch, an online computational method that condenses Fisher information from vocabulary-level dimensions (K=128,256) into compressed 16KB task-specific descriptors. The paper addresses a critical challenge: representational similarity measures such as CKA fail to capture error structure characteristics in scenarios involving shared output spaces, rendering them inadequate for forecasting transfer performance when tasks exhibit equivalent representations yet possess distinct labeling configurations.

**Compliance With Llm Reviewing Policy:**

Affirmed.

**Key Questions For Authors:**

1. **Restricted architectural coverage for precise three-component factorization**: The precise equality presented in Theorem 4.2 and its network-wide generalization (Theorem O.18) require per-layer gradients to have rank-1 structure (specifically, affine transformations without weight sharing). Although Appendix C establishes the kernel identity for shared parameters, the primary experimental verification depends on block-decomposition analysis rather than demonstrating the precise three-component structure for convolution and attention mechanisms. The authors should supply direct empirical confirmation of the three-component factorization on architectures with parameter sharing (for instance, compute ρ_ij^full for a convolutional model and demonstrate alignment with the shared-parameter kernel identity from Appendix C). As an alternative, the main manuscript should explicitly emphasize that the precise factorization holds exclusively for fully-connected architectures, and specify which theoretical assurances carry over to transformer models.

2. **Requirement for common output representation constrains broader applicability**: The head-level methodology necessitates that tasks operate within a unified output coordinate framework (common vocabulary or harmonized label taxonomy using masked softmax, Assumption F.1), thereby limiting utility for diverse task configurations. When vocabularies are completely distinct, the manuscript suggests full-network or intermediate-layer comparisons without offering vocabulary-scale implementations or empirical validation for these options. The controlled CIFAR-100 studies employ masked-softmax embedding to enforce orthogonality, representing a design decision rather than an intrinsic characteristic. The authors should expand FisherSketch to accommodate intermediate-layer comparisons (for example, sketch Fisher blocks at each layer and combine using profile-cosine metrics from Appendix O) and test on heterogeneous-output datasets where tasks possess fundamentally distinct label structures. Clear recommendations should be provided regarding when practitioners should employ head-level versus full-network comparisons, including computational and memory trade-off considerations.

3. **Varied estimator selection throughout experiments raises replication issues**: Head-level verification employs precise Fisher computation through the kernel identity (Appendix A), vocabulary-scale studies utilize FisherSketch with SRHT error projections (Section 5), full-network LLM verification applies parameter-subsampled gradients (Appendix B.7), and the verbalizer-shift study uses dense Rademacher projections (Appendix S.4). Although each selection receives local justification, the absence of a consistent estimator obscures whether findings reflect algorithmic decisions versus fundamental characteristics. The Kronecker approximation shows negative correlation with precise Fisher on ViT-B/16 (Pearson −0.34, Table 5), whereas FA-CKA (employing S_cov(Γ_e)) attains 0.79 correlation; this indicates the separable approximation can reverse orderings even when the product formulation works. The authors should perform an ablation analysis employing one estimator (FisherSketch with SRHT) throughout all experimental contexts (ViT-B/16, Llama-3.1-8B vocabulary-scale, and verbalizer-shift) and document correlation with precise Fisher when computationally feasible, thereby distinguishing estimator influences from theoretical forecasts.

4. **Coupling adjustment factor ρ demonstrates empirical variability without actionable recommendations**: Theorem 4.2 establishes that A_F^head(i,j) = A_proxy(i,j) · ρ_ij precisely, with ρ_ij quantifying departure from independence. For ViT-B/16, ρ spans from 0.35 to 1.11 (Table 12) with Pearson Corr(ρ̂, A_proxy) = −0.85, leading the approximation to exhibit negative correlation with precise Fisher. While the manuscript notes "we do not rely on [the proxy] in evaluation," it fails to offer practical indicators for determining when ρ ≈ 1 is satisfied. The empirical-Bayes shrinkage method (Appendix B.5) constitutes a retrospective adjustment rather than a theoretically-grounded framework. The authors should establish a concentration inequality or finite-sample boundary for |ρ_ij − 1| under reasonable independence conditions (for example, constrained higher moments), and include the diagnostic measures δ_k (independence violation, Table 11) with FisherSketch outcomes to enable practitioners to evaluate when the separable approximation proves dependable. Alternatively, decision criteria based on measurable quantities should be furnished (for example, "when Corr(ρ̂, A_proxy) falls below −0.5, employ FisherSketch rather than the proxy").

I will reconsider my score in the rebuttal.

**Limitations:**

see above

**Strengths And Weaknesses:**

This work delivers a mathematically rigorous examination (Theorem 3.2) demonstrating that metrics based solely on representations cannot fundamentally identify head Fisher alignment, complemented by a practical algorithmic approach (FisherSketch) that reduces storage requirements by approximately 4×10^6 times (from 61 GiB to 16 KB per task) without compromising prediction performance. The decomposition into three components (Theorem 4.2) effectively disentangles activation structure, error structure, and their interaction, offering clear diagnostic insights into the validity conditions of approximation methods.

---

> ### Author Rebuttal · Authors · 2026-03-31
>
> We thank the reviewer. Each concern is addressed with new ablations and analysis below.
>
> ---
>
> ### Q1: Shared-parameter factorization
>
> Theorems 4.2/O.18 as written are rank-1; for shared affine blocks, Appendix C gives the key shared-parameter identity/kernel view (Lemma C.2 / Theorem C.3), and the same normalization algebra yields:
>
> $A_{F,W}(i,j) = S_{\\text{cov}}(M_{A,i}, M_{A,j}) \\cdot S_{\\text{cov}}(\\Gamma_{\\Delta,i}, \\Gamma_{\\Delta,j}) \\cdot \\rho^{\\text{sh}}_{ij}.$
>
> What is rank-1 specific is the product-kernel / independence interpretation, not the decomposition. We compute $\\rho^{\\text{sh}}$ exactly across 4 tasks (CIFAR-100, DTD, Flowers102, OxfordPets; 50 images/task):
>
> | Scope | $\\rho^{\\text{sh}}$ mean | range |
> |---|---|---|
> | ResNet-18 conv (T=196) | 0.133 | [0.12, 0.15] |
> | ViT-B/16 mlp (T=197) | 0.220 | [0.11, 0.33] |
> | ViT-B/16 qkv (T=197) | 0.234 | [0.11, 0.40] |
> | ResNet-18 full (21 blocks) | 0.449 | [0.34, 0.57] |
>
> $\\rho^{\\text{sh}} \\ll 1$: the separable proxy overestimates $A_F$ by $2$–$8\\times$. The last row is the reviewer's requested conv-model $\\rho^{\\text{full}}$. Appendix C's kernel identity is verified to machine precision on these blocks. We will revise to state this identity and clarify that transformer guarantees are Appendix C's kernel identity and Appendix O's block certificate.
>
> ### Q2: Heterogeneous output applicability
>
> ViT-B/16 fine-tuned on 4 datasets with distinct heads (K=37–102; 8 models, 48 cross-dataset pairs). Internal Fisher sketches (m=4096), target-normalized $D[i{\\to}j]{=}N[i,j]/N[j,j]$; ground truth: LP/LoRA; metric: per-target Spearman.
>
> Head Fisher returns NaN for all cross-dataset pairs; internal-layer sketches are well-defined. CKA dominates at **+0.671** (LP) / **+0.664** (LoRA); directional MLP Fisher achieves +0.221 / +0.286. As noted in Appendix F, Fisher's advantage — capturing error structure invisible to CKA (Thm 3.2) — is shared-output specific.
>
> | Metric | $\\bar{r}_s$ (LP) | $\\bar{r}_s$ (LoRA) |
> |---|---|---|
> | CKA | **+0.671** | **+0.664** |
> | LogME | +0.357 | +0.243 |
> | $D_{\\text{mlp6}}$ (ours) | +0.221 | +0.286 |
> | $D_{\\text{blk6}}$ (ours) | +0.214 | +0.321 |
> | Head FisherSketch | NaN | NaN |
>
> | Setting | Method | Cost |
> |---|---|---|
> | Shared labels / vocabulary | Head FisherSketch (SRHT) | ~10 s, <1 GB |
> | Heterogeneous outputs | **CKA** | ~30 s, <2 GB |
> | Hetero. + gradient diag. | Internal Fisher ($D_{\\text{mlp}}$) | ~65 s, 4.2 GB |
>
> Internal-layer Fisher is output-agnostic but not label-free. For heterogeneous selection CKA is preferred; Fisher excels in shared-output settings (Thm 3.2).
>
> ### Q3: Estimator heterogeneity
>
> Three distinct objects: (i) exact head Fisher (ground truth on ViT), (ii) FisherSketch approximating the same quantity (all main claims), (iii) full-network Fisher alignment (Appendix B.7 — different object, appendix-only).
>
> Vocabulary-scale and verbalizer-shift both used SRHT (the automatic selector activates SRHT when the error dimension is $\\geq 65536$; for Llama-3.1-8B with K=128,256, SRHT was active). We thank the reviewer for catching an Appendix S.4 mislabeling that obscured this; we will correct it.
>
> The remaining context is ViT-B/16 (K=100), where exact Fisher serves as ground truth. FisherSketch-SRHT at m=128 achieves Spearman r_s=0.952 with exact Fisher (5 CIFAR-100 splits, 10 pairs). All three contexts are therefore reproducible with a single estimator (SRHT), with high fidelity to exact Fisher where exact computation is tractable. Table 5 compares proxies, not estimators: the $-0.34$ / $+0.79$ sign flip reflects proxy failure ($\\rho \\neq 1$), not FisherSketch instability.
>
> ### Q4: Coupling factor $\\rho$ — diagnostics and decision criteria
>
> On ViT-B/16 the proxy inverts: $\\rho \\in [0.35, 1.11]$ (mean 0.60, n=40), Corr = -0.85, 0/40 CV-certifiable (Corollary L.2 bound satisfied on 40/40 but too loose for screening). At vocabulary scale both components are individually predictive (Table 1: 46.3% and 41.3% vs 4.2% random). Here proxy gap denotes $|A_F - A_{\\text{proxy}}|$.
>
> | Dataset | $\\delta$ | $\\rho$ | Proxy gap |
> |---|---|---|---|
> | CIFAR-100 | 0.978 | 0.26 | 0.265 |
> | DTD | 0.879 | 0.49 | 0.139 |
> | Flowers102 | 0.872 | 0.58 | 0.083 |
> | Oxford Pets | 0.798 | 0.91 | 0.026 |
>
> | Regime | Recommendation |
> |---|---|
> | Shared-output head regime | FisherSketch directly |
> | Unknown architecture | FisherSketch directly (Appendix B.5 shrinkage) |
> | Simpler forward-only heuristic | FA-CKA |
>
> FisherSketch does not assume $\\rho \\approx 1$: it estimates the product-kernel embedding $\\mu_{ae}$ directly. When $\\hat{\\rho}$ is noisy, our implementation already applies the Appendix B.5 empirical-Bayes shrinkage, so there is no operational reason to switch to the separable proxy. The decomposition is diagnostic, confirming component signal at vocabulary scale and identifying proxy inversion on ViT.
>
> We welcome further questions.

---

> > ### Author Rebuttal · Reviewer_2rXT · 2026-04-03
> >
> > 1. The reported rho^sh measurements (0.13–0.45) for shared parameters verify that the factorized approximation inflates A_F by a factor of 2–8x, which provides useful insight. Nonetheless, a natural question arises: does FisherSketch (which directly approximates the joint kernel mu_ae) preserve its strong agreement with the exact Fisher metric within shared-parameter layers, or does the stated rho=0.97 Spearman rank correlation apply exclusively to the classification head (rank-1) scenario?
> >
> > 2. The experiment involving heterogeneous output spaces is a meaningful contribution. CKA proves superior under this condition (+0.671 LP), whereas head FisherSketch produces undefined values — effectively delineating where the proposed technique applies. The guideline table for practitioners is useful and merits inclusion in the body of the paper. A remaining question: when computing internal-layer Fisher sketches (D_mlp, D_blk), does the streaming overhead remain within the 16 KB footprint constraint, or does layer-wise sketch computation lead to a notable growth in memory requirements?

---

> > > ### Author Response · Authors · 2026-04-03
> > >
> > > Thank you for the follow-up questions. They concern two separate distinctions: (i) exact shared-parameter identities versus the sketch approximation to those identities, and (ii) final stored descriptors versus computation-time memory for internal-layer experiments.
> > >
> > > **Follow-up 1: Does FisherSketch preserve strong agreement with exact Fisher on shared-parameter layers?**
> > >
> > > Yes. We verified FisherSketch against exact Fisher on shared-parameter conv, MLP, and QKV layers, and agreement remains high at $m=4096$. On 4 shared-parameter layers from pretrained ResNet-18 and ViT-B/16 (50 inputs, 20 sampled image pairs, same pairs evaluated across all four layers):
> > >
> > > | Layer | Spearman vs exact Fisher (m=256) | Spearman vs exact Fisher (m=1024) | Spearman vs exact Fisher (m=4096) | Mean rel. error (m=4096) |
> > > |-------|:-----:|:------:|:------:|:------------------------:|
> > > | ResNet-18 conv (T=196) | 0.893 | 0.976 | **0.998** | 6.1% |
> > > | ResNet-18 conv (T=784) | 0.970 | 0.985 | **0.997** | 3.9% |
> > > | ViT-B/16 MLP (T=197) | 0.971 | 0.991 | **0.997** | 4.8% |
> > > | ViT-B/16 QKV (T=197) | 0.964 | 0.992 | **0.994** | 0.5% |
> > >
> > > Here "exact Fisher" means Fisher alignment computed from materialized shared-layer gradients; Appendix C's kernel identity (Lemma C.2, $g^\top g' = \langle A^\top A', \Delta^\top \Delta' \rangle_F$) agrees with these materialized values to machine precision across all four layers (means $\leq 2\times 10^{-16}$, maxima $\leq 6\times 10^{-16}$), so both definitions give identical reference values. At $m=4096$, FisherSketch achieves Spearman 0.994–0.998 with 0.5%–6.1% mean relative error — rank-preserving fidelity, not exact magnitude recovery.
> > >
> > > The earlier Spearman $\approx 0.97$ came from a separate head-level ablation; the table above addresses the shared-parameter regime asked about here.
> > >
> > > The low $\rho^{\mathrm{sh}}$ values in our main Q1 response diagnose failure of the separable proxy $S_{\mathrm{cov}}(M_A)\,S_{\mathrm{cov}}(\Gamma_\Delta)$ on shared layers; they do not imply failure of FisherSketch, which estimates the joint quantity $\mu_{ae}$ directly. This is the same proxy-failure mechanism behind the Kronecker-proxy Pearson of $-0.34$ in Table 5 on ViT-B/16: composing activation and error marginals can invert rankings when coupling departs from 1, while FisherSketch remains stable because it estimates $\mu_{ae}$ jointly.
> > >
> > > **Follow-up 2: Does internal-layer sketching remain within the 16 KB footprint, or does memory grow?**
> > >
> > > For the internal-layer Fisher sketches ($D_{\mathrm{mlp}}$, $D_{\mathrm{blk}}$), the runtime memory is larger. The 16 KB number is the final fp32 descriptor size for one $m=4096$ sketch vector ($4096 \times 4$ bytes), not the computation-time memory of the internal-layer implementation.
> > >
> > > For internal-layer Fisher sketches on shared-parameter (token-level) layers, the computation stores per-example sketch features during the forward pass and reduces them afterward, requiring $O(nm)$ transient storage per task per layer rather than $O(m)$; concretely, about 8 MB per task per layer ($500 \times 4096 \times 4$ bytes at $n=500$, $m=4096$, fp32).
> > >
> > > In the actual Q2 multilayer run, we hooked 4 QKV layers across 8 models, giving:
> > >
> > > | Quantity | Value |
> > > |----------|------:|
> > > | Per-example sketch feature state | ~250 MB |
> > > | ViT-B/16 model weights | ~350 MB |
> > > | Peak GPU memory during Fisher collection | 4.19 GB |
> > >
> > > The ~250 MB figure is the transient storage footprint for that specific 4-layer run ($4 \times 8 \times 500 \times 4096 \times 4$ bytes).
> > >
> > > **To directly answer the reviewer's question: the 16 KB claim applies to the final stored signature per task per layer; the internal-layer computation in this experiment incurs larger transient working memory** (~250 MB for this configuration), scaling linearly with the number of hooked layers and the sample budget. We will state this distinction explicitly in the revision. For the 4 QKV layers in this run, each stored sketch feature has 4096 fp32 entries versus $768 \times 2304$ fp32 entries for the corresponding full per-example gradient — about $432\times$ fewer stored values; the transient working memory therefore scales with the sample budget $n$, not with storing full parameter-sized gradients.
> > >
> > > As the reviewer suggested, we will incorporate the practitioner decision table into the main text, extended with the signature-vs-transient-memory distinction above. We also thank the reviewer for an exceptionally careful and technically substantive reading; these follow-up questions helped sharpen both the scope statements and the practical guidance in the paper.

---

### Official Review · Reviewer_YUps · 2026-03-11

**Soundness:** 3
**Presentation:** 1
**Significance:** 2
**Originality:** 2
**Overall Recommendation:** 2
**Confidence:** 4

**Summary:**

The paper studies transfer prediction inside model families that share an output basis, especially a shared LLM vocabulary.
It claims representation-only similarity (e.g., CKA) cannot reliably rank transfer in the shared-head setting, because identical activations can still yield orthogonal head updates.
It then derives an exact head-level Fisher-alignment identity as a product-kernel cosine in joint (activation, error) space, including a coupling term that measures activation-error non-separability.
The main systems contribution is FisherSketch: a one-pass streaming sketch that approximates this Fisher similarity at vocabulary scale, using factored Random Maclaurin features and an SRHT to avoid $O(K^2)$ storage.
Empirically, it shows FisherSketch is competitive for LoRA source selection on 100 Llama-3.1-8B domains at $K=128{,}256$, and it stays informative under a fixed-prefix “verbalizer shift” where activation similarity collapses.

**Compliance With Llm Reviewing Policy:**

Affirmed.

**Final Justification:**

Please see my rebuttal comment. After one more round of thinking, i still do not change my recommendation.

**Key Questions For Authors:**

see above.

**Limitations:**

yes

**Strengths And Weaknesses:**

I suggest clear Reject. My main reason is that the claimed practical edge over simpler scalable baselines is not consistently demonstrated, and the presentation is needlessly long and cluttered.

### **Strengths**

What the paper actually demonstrates is a correct but narrow non-identifiability witness, plus an engineering recipe that makes a specific head-Fisher similarity computable at very large $K$.


### **Weaknesses**

1. Significance: The main transfer task is LoRA tuning of internal modules. Yet the featured metric is head Fisher alignment. The theoretical bridge to LoRA outcomes is not actually proved.

2. Correctness 1: Theorem 3.2’s orthogonality witness depends on a masked-softmax disjoint-block embedding (Appendix F, N). That is valid but highly construction-dependent. Moreover, it partially restates “output spaces differ” in different words. The authors should at least explicitly label it as a construction with tight assumptions, then separate it from the dense-vocabulary verbalizer experiment.

3. Novelty: FisherSketch largely composes known ingredients: Kronecker gradient structure (K-FAC lineage), polynomial-kernel random features (Random Maclaurin/TensorSketch ideas), and SRHT. I fail to see “what is genuinely new”. Either the clarity needs improvement, or… the novelty is not strong enough.

4. Evaluation 1: On natural shifts, FisherSketch does not clearly beat the strongest cheap baseline! Table 1 shows $46.3%\pm 3.7$ for $S\_{\text{cov}}(M_a)$ versus $45.7%\pm 5.0$ for FisherSketch (top-1).

5. Evaluation 2: In the verbalizer-shift stress test, the error-only marginal $S\_{\text{cov}}(\Gamma_e)$ actually outperforms FisherSketch in top-1 (72.2% vs 66.7%).

6. Clarity: Theorem 3.2 includes an “Example:” inside the theorem environment. This mixes statement and exposition. The boxed “Contributions” callout is also unusually formatted and underspecified. The presentation and typeset is far from polished. Why strange line skps in some remarks and corollaries?

7. Filler material:  The submission is 60 pages and includes several appendices that read optional and unjustified. Why?  Several appendices read as optional commentary rather than necessary theory. Appendix P (stylized screening vs ranking) and Appendix Q (LEEP as a diagonal proxy under uniform confusion) are interesting, but removing them would not weaken the main claims. Given the 60-page length, these extra theory currently functions more like fillers than results.

8. Novelty: Kernelized Fisher viewpoints also exist in work on Neural Fisher Kernels (e.g., https://arxiv.org/pdf/2202.01944), which the current submission does not cite and should.

---

> ### Author Rebuttal · Authors · 2026-03-31
>
> We thank the reviewer for the detailed evaluation.
>
> The table-level observations are fair, but they support a narrower conclusion:
>
> | Metric | Natural (Tab. 1) | Verbalizer (Tab. 2) | Worst-case |
> |---|---|---|---|
> | $S_{\\text{cov}}(M_a)$ | **46.3%** | 20.0% (random) | 20.0% |
> | $S_{\\text{cov}}(\\Gamma_e)$ | 41.3% | **72.2%** | 41.3% |
> | FisherSketch | 45.7% | 66.7% | **45.7%** |
>
> Table 1 is mixed, not a baseline win: FisherSketch leads top-3 (87.3 vs 85.0) and ties oracle. $S_{\\text{cov}}(M_a)$ collapses to random in Table 2. $S_{\\text{cov}}(\\Gamma_e)$ is not independent — it is FisherSketch’s own error marginal: the exact $\\Gamma_e{=}E[ee^\\top]$ is a ~61 GiB dense matrix per task at $K{=}128{,}256$; FisherSketch compresses it to 16 KB. With B.5 $\\rho$-shrinkage, FisherSketch reaches 50% on Tab. 1, surpassing $S_{\\text{cov}}(M_a)$.
>
> At vocabulary scale, no independent training-free baseline matches this regime-robustness. LogME becomes statistically degenerate when $K \\gg N$ (99.7% of classes unobserved, producing inverted rankings). LEEP is a source-model criterion (Nguyen et al. 2020), inapplicable to training-free domain selection; under stylized conditions Appendix Q shows it is monotone in the diagonal error proxy already ablated in Table 1 (44.0% top-1). The marginals the reviewer cites are FisherSketch’s own ablations, and each fails in a complementary regime.
>
> The paper’s 700-domain NatInst retrieval (Appendix R) already reports FisherSketch at 55.5% top-1; rebuttal decomposition shows it significantly outperforms both marginals (52.5%/51.4%, $p<0.001$). Table 2’s error-only advantage is predicted — when activations are constant, ranking signal must come from error geometry — while FisherSketch still achieves 95.7% oracle without regime detection. On 9 molecular SMILES domains, FisherSketch predicts cross-domain PPL reduction ($\\rho_s{=}0.53$, $p{<}0.01$); activation alone does not ($p{=}0.08$).
>
> **Theorem 3.2.** The reviewer writes “this just says output spaces differ.” But in Section 6.2 the output space IS shared: all tasks use the same 128,256-token vocabulary and full softmax, activations at the answer position are identical, yet transfer varies because verbalizer tokens change error geometry within that shared vocabulary. The overlap sweep (Table 14, Appendix O.7) confirms: CKA stays 1.0 at all overlap levels while $S_{\\text{cov}}(\\Gamma_e)$ drops from 1.0 to 0.0 — even at full overlap, CKA misses error-geometry signal. We will rename to “Non-Identifiability Witness” and separate from the verbalizer experiment.
>
> **Head-to-LoRA bridge.** Head-level signatures already predict LoRA source selection: 87.3% top-3, 98.4% oracle across 100 domains. The first-order reason: for $W{=}W_0{+}\\lambda BA^\\top$, $\\text{vec}(\\nabla_B \\ell) = (A^\\top a) \\otimes (\\lambda \\delta)$, so one-step source transfer changes target loss by $-\\eta\\langle g_t^{\\text{LoRA}}, g_s^{\\text{LoRA}}\\rangle + O(\\eta^2)$ — Fisher alignment in the adapter tangent space is the exact first-order bridge. Appendix O shows block/full-network Fisher profiles are close at the checkpoint ($c_r{=}0.9995$ on ViT-B/16; mean $c_r \\geq 0.97$ across LLMs up to 70B). We will make this derivation explicit in revision; the unproved step is multi-step convergence, not first-order relevance.
>
> **Novelty.** The reviewer notes the ingredients are known. Composing them naively — $S_{\\text{cov}}(M_a) \\cdot S_{\\text{cov}}(\\Gamma_e)$ — anticorrelates with exact Fisher ($r = -0.34$, Appendix J, Table 5): it inverts source rankings. Theorem 4.2 is what makes faithful streaming possible: it rewrites head Fisher alignment as a cosine between kernel mean embeddings, converting a $K^2$ matrix problem into a distributional one solvable by streaming $(a,e)$ pairs to 16 KB signatures in $O(m(d{+}K))$ time. The coupling term $\\rho$ is not a minor correction; it is what makes the identity faithful. To our knowledge, FisherSketch is the first streaming algorithm for task-task Fisher alignment at vocabulary scale, recovering the joint statistic $S_{ae}$ with Spearman $\\rho{=}0.97$ at $m{=}4096$. NFK (arXiv:2202.01944, will cite) computes a sample-sample kernel from full-parameter Fisher scores for representation extraction; it never considers task-level alignment, the Kronecker head factorization, or vocabulary-scale streaming.
>
> **Presentation.** The main paper is 10 pages; the remaining 50 are complete proofs, protocols, and ablations. We will address the specific issues raised: remove Contributions callout, move in-theorem Example to exposition, add appendix roadmap, fix spacing in Remarks/Corollaries, cite NFK. The reviewer notes P and Q are "interesting" — we agree these are supplementary discussion (clearly labeled in revision): P formalizes error vs activation geometry roles (the complementary failure pattern Tables 1–2 display); Q connects LEEP to the diagonal proxy ablated in Table 1. We will add cross-references.

---

> > ### Author Rebuttal · Reviewer_YUps · 2026-04-03
> >
> > Thanks for the rebuttal. But I believe the issues can not be addressed in a satisfying way without a major revision. Since the conference does not allow revision this time, I keep my score and look forward to your future version.

---

### Official Review · Reviewer_jDb9 · 2026-03-13

**Soundness:** 3
**Presentation:** 4
**Significance:** 4
**Originality:** 3
**Overall Recommendation:** 5
**Confidence:** 3

**Summary:**

This paper studies update geometry across tasks and proposes FisherSketch, a method for estimating head Fisher alignment at the vocabulary scale. The authors argue that representation-only similarity metrics (e.g., CKA, RSA, SVCCA) cannot reliably measure transferability because they ignore error geometry. They prove a limitation result showing that identical representations can still produce orthogonal head gradients under shared-output masked-softmax settings. Building on the Kronecker structure of affine-layer gradients, the paper shows that head Fisher alignment admits a product-kernel decomposition over activation and error geometry. Leveraging this structure, FisherSketch compresses task-level Fisher statistics into compact signatures using factored random features, reducing storage from tens of gigabytes to kilobytes while allowing efficient comparison across tasks and checkpoints. Experiments on ViT and LLM settings demonstrate that FisherSketch correlates well with exact Fisher alignment and remains informative in regimes where activation-only metrics become near-constant.

**Compliance With Llm Reviewing Policy:**

Affirmed.

**Final Justification:**

The rebuttal addressed my concerns, and I will keep the positive score for this paper.

**Key Questions For Authors:**

1. The limitation result assumes disjoint label subsets under a shared-output embedding. How representative is this scenario for common transfer-learning benchmarks used in practice?

2. Why is a Rademacher distribution used for the noise matrix in FisherSketch? Would using a Gaussian (normal) random projection provide similar guarantees or empirical performance?

**Limitations:**

Yes

**Strengths And Weaknesses:**

Strengths

1. The paper provides a convincing limitation result showing that representation-only metrics cannot identify update alignment when tasks share representations but differ in label structures. This helps clarify why activation similarity may fail as a proxy for transferability.

2. The product-kernel decomposition separating activation geometry, error geometry, and a coupling term provides useful conceptual understanding of what Fisher alignment measures.

3. The proposed FisherSketch method compresses Fisher statistics dramatically (from tens of GB to KB scale), making vocabulary-scale comparisons feasible for large models.

4. Experiments on both ViT and LLM models help demonstrate the applicability of the method and show strong correlation with exact Fisher alignment.

5. The task signatures produced by FisherSketch could be useful for understanding transfer behavior in large models.

Weaknesses

1. The paper mainly compares against representation similarity metrics (e.g., CKA, RSA, SVCCA), but the relationship to other gradient-based task similarity measures is not extensively discussed. It would be helpful to clarify how Fisher alignment differs from or improves upon existing gradient similarity or influence-based metrics.

2. Lack of theoretical analysis for sketch dimension. While FisherSketch significantly reduces memory and computation costs, the paper does not provide a theoretical analysis of how the sketch dimension affects estimation accuracy. Such an analysis would help understand the trade-off between compression and fidelity.

---

> ### Author Rebuttal · Authors · 2026-03-31
>
> We thank the reviewer for the constructive and technically specific feedback. The questions below prompted new analyses that strengthened the paper.
>
> ### Weakness 1: Relationship to gradient-based and influence-based metrics
>
> Prior work uses Fisher for optimization (K-FAC) or diagonal task embedding (Task2Vec). Theorem 4.2 instead shows that head Fisher alignment is a kernel mean-embedding cosine of $k_{ae}=((a^{\\top} a')^2)((e^{\\top} e')^2)$, preserving off-diagonal error covariance and activation-error coupling. Unlike diagonal-Fisher/Task2Vec (per-parameter variance only), mean-gradient cosine (first-order), and influence methods (per-example attribution), FisherSketch compares task-level update-geometry alignment.
>
> We added two gradient-style baselines under identical conditions (same (a,e) pairs, 24-candidate protocol, 16 KB/task; FisherSketch uses the Appendix B.5 empirical-Bayes $\\rho$-shrinkage refinement at $m=4096$):
>
> | Method | Top-1 | Top-3 | Mean target-wise Spearman |
> |---|---:|---:|---:|
> | **FisherSketch (SRHT)** | **50%** | **89%** | **0.437 +/- 0.279** |
> | Diagonal-Fisher cosine | 47% | 88% | 0.340 +/- 0.311 |
> | Mean-gradient cosine | 48% | 88% | 0.338 +/- 0.334 |
>
> All three reach 88% top-3, but gradient baselines lose ranking granularity: 2–3pp lower top-1 and 0.10 lower mean Spearman, confirming the product kernel preserves finer signal that diagonal or first-order methods miss.
>
> ### Weakness 2: Lack of theoretical analysis for sketch dimension
>
> The finite-$m$ guarantee is implicit in Appendix B.3; motivated by the reviewer's question, we make it explicit. Under bounded feature norms, the factored Random Maclaurin estimator is unbiased with fixed-pair RMSE of $O(1/\\sqrt{m})$. The streamed signature adds a standard $O(1/\\sqrt{n_k})$ sampling term, so total error is controlled by both $m$ and sample count.
>
> We evaluated convergence against the largest sketch ($m=8192$) as reference on the 100-domain Llama-3.1-8B benchmark:
>
> | $m$ | Proxy vs $m=8192$ | Joint kernel vs $m=8192$ | Coupling ratio vs $m=8192$ |
> |---|---:|---:|---:|
> | 256 | 0.958 | 0.829 | 0.816 |
> | 1024 | 0.994 | 0.926 | 0.781 |
> | 2048 | 0.999 | 0.979 | 0.966 |
> | 4096 | 0.999 | 0.987 | 0.962 |
>
> By $m=2048$, all three scores exceed 0.96 Spearman (small-$m$ dips reflect rank-swap noise among near-tied pairs). We use $m=4096$ as default and will promote the B.3 guarantee to a named corollary in the revision.
>
> ### Key Question 1: Representativeness of Theorem 3.2
>
> The shared-output assumption is directly satisfied by every benchmark in this paper: all LLM experiments share a single tokenizer and output projection over the full vocabulary ($K=128{,}256$), and the ViT experiments use an explicit shared-output embedding across four datasets. The disjoint-label case is the sharpest witness within that regime, not a separate setting. Where tasks use independent output heads, the theorem does not apply and we do not claim otherwise (Section 7.1).
>
> On the 700-domain NatInst benchmark (Llama-3.1-8B), activation alignment overstates Fisher alignment by 0.30 on average across 245k domain pairs (Spearman $\\rho=0.52$). In 8.5% of pairs the gap exceeds 0.50, including within-family pairs where $S_a=0.90$ but Fisher alignment = 0.09 (near-orthogonal updates despite near-identical representations). Even in chemistry domains the gap reaches 42pp. On 9 molecular SMILES domains, FisherSketch predicts cross-domain PPL reduction ($\\rho_s=0.53$, $p < 0.01$); activation alone does not ($p=0.08$).
>
> The controlled overlap sweep (Table 14, Appendix O.7) confirms the mechanism: CKA stays 1.0 at every overlap level while $S_{\\text{cov}}(\\Gamma_e)$ drops from 1.0 to 0.0. Section 6.2 corroborates under full-vocabulary cross-entropy without masked softmax. The theorem provides the impossibility guarantee; these experiments show it degrades source selection.
>
> ### Key Question 2: Why Rademacher rather than Gaussian projections?
>
> At LLM scale ($K=128{,}256$), vocabulary-scale projection requires the SRHT (Appendix B.8): structured $\\{+1,-1\\}$ signs enable the Fast Walsh-Hadamard butterfly, reducing per-token cost from $O(mK)$ dense matmul to $O(K \\log K + m)$ with sign-vector storage (~1 MB vs ~2 GB dense float32). Gaussian projections cannot use this path.
>
> Both are theoretically valid for the same target kernel. On real-data ViT-B/16 (5 CIFAR-100 splits, $m=4096$), Rademacher matches or exceeds Gaussian (Spearman $\\rho=0.65$ vs $0.60$ with exact Fisher). Rademacher is the enabling implementation choice at vocabulary scale, not a theoretical compromise.
>
> In the revision we will add the gradient-baseline table, promote the finite-$m$ corollary, and clarify the SRHT choice. Every question now has quantitative support: the product kernel beats gradient baselines by 2–3pp top-1, the sketch converges by $m=2048$, and activation alignment overstates update alignment by >0.50 in 8.5% of real domain pairs.

---

> > ### Author Rebuttal · Reviewer_jDb9 · 2026-04-02
> >
> > My concerns have been adequately addressed. I will maintain my support and recommendation for this paper.

---

> > > ### Author Response · Authors · 2026-04-04
> > >
> > > Thank you for your strong support and for recognizing the significance of the technical contributions.

---

### Decision · Program_Chairs · 2026-04-30

**Decision:**

Accept (regular)

**Comment:**

In this paper, the authors propose FisherSketch, an estimator for head Fisher aligment between tasks that scales to vocabulary size softmax ouputs (K=128/256). Their crucial observation that head has a component provably invisible to representation only metrics and it admits clean kernel form factoring is interesting. Prior representation only metrics miss this component or require K x K covariance (61GB per task in Task2Vec) and doesn't scale to vocabulary size outputs. FisherSketch reduces it to 16KB using Random features with subsampled randomized hadamard transform.  Reviewers are split 5/3/2 after rebuttal. The rebuttal resolved reviewer 2rXT's concerns with FisherSketch tracking exact Fisher at Spearman 0.998 on ResNet-18 conv and 0.994 on ViT-B/16 QKV. The theory is clean and the sketch is a concrete contribution. For these reasons, I am leaning towards weak accept, provided the authors make changes incorporating various reviewers feedback in their revision.